# Incorporating social knowledge structures into computational models

Koen M. M. Frolichs [1,2] ✉, Gabriela Rosenblau[3] & Christoph W. Korn [1,2] ✉

To navigate social interactions successfully, humans need to continuously learn about the personality traits of other people (e.g., how helpful or aggressive is the other person?). However, formal models that capture the complexities of social learning processes are currently lacking. In this study, we specify and test potential strategies that humans can employ for learning about others. Standard Rescorla-Wagner (RW) learning models only capture parts of the learning process because they neglect inherent knowledge structures and omit previously acquired knowledge. We therefore formalize two social knowledge structures and implement them in hybrid RW models to test their usefulness across multiple social learning tasks. We name these concepts granularity (knowledge structures about personality traits that can be utilized at different levels of detail during learning) and reference points (previous knowledge formalized into representations of average people within a social group). In five behavioural experiments, results from model comparisons and statistical analyses indicate that participants efficiently combine the concepts of granularity and reference points—with the specific combinations in models depending on the people and traits that participants learned about. Overall, our experiments demonstrate that variants of RW algorithms, which incorporate social knowledge structures, describe crucial aspects of the dynamics at play when people interact with each other.

Humans constantly meet new people and interact with them. Successfully navigating social interactions crucially depends on quick and accurate learning about others' personality from brief encounters. Humans are skilled in amassing social information by abstracting from concrete, situation-specific observations to personality traits. However, so far these learning processes have not been described in terms of underlying computational mechanisms. Here, we set out to specify how social knowledge structures shape learning about others.

Rescorla-Wagner (RW) models, that in the cognitive sciences fall under the wide umbrella term of reinforcement learning (RL)[1,2], entail simple and robust algorithms that characterize dynamic learning processes across a wide range of (non-)social tasks[2–10]. In general, RL models have been fundamental for understanding learning about the value of objects or situations regarding how rewarding these are (in the form of food, money, pain, etc.). Specifically, these models describe learning as stepwise reductions of prediction errors (PEs) through outcomes or feedback, where the PE is the difference between a predicted value and the actually received outcome or feedback in a given time step. Learning results from using these PEs to update the estimates of future values.

Akin to non-social learning, social learning can be described by such RL and RW models[8,11]. Variants of these models capture how humans learn from others i.e., observational learning[12–15] and for others[16]. Moreover, studies have shown that they can also account for how humans learn about others on specific characteristics such as trustworthiness, generosity or emotional states[8,17–19].

[1]Institute for Systems Neuroscience, University Medical Center Hamburg-Eppendorf, Hamburg, Germany. [2]Section Social Neuroscience, Department of General Psychiatry, University of Heidelberg, Heidelberg, Germany. [3]Department of Psychological and Brain Sciences, George Washington University, Washington, DC, USA. ✉e-mail: Koen.Frolichs@med.uni-heidelberg.de; Christoph.Korn@med.uni-heidelberg.de

However, these models likely fail to explain the complexities of social learning about others' multi-faceted personality because they only rely on environmental feedback and are typically restricted to learning about the current situation or dimension at hand. Human personality cannot be accurately represented on a one- or low-dimensional continuum (such as nutritional or monetary value).

Research in personality psychology presupposes a multi-dimensional structure to human personality traits[20,21]. A substantial literature in this field has revealed that the vast number of commonly used personality traits can be reduced to a few independent dimension or factors[22]. The Big-Five personality factors (neuroticism, extraversion, openness to experience, agreeableness, and conscientiousness) are the most commonly discussed factors to describe various samples of people across the world[23,24]. According to influential theories, such factors can be represented in human cognition via schemata that bundle frequently co-occurring features[25,26]. As a consequence learning can be influenced by schemata with preconceived personality prototypes (i.e., stereotypes)[26] and the group the other belongs to (in- versus out-group)[27]. In some cases, people might rely more on concrete prototypes and trait-exemplifying behaviours when judging others' personality. In situations people are experienced with, they can abstract from concrete exemplars to more general personality traits and to more fine-grained relations between them[28].

In order to capture the complexities of social learning about others' personality, computational models should implement such prototypes and abstractions to allow for multi-dimensional learning[29,30]. So far, studies that explore learning about others have not investigated personality traits specifically or have looked at single traits[31]. Focusing on learning about a single aspect of personality such as trustworthiness[32,33] or on learning about specific groups of people[34,35] makes learning processes amenable to standard RW models. But models on single aspects do not take the multi-dimensionality of human personality into consideration and thus cannot appropriately describe learning about personality structures in real life.

In this study, we aimed at testing how humans learn about the multi-dimensional personality of others. In order to account for the complexities of learning about human personality, we employed hybrid learning models that weigh prior experience, contextual knowledge, and RW models[36]. We therefore constructed and tested a number of computational models with varying complexities, from a simple linear regression that functions as a baseline to hybrid models that combine standard RW learning models[37] with two social knowledge structures that we refer to as Reference Points (RPs) and Granularity (G).

Reference points capture prior experience and expectations that humans have distilled about many other people (from in- or out-groups). Using these RPs, humans can compare newly encountered people with average people. In this study, RPs thus function as points of comparison.

Granularity refers to the level of detail or abstraction with which humans represent the structure of personality traits. As soon as humans learn some new characteristic of a person, their whole view of that person might change (for better or worse), which implies that they update their expectations across many traits based on new information about a single characteristic. That is, receiving information on a single trait may be generalized to similar traits. In order to do so, humans may use representations (or schemata) of the underlying similarities between personality traits (e.g., if you learn that someone is very polite you will also expect her to be friendly because these two traits are related). Alternatively, when information is sparse or no detail is needed, one can generalize individual traits into coarse representations that distil the most important parts of personality (i.e., akin to the factors of the Big-5).

We expect that the use of both reference points and granularity underlies social learning about others. Different learning situations might require information to be represented on a coarse-or fine-grained level of detail. Using variations of social learning experiments and computational models, we tested how well both reference points and granularity accounted for learning behaviour. Models were evaluated against participants' behavioural data to illuminate what strategies participants might use. Additionally, we used model simulations to quantify the different learning strategies based on their performance (when solving the same learning task as participants). This allowed us to evaluate if participants behaved in accordance with these best performing models given the set of our models. Crucially, several statistical analyses (e.g., regressions & correlations) were conducted to support the model-based analyses.

In recent work, we showed how adults and adolescents learn about others' preferences[38,39] by trading off reference points (in the form of prior knowledge about the average population's preferences) and different levels of granularity (in the form of updating information based on past feedback about preferences for similar items). This type of updating requires social knowledge of the similarities between preferences.

The current study builds on our previous studies by applying a considerably wider range of models to learning about a much more generic type of social information, i.e., others' personality traits in various social contexts.

We tested how humans use social knowledge structures for social learning by comparing a set of models that differently formalize reference points and granularity. Specifically, we modelled granularity on two levels: a fine-grained level specifying the similarities between all relevant traits and a coarse-grained level, which can be seen as dimensionality reduction of the fine-grained level where only the factors of the Big-Five are represented. We also investigated which type of granularity affords best fitting learning outcomes and whether participants used optimal representations based on the group of people they were learning about. Additionally, we tested the potential of our models to capture stereotypes. Taken together, we expected that the winning models combine reference points and granularity to refine computational models of social learning.

## Results

### Experimental outline

In all five experiments included here, participants rated a number of unknown people on personality traits. Participants never met these people. Instead, they were presented—trial-by-trial—with the self-ratings that these unknown people had previously given (Fig. 1). That is, participants could learn from the explicit self-views of other persons (and not via direct interactions).

In each trial, participants rated a person on a given trait (Fig. 1). Directly after each rating participants received feedback about the other person's self-rating on the given trait. Through this feedback, participants could learn about the other person's character traits by adapting their ratings to those of the person in question. In each experiment, participants learned consecutively about four or five other persons, i.e., four or five profiles. Every experiment was changed slightly to test consecutive hypotheses (see Table 1, and the section Differences between experiments in the methods; Supplementary Tables 1–3 list the items used in the experiments). The instructions mentioned the possibility to learn about the other persons but participants were not monetarily incentivized to give an accurate impression of the others' traits.

Before we present the results, we briefly outline the different experiments and models. The first experiment functioned as our baseline from which the other experiments were varied.

Experiment 1 [Real Profiles & Wide Traits] included real feedback from people who had given self-ratings in a previous study on positive and negative traits from all Big-Five factors (Korn et al., 2012). A total of 60 traits were used. A minimum of 8 traits and a maximum of 19 traits were presented per factor (see Supplementary Table 1).

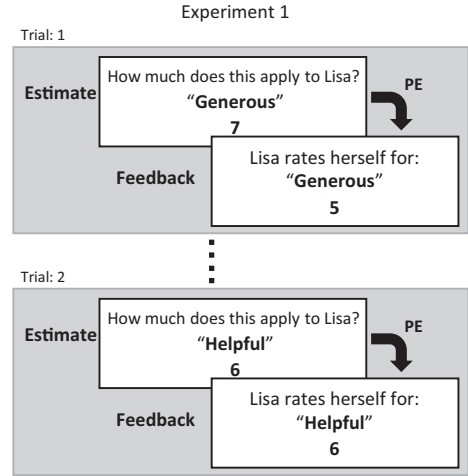

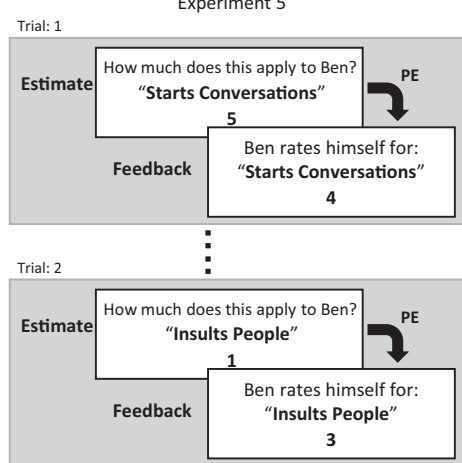

**Fig. 1 | Overview of the experimental tasks.** In this study, we tested computational models of how humans learn about others. In five distinct experiments participants performed a social learning task on several profiles of other persons (with every profile being presented in a separate run). General overview of the learning task: Two trials for Experiment 1 (left) and Experiment 5 (right) are shown. During the learning task, participants estimated which self-ratings a person (here called profile) had given for specific traits (on a Likert scale from 1 does not apply at all to 8 does apply very much). After each estimate, participants received direct feedback in the form of the actual self-rating of that person. This process continued for all traits (in random order). The tasks shown here are translated to English, original German trait words and sentences can be found in Supplementary Tables 1–3.

## Table 1 | Overview of the stimuli for the five experiments

| Experiment | Stimuli | *n* Factors | Profiles | *n* Profiles | Scale | *n* Trials |
|---|---|---|---|---|---|---|
| 1 (Original) | 60 traits | 5 | Real (Online) | 4 | 1–8 | 240 |
| 2 (Constructed Profiles) | 60 traits | 2 | Constructed | 4 | 1–8 | 240 |
| 3 (Two Factors) | 60 traits | 2 | Real (Online) | 4 | 1–8 | 240 |
| 4 (Fashion Models) | 60 traits | 5 | Real (Fashion) | 4 | 1–8 | 240 |
| 5 (IPIP items) | 50 IPIP items | 5 | Real (IPIP) | 5 | 1–5 | 250 |

In Experiments 1–4, we used personality adjectives (i.e., trait words such as generous, diligent; see Supplementary Table 1 for Experiments 1 & 4 and Supplementary Table 2 for Experiments 2 & 3). Sixty adjectives were presented per profile (i.e., for each of the four people about which participants learned). We used traits from all five factors of the Big-Five or traits from only two factors (i.e., agreeableness and conscientiousness). In Experiment 5, for each of the five profiles, participants saw 50 items from the German translation of the IPIP (International Personality Item Pool, which consists of lexical Big-Five factor markers; see Supplementary Table 3). Profiles for Experiments 1 were selected from self-ratings of people from an unrelated sample of a previous lab study (Korn et al., 2012). Profiles for Experiment 2 were constructed by specifying the mean for the two factors and randomly adding noise according to a specified SD. Profiles for Experiment 3 were selected from the self-ratings of participants in Experiment 2. Profiles for Experiment 4 were selected from self-ratings given by a group of female fashion models for a related online study. All four selected persons have worked internationally as fashion models for several years. Profiles for Experiment 5 were selected from self-ratings of a large online dataset on the IPIP with over 1 million participants (Open Source Psychometrics Project; https://openpsychometrics.org/). We selected five profiles with average ratings on 4 out of the 5 factors (mean within 1 SD) but divergent scores on the remaining factor (mean above 1 SD). That is, each profile was divergent on another factor.

Experiment 2 [Constructed Profiles & Narrow Traits] consisted of artificially constructed feedback on 30 positive traits from the two factors agreeableness and conscientiousness. This experiment allowed a tighter experimental control on a narrow set of traits.

Experiment 3 [Real Profiles & Narrow Traits] combined aspects of Experiments 1 & 2 to clarify the differences in their results. Real feedback on 30 traits from each of the two factors agreeableness and conscientiousness was given.

Experiment 4 [Fashion Models] used real feedback from self-ratings of an out-group. We chose fashion models because they are members of an out-group that everybody knows but that few people personally interact with. The same traits as in Experiment 1 were used.

Experiment 5 [IPIP Items] consisted of feedback from real self-ratings on 50 items from the German translation of the International Personality Item Pool (IPIP). For each of the Big-Five factors 10 sentences were presented. This experiment thus aimed at generalizing our findings across different types of items describing personality traits.

### Computational models

In each of the five experiments, we tested (a minimum of) five computational models to explore which model best described participants' learning about others. In our models, we tested two concepts: Reference Points (RPs) and Granularity (G) (Fig. 2). In brief, reference points capture an average person (of a specific group). That is, participants may have an average person in mind when rating an unknown person and use this average person to guide their learning. Granularity refers to the level of generalization during learning, i.e., the level of detail in the similarity between represented traits: Coarse representations of trait relationships imply that participants only consider differences between the Big-Five factors—but do not distinguish between traits within a factor. Fine-grained representations take the similarities of all traits into account. In short, granularity generalizes updating based on PEs across similar items.

In the following, we verbally summarize the five models (please refer to the methods section, Fig. 2 and, Supplementary Fig. 1 for mathematical and conceptual details about the models). Each model is described as if it was learning about one profile, models never combine information or learning across profiles.

Model 1 [No Learning] functions as our baseline model. This regression model entails a simple linear transformation of population means (on trait ratings) as reference points to predict others' traits.

Model 2 [Coarse Granularity] combines Rescorla-Wagner (RW) learning and coarse granularity. That is, it applies coarse granularity to

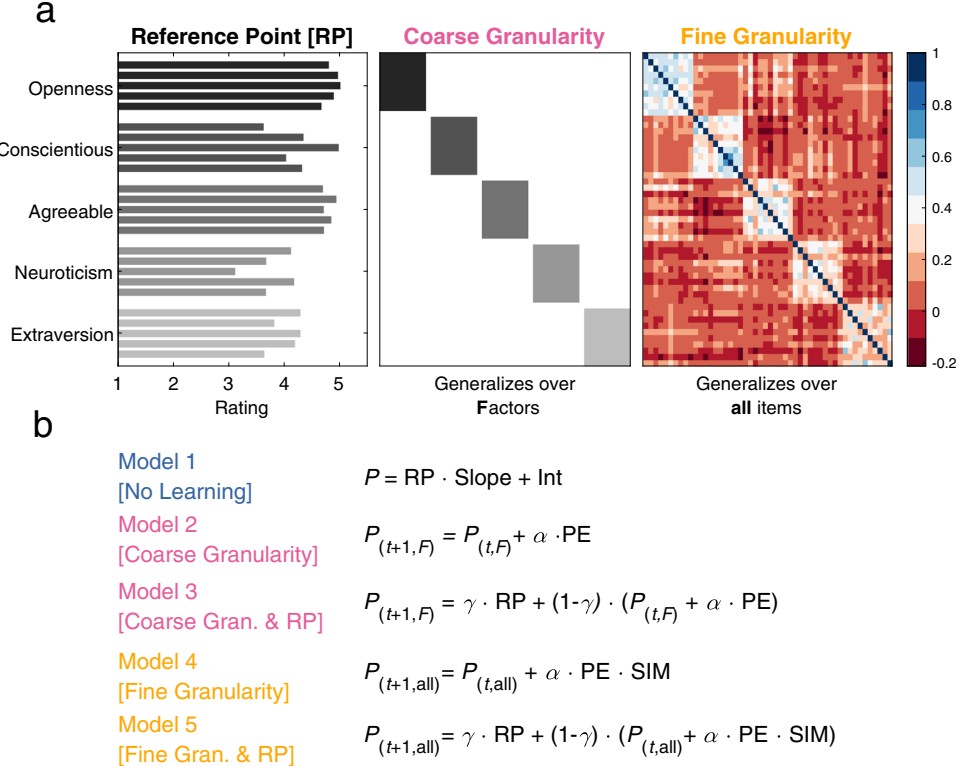

**a**

Reference Point [RP]   Coarse Granularity   Fine Granularity

Openness

Conscientious

Agreeable

Neuroticism

Extraversion

1  2  3  4  5
Rating

Generalizes over **F**actors

Generalizes over **all** items

**b**

Model 1
[No Learning]

$$P = \mathrm{RP} \cdot \mathrm{Slope} + \mathrm{Int}$$

Model 2
[Coarse Granularity]

$$P_{(t+1,F)} = P_{(t,F)} + \alpha \cdot \mathrm{PE}$$

Model 3
[Coarse Gran. & RP]

$$P_{(t+1,F)} = \gamma \cdot \mathrm{RP} + (1-\gamma) \cdot (P_{(t,F)} + \alpha \cdot \mathrm{PE})$$

Model 4
[Fine Granularity]

$$P_{(t+1,\mathrm{all})} = P_{(t,\mathrm{all})} + \alpha \cdot \mathrm{PE} \cdot \mathrm{SIM}$$

Model 5
[Fine Gran. & RP]

$$P_{(t+1,\mathrm{all})} = \gamma \cdot \mathrm{RP} + (1-\gamma) \cdot (P_{(t,\mathrm{all})} + \alpha \cdot \mathrm{PE} \cdot \mathrm{SIM})$$

**Fig. 2 | Overview of the models and knowledge structures.** To explore participants' behaviour we constructed five main computational models that made use of two main knowledge structures: Reference Points (RPs) and Granularity (G). **a** The Reference Points represent what participants can use as a basis for estimating an average person (shown is a selection of student personality trait averages). Participants can use these RPs on each trait to compare this average rating with their current estimate for a specific person. Traits are ordered based on the Big-Five Factors (different shades of grey). Granularity (G) refers to the level of detail in the represented structure of others' personality traits. The granularity matrix generalizes the PEs across similar items in two distinct ways: for coarse granularity it generalizes per Big-Five factor, and for fine granularity it updates every individual trait based on how correlated they are to the current trait. **b** Using both RPs and granularity the models can be divided into three sets, which are depicted in three

different colours. First, No Learning (blue), consists of a single regression model, Model 1 [No Learning] that functions as a baseline model. Second, Coarse Granularity (pink), updates based on the (Big-Five) factor to which the current adjective belongs. Model 2 [Coarse Granularity] uses the standard Rescorla–Wagner (RW) function to update the factor estimates and Model 3 [Coarse Granularity & Population RP] combines Model 2 with information from the RP. Third, Fine Granularity (orange), consists of two models that update all adjectives based on their correlation with the current trait. Model 4 [Fine Granularity] updates all items according to the Fine Granularity and Model 5 [Fine Granularity & Population RP] combines model 4 with information from the RP (see Supplementary Fig. 1 for details on the models). P prediction, Int intercept, RP reference point, α learning rate, PE prediction error, γ weighting parameter, F (generalizes over Factor) coarse granularity, All (generalizes over All items) fine granularity, SIM similarity matrix.

the personality domain (i.e., learns a single value for each Big-Five factor).

Model 3 [Coarse Granularity & Population RP] combines Models 1 & 2 and thus allows to balance participants' use of population means as reference points and learning according to coarse granularity.

Model 4 [Fine Granularity] employs RW learning by updating the estimated values of all the traits on every trial based on how similar the other traits are to the trait that is currently presented.

Model 5 [Fine Granularity & Population RP] combines Models 1 & 4, i.e., population reference points and learning according to fine granularity.

In Experiment 4, three models are added that have the same mathematical equation as Models 1, 3 & 5 but use stereotypical RPs calculated separately for an out-group (i.e., a group of fashion models).

**Experiment 1: real profiles & wide traits**

In our first experiment, participants rated four people on 60 self-ratings of trait words sampled from a list of positive and negative German trait adjectives such as polite and aggressive. These traits were selected to be representative of the Big-Five factors, i.e., adjectives for each Big-Five factor were included albeit not in equal numbers (see Supplementary Table 1). The real self-ratings of the four other persons

were combined with fabricated personal information (e.g., name and age) into four distinct veridical profiles.

Analyses that were not based on computational models indicated that participants were learning i.e., the absolute PEs decreased over trials (Fig. 3c, top). A pairwise Pearson correlation between trial number and the mean of the absolute PE over all participants showed a negative correlation, $r(58) = -0.523$, $p < 0.001$. Furthermore, we conducted a general linear model (GLM) analysis that consisted of three separate regressors to predict participants' answer accuracy (i.e., higher accuracy means lower PE). Each of the regressors represented a substantial part of the models. Regressor 1 captures learning in the standard Rescorla-Wagner model by tracking the total number of previous trials for each item (i.e., if participants are learning, one should see a decrease in PEs over trials). Regressor 2 captures the coarse granularity by tracking the total number of previous trials within a factor for each item (i.e., if participants learn based on each factor, one expects to see a decrease in PEs over trials within this factor). Regressor 3 assesses the fine granularity by computing the summed absolute correlations of the previous items with the current item (i.e., assuming that the correlation is the information density of an item to the current item, one can expect that the sum of all previous items predicts the decrease in PE). Results from this GLM indicated

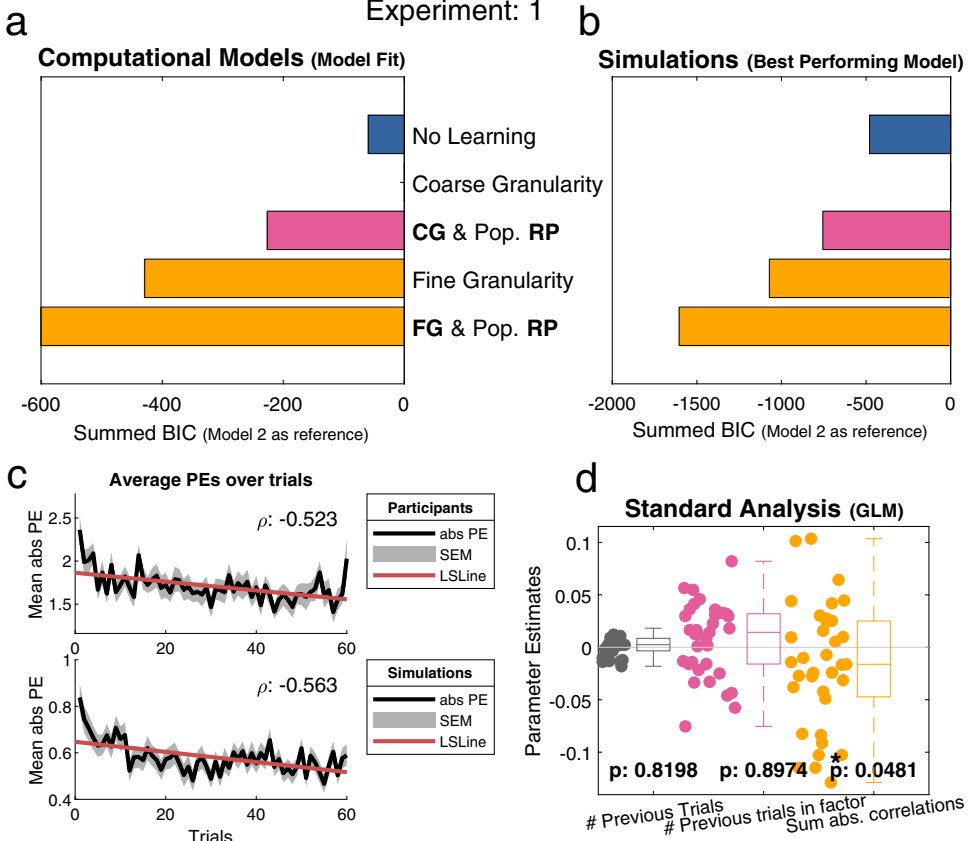

**Fig. 3 | Overview of the main analyses for experiment 1.** In this experiment, participants learned about the personalities of four strangers. Results indicate that participants used fine-grained correlation structures during learning. **a** Model comparison results using fixed-effects analysis (losing model as reference) indicate Model 5 [Fine Granularity (FG) & Population Reference Point (RP)] as the best fitting model ($n = 35$). **b** Simulated data ($n = 35$), the best performing model indicates which of the models performs the task most optimally (see Supplementary Fig. 7). The best performing model (Model 5) demonstrates that participants used the best strategy. **c** A decrease of the prediction errors (PEs) over time can be interpreted as learning. Both plots display the average absolute PEs over time ± SEM. We calculated a pairwise Pearson correlation between trial number and the mean absolute PEs to determine if the PEs decrease over time. Top) Participants' data shows a decrease in the PEs over time ($\rho$:−0.523, least squares line (red)). Bottom) Simulated data using Model 5 shows a similar decrease in PEs over time ($\rho$: −0.563). **d** General

linear model (GLM) on three core model features: (1) Rescorla–Wagner RL, 2) coarse models, and 3) fine models that predict the accuracy per trial per participant. Only the third regressors was significant ($n = 35$), indicating participants' use of fine granularity: (one-sided t-test) regressor 1: $t(34) = 0.927$, $p = 0.8198$, regressor 2: $t(34) = 1.2915$, $p = 0.8974$, regressor 3: $t(34) = −1.7109$, $p = 0.0481$. Individual data points are participants' parameter estimates which are summarized by boxplots (median (middle line), 25th, and 75th percentile (box), the whiskers extend to most extreme data points not considered outliers (1.5 times interquartile range), outliers are indicated with + signs). Conclusions based on this GLM should take into account that all three regressors are highly correlated ($\rho$ between 0.76 and 0.92). [One-sided t-test; * indicates $p < 0.05$, ** indicates $p < 0.001$, no correction for multiple comparisons]. CG coarse granularity, FG fine granularity, RP reference point, # number of, PEs prediction errors, SEM standard error of the mean, LSLine least squares line.

that participants used the fine-grained correlation structures (i.e., regressor 3); Fig. 3d).

Bayesian model comparisons revealed Model 5 [Fine granularity & Population RP] as the best fitting model among our set of five models according to both fixed- and random-effects analyses (Fig. 3a & Supplementary Fig. 2a). That is, participants relied on previous knowledge about an average person when judging the traits of others. Importantly, they not only used population averages as reference points to make estimates but they also scaled the extent to which they updated these estimates by the fine-grained similarity between the currently presented traits and potential other traits in the item set (in line with results from the GLM). These results indicate that participants used a representation of the similarities between traits for their reference group. They represented how single traits relate to one another and used this fine-grained structure in combination with average population ratings to make estimates about others.

To assess the winning model's ability to capture learning we simulated how this model learns (Fig. 3c, bottom). A pairwise Pearson correlation between trial number and the mean of the absolute PE over all simulations showed a negative correlation, $r(58) = −0.563$, $p < 0.001$.

These results are in-line with participants' results. Moreover, we tested which of the five models was the best performing strategy. This analysis disregards participants' behaviour and solely focusses on the models performing the task. These simulations give us an unbiased estimate which model (out of the set of available models) is best suited for the current task. This was achieved by fitting the models on the actual experimental task (i.e., profile responses) rather than participants' responses (see Supplementary Fig. 7 for a detailed explanation of the rationale). Analyses that quantified this model performance corroborated that Model 5 [Fine granularity & Population RP] was also the best performing model for this task (Fig. 3b). In summary, these simulations indicated that participants used a good strategy for learning in this task.

To test the robustness and distinguishability of our models, we performed parameter recovery and calculated a confusion matrix. For parameter recovery, we independently simulated data 200 times using uniformly distributed and randomly sampled parameters with noise added in the last step. Parameter recovery compares known input parameters versus recovered output parameters and uses correlations as a metric of fit, where high correlations indicate better recovery. The correlations for the three parameters of the winning model were

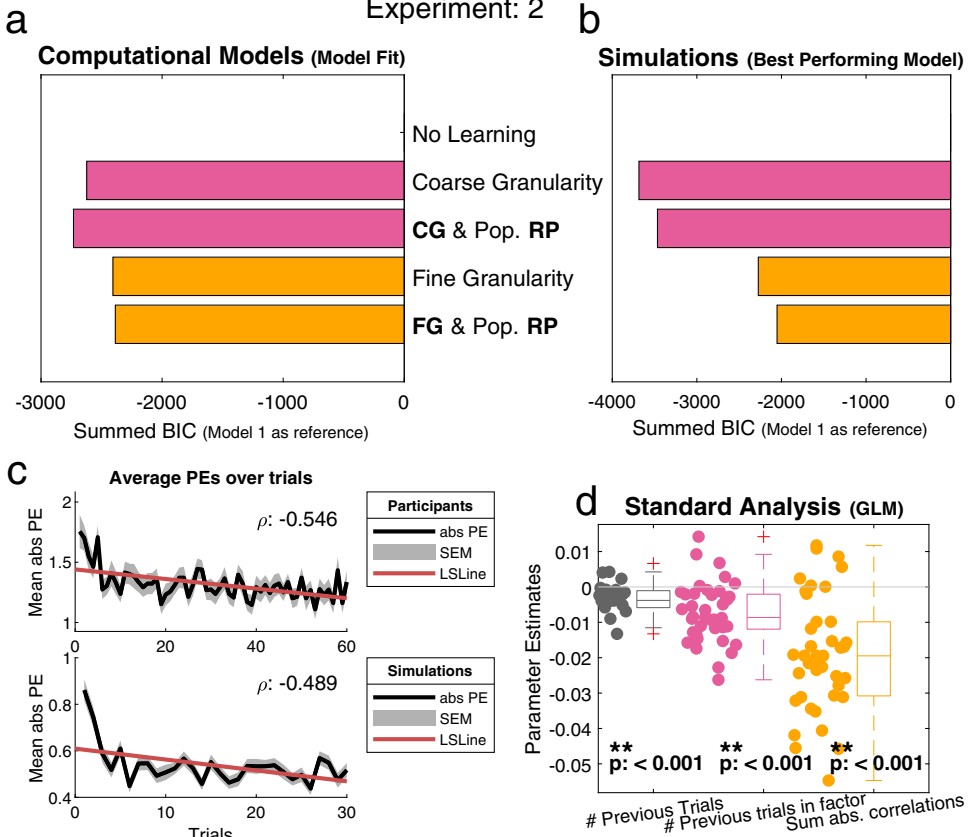

**Fig. 4 | Overview of the main analyses for experiment 2.** In this experiment participants learned about artificial profiles which did not have the trait similarity structures. Results indicate that participants use a coarse granularity structure when less social information is present. **a** Model 3 [Coarse Granularity & Population RP] is the best fitting model ($n=41$). This model uses the average population as a reference point and coarse granularity for generalization. **b** Simulated data for the best performing model ($n=41$). Unlike participants' data, Model 2 [Coarse Granularity] was the best performing model, demonstrating that participants could have used a more optimal strategy. **c** Both plots display the average absolute PEs over time ± SEM. Top) Participants' data shows a decrease in the PEs over time ($\rho$:−0.546, least squares line, red), this indicates participants were learning over time. Bottom) simulated data from the best fitting model (Model 3) shows a similar decrease in PEs over time, indicating that the models learned in a similar way to participants. **d** All three regressors (representing: 1 RW learning, 2 Coarse granularity, 3 Fine

granularity), were significant (one-sided t-test), regressor 1: $t(40)=-5.4617$, $p<0.001$, regressor 2: $t(40)=-5.7377$, $p<0.001$, regressor 3: $t(40)=-7.7059$, $p<0.001$, indicating that participants ($n=41$) learned over time but also made use of both coarse and fine granularity. However, these regressors were correlated and conclusions regarding this GLM should thus be drawn with caution. Participants' parameter estimates (for each regressor) are indicating by the individual data points, which are summarized by the adjacent boxplots of the same colour. The boxplots indicate the median (middle line), and the box is formed by the 25th, and 75th percentile. The whiskers extend to most extreme data points not considered outliers (1.5 times interquartile range), outliers are indicated with + signs. [One-sided t-test; * indicates $p<0.05$, ** indicates $p<0.001$, no correction for multiple comparisons]. CG coarse granularity, FG fine granularity, RP reference point, # number of, PEs prediction errors, SEM standard error of the mean, LSLine least squares line.

---

deemed satisfactory (see Supplementary Figs. 4–6): learning rate ($r\geq0.79$), weighting parameter ($r\geq0.97$) and starting value ($r\geq0.82$). Data were also simulated 200 times for the construction of the confusion matrix—now for every model separately. All models were fitted to this simulated data using BIC as the metric of model fit. Ideally, data simulated with a model will also be best fitted by that model, indicating that models can be distinguished. All five models were recovered correctly in ≥96.5% of the simulations (see Supplementary Fig. 3). Having found evidence that participant employed social knowledge structures for learning about veridical profiles over a wide range of traits, we conducted a second experiment, in which participants performed the same task but were presented with artificially constructed profiles over a narrow range of traits. That is, Experiment 2 entailed more controlled feedback.

### Experiment 2: constructed profiles & narrow traits
In this experiment, we removed the veridical structure of trait self-ratings by artificially constructing feedback for 60 positive trait adjectives belonging to two Big-Five factors, agreeableness and conscientiousness (30 adjectives belonged to each factor) for four

profiles. An average value was determined for each factor based on independent self-ratings from a previous study (Korn et al., 2012). To create feedback for each adjective separately, normally distributed noise was added to the average factor to create 30 distinct trait ratings. It is important to note that through this procedure we kept the reference point component consistent with average trait ratings in the population. We only changed the relationship between traits, i.e. the trait similarity structure. We hypothesized that for this experiment using a fine-grained similarity structure (i.e. single trait similarities) should be less advantageous than employing coarse granularity (e.g., average values per Big-Five factor). Otherwise, the experimental set-up was analogous to the first experiment (i.e. participants were not aware that the profiles were artificial nor that they would only learn about two factors). Participants were not given any clues that would encourage them to use a coarse instead of fine granularity structure during learning.

To asses if participants were learning over time (i.e. showed a decrease of PEs over trials), we calculated the Pearson correlation coefficient to compare the mean absolute PEs with the trial number and found a negative correlation, $r(58)=-0.546$, $p<0.001$ (Fig. 4c,

top). Calculations of a GLM with three regressors, each representing an integral part of our computational models, indicate that participants learned over trials but also seemed to use both coarse and fine granularities during learning (Fig. 4d).

As hypothesized before we indeed found that the winning model for this experiment was Model 3 [Coarse Granularity & Population RP]. Model 3 assumes that participants use a coarse representation of traits for generalizing. It only keeps track of a factor average combined with using an average person as RP. Both fixed- and random-effects analyses showed that Model 3 was the best fitting model (Fig. 4a & Supplementary Fig. 2b). This indicates that—in absence of innate similarity structure within traits—participants use the coarser granularity combined with the population mean as a reference point.

Model simulations with the winning model showed a similar decrease in the absolute PE over time as participants' data $r(58) = -0.489$, $p < 0.001$ (Fig. 3c, bottom). Simulations for the best performing model had a better fit for Model 2 [Coarse Granularity] than for Model 3 (Fig. 4b), which shows that in this experiment participants properly used coarse granularity but could improve their performance by abandoning the use of the reference points altogether.

Tests for model robustness and distinguishability were implemented by performing parameter recovery and calculating a confusion matrix. Parameters for the winning model were recovered satisfactorily (learning rate, $r \geq 0.85$, weighting parameter, $r \geq 0.98$, and starting value, $r \geq 0.75$) (Supplementary Figs. 4–6). The confusion matrix indicated that models were distinguishable with all models being recovered in $\geq 99\%$ from the simulated data (Supplementary Fig. 3). In Experiment 2, we not only manipulated the feedback but also included traits that belonged to two instead of the five Big-Five personality factors originally used in Experiment 1. To rule out the possibility that this narrower item set was responsible for the observed changes in the winning models, we conducted a third experiment in which participants received veridical feedback (like Experiment 1) but with items that belonged to only two factors (like Experiment 2).

### Experiment 3: real profiles & narrow traits

Experiment 3 differed from the previous experiments in two distinct ways. First, as in Experiment 2, participants only rated others on traits belonging to two Big-Five factors (agreeableness & conscientiousness). This was a narrower item set compared to the one used in Experiment 1, in which traits from all five factors were included. Second, in contrast to Experiment 2, self-ratings were sampled from real self-ratings (given by participants from Experiment 2). Similar to the previous experiments, participants rated 60 trait adjectives in total (30 for each of the two factors). As the innate personality structure between traits remained intact, we hypothesized that, similar to Experiment 1, participants would use a fine-grained representation of the traits combined with a reference point of the population mean when rating others traits, i.e., we expected Model 5 to win.

The Pearson correlation between trial number and absolute PEs revealed a negative correlation ($r(58) = -0.42$, $p < 0.001$), indicating that participants learned over the course of a run (Fig. 5c, top). In a further analysis we calculated a GLM with three regressors. These three regressors captured learning over time as well as coarse- & fine-granular learning. We found all three regressors to be significant, indicating that participants learned over time but also made use of both the granularity structures (Fig. 5d). For an in-depth and more stringent look at the knowledge structures used by the participants we used computational models.

Using a reduced number of factors and veridical feedback in this experiment, we replicated our results from Experiment 1. Model comparison showed that the winning model according to our measures of fixed- and random-effects was Model 5 [Fine granularity & Population RP] (Fig. 5a & Supplementary Fig. 2c). This indicates that, in

accordance with our hypothesis, even when learning on a reduced number of factors, participants still used fine-grained similarities between traits combined with the population means as reference points to aid learning.

We performed model simulations to assess whether the models showed learning over time. These simulations, using the winning model, showed a decrease in the absolute PE akin to participants ($r(58) = -0.764$, $p < 0.001$) (Fig. 5c, bottom). Model simulations for the best performing model confirmed that Model 5, which assumes fine-grained representations of traits combined with RPs in the form of population means, was also the best performing model of the five models in the set (Fig. 5b).

We tested for model distinguishability by calculating a confusion matrix that showed that all models could be retrieved with $\geq 98\%$ accuracy (Supplementary Fig. 3). Parameter recovery was also robust with high correlations between employed and recovered parameters (learning rate, $r \geq 0.83$, weighting parameter, $r \geq 0.97$, and starting value, $r \geq 0.78$) (Supplementary Figs. 4–6).

In the first three experiments, we found evidence that participants consistently relied on average population ratings as reference points. Next we wanted to test if participants would deviate from these standard population averages as a reference point when learning about a subpopulation that could be perceived as dissimilar to participants. To achieve this, we varied the group of the profiles from students, which are the same as the participants (i.e., an in-group), to an out-group (i.e., fashion models), who are likely to be perceived as having a different personality from the students.

### Experiment 4: fashion models

Based on our evidence from Experiments 1–3, Experiment 4 was preregistered on the Open Science Framework (https://osf.io/8r6gv) and conducted accordingly. That is, we preregistered the computational modelling approach (the models and their respective analysis) and made no deviations from this preregistration. However, any additional analyses (e.g., GLM and correlations) were added later and were thus not part of the preregistration. In this experiment we tested participants on the self-ratings of an out-group, i.e., fashion models. We chose fashion models as the out-group because we assumed participants to have prior knowledge (or opinions) about them coupled with a high probability that they would not know a fashion model personally. This meant that participants might change their reference point to a stereotypical reference point. Like the other experiments, self-ratings were combined with fabricated personal information into profiles. Participants were tasked with learning four profiles on all five of the Big-Five factors (i.e., the same stimuli as in Experiment 1). In order to test for stereotypical views, we added three computational models to our regular five models. In these additional models, we changed the previous reference points to an average of fashion model ratings that were assessed before the learning task. Our hypothesis was that participants would still use the fine granularity structure but combine it with this new stereotypical reference point instead of using the student reference point.

First, we investigated whether participants held stereotypical views of the fashion models. Before starting the learning task, participants were asked to rate their impression of an average fashion model on all 60 traits that were used in the experiment. From these ratings, we calculated a new reference point that was based on these perceived fashion model self-ratings. To compare the previously used student reference points (M: 6.01, SD: .69) with the perceived fashion model reference points (M: 5.13, SD: .32), we conducted a paired sample $t$-test, which indicated that there was a significant difference between the two reference points, $t(59) = -9.7137$ $p < 0.001$.

As in the other experiment we conducted a Pearson correlation analysis to test for a relation between the trial numbers and absolute PEs. This produced a negative correlation $r(58) = -0.307$, $p < 0.001$,

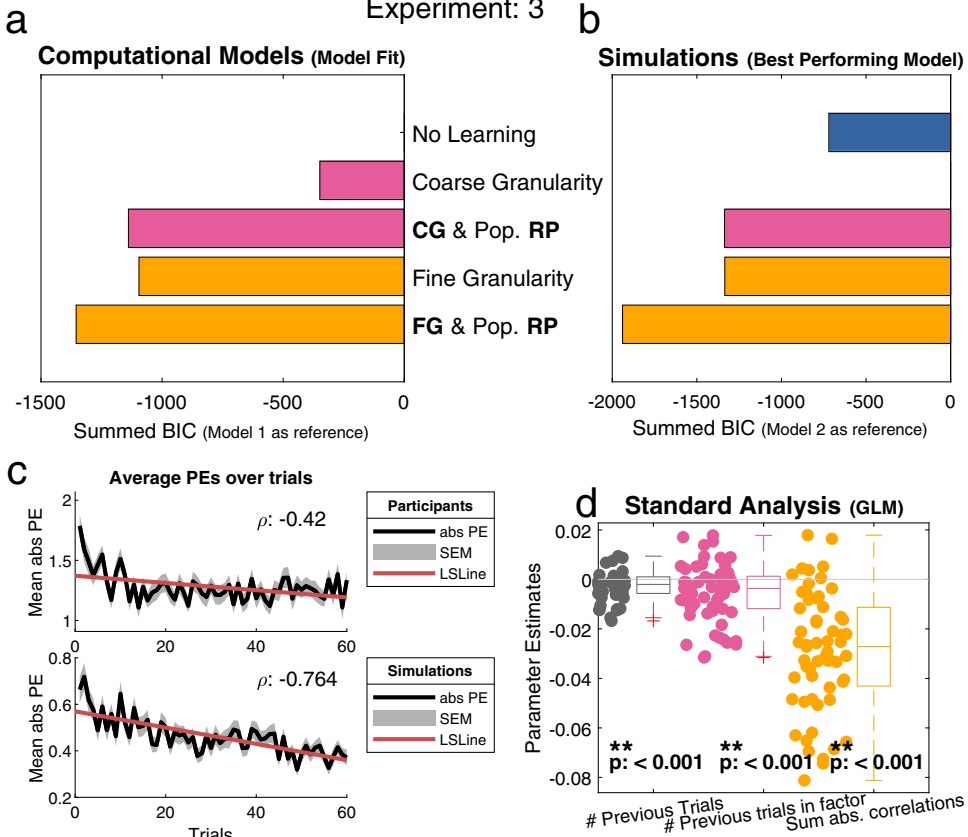

**Fig. 5 | Overview of the main analyses for Experiment 3.** In this experiment participants learned about real profiles on two Big-Five factors. Results indicate participants' use of a fine correlation structure. **a** Similar to experiment 1, Model 5 [Fine Granularity & Population RP] is the best fitting model for experiment 3 ($n = 59$). This model uses the average population as a reference point and fine granularity for generalization. **b** Simulated data to find the best performing model ($n = 59$). In line with results from the participants, model 5 [Fine Granularity & Population RP] was the best performing model, demonstrating that participants used the best possible strategy. **c** Both plots display the average absolute PEs over time ± SEM. Top) Participants' data shows a decrease in the PEs over time (ρ:−.42, least squares line (red)), which indicates that participants learned over time. Bottom) Simulated data from the best fitting model (Model 5) also shows a decrease in PEs over time (ρ:−0.764), showing that the models learned in a similar way to

participants. **d** All three regressors (representing: 1 RW learning, 2 Coarse granularity, 3 Fine granularity), were significant (one-sided $t$-test), regressor 1: $t(58) = −3.414$, $p < 0.001$, regressor 2: $t(58) = −3.6269$, $p < 0.001$, regressor 3: $t(58) = −9.4348$, $p < 0.001$, showing participants ($n = 59$) learned over time but also made use of both coarse and fine granularity. Individual parameter estimates are indicated by the coloured dots, which are summarized by the adjacent boxplots (median (middle line), 25th, and 75th percentile (box), most extreme points not considered outliers (whiskers), outliers (1.5 times interquartile range) indicated with + signs). Due to high correlations between these regressors, conclusions regarding these regressors should be drawn with caution. [One-sided $t$-test; * indicates $p < 0.05$, ** indicates $p < 0.001$, no correction for multiple comparisons]. CG coarse granularity, FG fine granularity, RP reference point, # number of, PEs prediction errors, SEM standard error of the mean, LSLine least squares line.

indicating participants learned over time (Fig. 6c, top). A further analysis that used a GLM to test for the use of RL and both coarse and fine granularities indicated, in-line with previous results, that participants used fine-granular representations (Fig. 6d).

Model comparison confirmed our preregistered hypothesis: both fixed- and random-effects analyses indicated Model 5-STE [Fine granularity & Stereotype RP] as the winning model (Fig. 6a & Supplementary Fig. 2d), suggesting that participants used a fine-grained representation of the personality structure together with stereotypical reference points.

Simulations of the winning model showed a decrease in the absolute PEs, $r(58) = −0.788$, $p < 0.001$, indicating that these models learned in a similar way to participants (Fig. 6c, bottom). Surprisingly, simulations for the best performing model indicated that Model 1 [No Learning] would have been the best strategy for this task (Fig. 6b). After analysing the specific profiles used in Experiments 1 and 4, we concluded that there was a higher correlation between the fashion model profiles and the standard population RP (i.e., the average student ratings) compared to the profiles for experiment 1 (Supplementary Fig. 8). This means that participants did not need all the

information that the more complex models provided and that the simple No Learning model sufficed in capturing the main complexities of each profile. Interestingly, the second best performing model was Model 5 [Fine granularity & Population RP], indicating that the very next best strategy would have used the standard population RP in favor of the stereotypic RP. The fact that participants still used Model 5-STE [Fine granularity & Stereotype RP] even though better strategies were available shows both the pervasiveness of the stereotypic RPs and the use of similarity learning functions as default strategies.

Both the parameter recovery and the confusion matrix were robust. Correlations between simulated and recovered parameters for the winning model were satisfactory (learning rate $r ≥ 0.81$, starting value $r ≥ 0.74$ and weighting $r ≥ 0.94$). The confusion matrix indicated that all models were distinguishable with all models being retrieved with ≥97.5% accuracy (Supplementary Figs. 3–6).

In the first four experiments, we found evidence that participants used both the granularity and the reference point in multiple meaningful ways for experiments with trait words as stimuli. In order to test the scope of these models we next tested them on a different set of stimuli. Specifically, we did not test use trait words

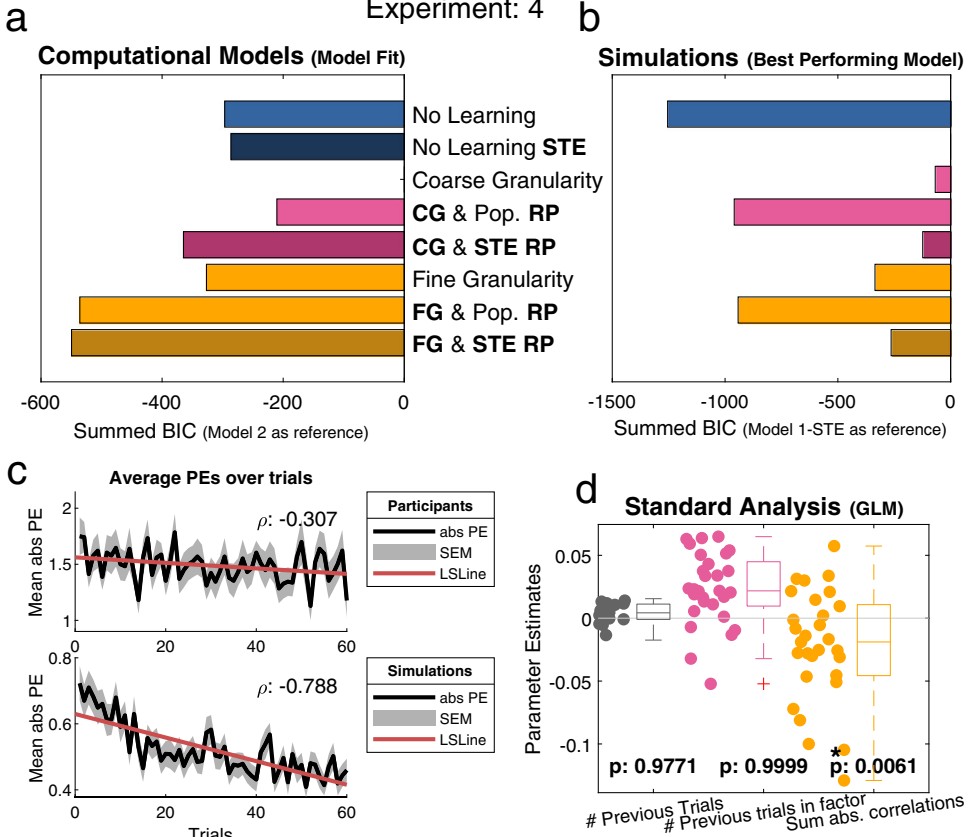

**Fig. 6 | Overview of the main analyses for experiment 4.** In this experiment participants learned about an out-group (i.e., fashion models) instead of the regular in-group (i.e., students). Additionally, three models were added to our original set of five models to capture stereotypic inclinations (STE). These stereotypic models have a darker colour in the figures and are indicated by STE in the model names. **a** As hypothesized, the best fitting model was Model 5-STE [Fine Granularity & Stereotypic RP] ($n$ = 29). This model uses the expected stereotypical self-ratings from models as a reference point and fine granularity for generalization. **b** Simulated data for the best performing model ($n$ = 29). Contrary to participants' data the best performing model was model 1 [No Learning]. This indicates that participants used too complex a strategy for learning about the fashion models (see Supplementary Fig. 8 for details). **c** Both plots display the average absolute PEs over time ± SEM. Top) Participants' data show a decrease in the PEs over time (ρ:−0.307, least squares line (red)), an indication of learning over time. Bottom) Simulated

data from the best fitting model (Model 5-STE) show a large decrease in PEs over time (ρ:−0.788). **d** Of the three regressors (representing: 1 RW learning, 2 Coarse granularity, 3 Fine granularity), only the third regressor was significant (one-sided $t$-test), regressor 1: $t(28)$ = 2.0906, $p$ = 0.9771, regressor 2: $t(28)$ = 4.3546, $p$ = 0.9999, regressor 3: $t(28)$ = −2.6794, $p$ = 0.0061, indicating participants ($n$ = 29) used fine granular representations during learning. Individual data points represent participants' parameter estimates. Boxplots summarize these parameter estimates (median (middle line), 25th, and 75th percentile (box), most extreme points not considered outliers (whiskers), outliers (1.5 times interquartile range) indicated with + signs). Due to the high correlations between regressors one should be careful when drawing conclusions based on these regressors. [One-sided $t$-test; * indicates $p < 0.05$, ** indicates $p < 0.001$, no correction for multiple comparisons]. CG coarse granularity, FG fine granularity, RP reference point, # number of, PEs prediction errors, SEM standard error of the mean, LSLine least squares line.

but rather statements reflecting traits (e.g. I'm the life of the party and I insult people).

## Experiment 5: IPIP items

Our final experiment was preregistered in conjunction with Experiment 4 (https://osf.io/8r6gv) and adhered to the same standards i.e., the modelling approach was preregistered and no deviations were made from this, additional analyses were not part of the preregistration. In this experiment, we assessed the robustness and generalizability of our models by testing them on a different set of stimuli. For this experiment, we tested the same sample of participants as in Experiment 4 immediately after they finished Experiment 4. We changed the learning task from the trait adjectives to the German translation of the 50-item International Personality Item Pool (IPIP). We selected five profiles from a large online sample (Open Source Psychometrics Project; https://openpsychometrics.org/) and used this same sample to calculate the population mean, i.e., the reference point for each trait. We expected that our five models would perform just as well on this experiment since both reference points and granularity can be applied to the personality

structures of the IPIP in the same manner. This also led us to expect that participants would use the same models as they did in the previous experiments (i.e., Model 5). We thus hypothesized that participants would use a fine-grained representation of the item similarity structure and the population mean as a reference point.

A correlation analysis of the absolute PEs found a small negative correlation ($r(48)$ = −0.311, $p < 0.001$) showing that participants learned over time (Fig. 7c, top). Furthermore, a GLM with separate regressors that each represent a core part of our models (i.e., Rescorla–Wagner learning, and the use of both coarse and fine granularity) did not find evidence for their use by participants, potentially as a result of the smaller rating scale (ranging from 1 to 5) and a lesser number of items per factor (Fig. 7d).

Fixed- and random-effects analyses showed that Model 4 [Fine Granularity] performed best (Fig. 7a & Supplementary Fig. 2e). This indicated that participants used a fine-grained representation of the item similarities but did not seem to make use of the reference point.

Simulating the winning model on the same dataset revealed a large negative correlation $r(48)$ = − 868, $p < 0.001$, in line with

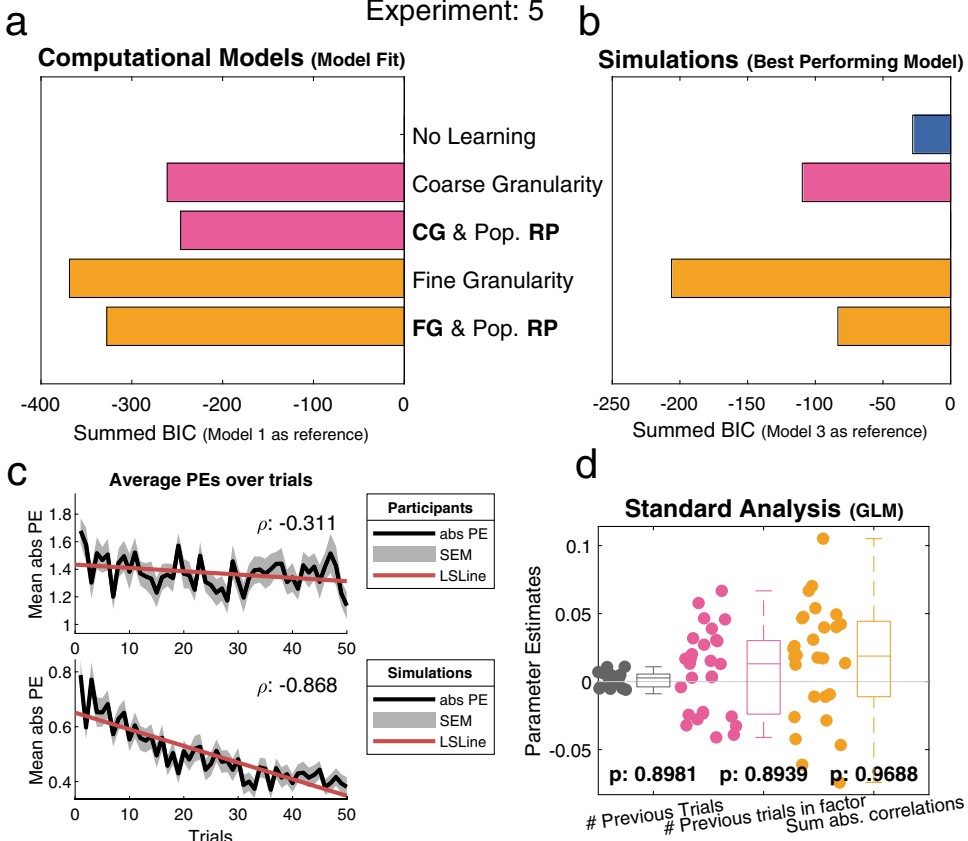

Fig. 7 | **Overview of the main analyses for experiment 5.** In this experiment participants learned about the German translation of the 50-item International Personality Item Pool (IPIP). **a** Model 4 [Fine Granularity] was the best fitting model (*n* = 28). **b** Model simulations (*n* = 28) confirmed that Model 4 was the best performing model and thus the best strategy to use for this specific experiment. Showing that the Population RP did not add enough information to be useful. **c** Both plots display the average absolute PEs over time ± SEM. (Top) Participants' data shows a decrease in the PEs over time (ρ:−0.311, least squares line (red)), indicating participants learned over time. Bottom) Simulated data from the best fitting model (Model 4) shows a large decrease in PEs over time (ρ:−0.868). Replicating participants' learning during the task. **d** A GLM with three regressors (1 RW learning, 2 Coarse granularity, 3 Fine granularity) resulted in no significant regressors (*n* = 28, one-sided *t*-test) regressor 1: *t*(27) = 1.3021, *p* = 0.8981, regressor 2: *t*(27) = 1.2775, *p* = 0.8939, regressor 3: *t*(27) = 1.9446, *p* = 0.9688. Participants' individual parameter estimates are indicated by coloured data points. These parameter estimates are summarized by the boxplots of the same colour. Boxplots indicate the median (middle line), and the box by the 25th, and 75th percentile, the whiskers are the most extreme points that are not considered outliers (1.5 times interquartile range), outliers are indicated with + signs. [One-sided *t*-test; * indicates *p* < 0.05, ** indicates *p* < 0.001, no correction for multiple comparisons]. CG coarse granularity, FG fine granularity, RP reference point, # number of, PEs prediction errors, SEM standard error of the mean, LSLine least squares line.

participants' data (Fig. 7c, bottom). Simulations for the best performing model showed that model 4 (Fig. 7b) was indeed the best strategy to use, demonstrating that the reference point did not add enough information to be useful.

Parameter recovery was robust with satisfactory correlations between simulated and recovered parameters (learning rate *r* ≥ 0.74, weighting *r* ≥ 0.89 and starting value *r* ≥ 0.51). Retrieval of models on their own simulated data for the confusion matrix was achieved with an accuracy ≥92% for all models (Supplementary Figs. 3–6).

Results from this experiment are a first indication that the models can be applied to a broader scope of social learning situations.

### Additional analyses

Additionally, we fitted all models described above with separate learning rate parameters for positive and negative PEs (Supplementary Fig. 9). This allowed us to investigate whether participants learned differently from positive and negative feedback. For all experiments, except for Experiment 4, these models were worse than the standard models. In Experiment 4, Model 12 [Fine Granularity & Stereotypical RP], with separate free parameters for positive and negative PEs, performed best. Fitted parameters for this model indicated participants

learned more from positive PEs. Interestingly, the total number of positive PEs over all participants was higher than the number of negative PEs (2953 vs 1873), indicating participants might have had too negative a perception of the fashion models.

When lacking experience with other groups, a consistent and reliable reference point would be to use one's own self-ratings on traits. For all models that use the population RP we added models that use the participants' self-ratings as RP (Supplementary Fig. 10) and fitted these models on all experiments. For all experiments, the models using self-ratings were worse than those using the population RP, indicating that participants relied on a more accurate population RP than on their own self-ratings.

## Discussion

In the present study, we investigated which potential strategies participants use in social learning about others. We formalized two concepts, which we refer to as reference points and granularity, and constructed a number of models to test their relevance for several social learning tasks over five different experiments. We found evidence for the flexible use of both reference points and granularity across all five experiments using both model-based and model-free analyses.

When participants learned about veridical profiles from real people on a wide range of traits, they made use of reference points that summarised the average person of the other group. On top of that, participants in Experiment 1 used a learning algorithm that combined Rescorla–Wagner (RW) learning and a fine granularity social knowledge structure i.e., they generalized updating via PEs according to the similarities between traits.

To delineate this concept of granularity further, we constructed profiles for Experiment 2. These artificial profiles lacked the fine-grained similarities, with only the coarse-grained structure being intact. Remarkably, this manipulation led to different behaviour. Participants made use of coarser personality structures while maintaining the use of reference points. Not only the profiles changed in Experiment 2 but also the number of factors used. To make sure this did not influence the results, we kept the same number of factors (i.e., two factors) but used real profiles once again for Experiment 3. In this experiment, model fit indicated that participants displayed the same behaviour as they did in Experiment 1. That is, participants used a fine-grained similarity structure and reference points based on the average population during learning. As these past three experiments consistently indicated participants' use of reference points, we wanted to explore next whether participants would change their reference points according to the group the person they are learning about belongs to. In order to achieve this, we presented participants with profiles from an out-group (i.e., fashion models) in Experiment 4. Model comparison revealed the use of fine granularity and a distinct reference point specific to this out-group. This indicates that participants held differing views about the out-group under consideration. These prior assumptions about the out-group influenced the employed reference points—while keeping intact participants' understanding of the underlying similarities between traits. These findings illustrate—in combination with the other experiments—that human trait learning seems to rely on both reference points and granularity. In the final experiment, Experiment 5, we explored the generalizability of these models to new stimuli by changing the task from learning about trait words to learning about personality statements. Model comparison suggested participants still used the fine granularity but no longer the reference point for these stimuli. This could indicate that the reference point might not add enough information for such statements; potentially because participants' representation had become too abstract.

All models displayed high robustness and replicability over all five experiments. Simulations done separately for each experiment indicated that both parameter recovery and confusion matrices were always within a satisfactory range (Supplementary Figs. 3–6). Results were robust across experiments and for all implementations of the computational models[40] and statistical analyses (e.g., regressions & correlations) largely corroborated these results. Finally, best fitting parameters for simulations and participants' data largely overlapped (Supplementary Figs. 11–15). Participants seemed to use both granularity and reference points throughout the experiments, often together in their most complex form (i.e., Model 5), exemplifying the complexity of human social learning. These results, coupled with the subtle differences in learning that are picked-up by the models for the varying experiments, highlight the usefulness of both granularity and reference points in computational models and indicate their importance for guiding social learning. In the following, we discuss these model components.

Across five experiments, we found evidence for the use of PEs in learning, which is in line with a host of previous work on (non-)social learning[12,15,16]. From a statistics perspective we found that participants showed, on average, a decrease of the absolute PEs over time for all experiments, importantly these results were replicated by simulations of the winning models (Figs. 3–7, panel C). These findings reinforce the validity of using the hybrid Rescorla-Wagner learning models, because the PEs are integral to these models.

In the current study we focused on learning about others, in this context we did not expect any difference in positive and negative feedback (i.e., PEs). Nonetheless, we performed an additional analysis that constructed models based on the existing five such that they took into account positive and negative feedback separately. Results from all five experiments largely corroborated our expectations: the regular models were better in every experiment, with a better fit for the positive/ negative models only being achieved in Experiment 4. This might be because the participants learned about an out-group for which they used an incorrect RP with a more negative viewpoint. Due to this negative RP there were simply more positive PEs during learning. In future research, we expect that positive and negative PEs are more influential when learning about oneself, as previous work has shown[41,42]. For example, stereotypical or false beliefs about others, as often seen in patients with borderline personality disorder[43] may result in excessive positive or negative PEs.

We defined reference points as an average person (of a group) and formalized them as the average of a group of independent self-ratings (i.e., individual trait ratings calculated from an independent sample). RPs were used as points of comparison during learning by participants in all experiments except for Experiment 5. We surmise the absence of RPs in Experiment 5 to be related to the smaller scale of the answer options (i.e., 1–5 compared to 1–8 for the other experiments) which decreases their usability as well as the relatively small amount of ten items within a factor.

The most striking use of the RP was exhibited in Experiment 4, where participants were tasked with learning about people from an out-group (i.e., fashion models). During this learning task they used a different (stereotypical) RP than the one they had been using for the other experiments where they learned about people from an in-group (i.e., students). Interestingly, the best performing models indicated participants would be better off using Model 1 [No Learning]. Additional analysis that compared the fashion model profiles and the student profiles from Experiment 1 showed higher correlations among the fashion model profiles and the standard population RP. This indicates that one could potentially estimate the average fashion model accurately with only the population RP available (Supplementary Fig. 8). Notably, participants were persistent in their use of a learning model scaled by fine granular knowledge and the stereotypical reference point. We interpret this finding as a case of stereotypical reference points in line with Jussim et al.[44] who define stereotypes as cognitive categories that people use when thinking about groups and about individuals from those groups. It is widely accepted that stereotypic expectancies can guide learning[45], as exemplified in Experiment 4.

Finally, we introduced a separate set of analogous models with participants' self-ratings as RPs for all experiments (Supplementary Fig. 10). Results from these experiments indicate that the average (perceived student) RPs to always be slightly better, indicating the higher accuracy from such perceived group averages compared to self-ratings.

Granularity was introduced to create computational models that are more akin to real-life learning, which necessitates generalization across similar stimuli and situations. Previous learning models only allowed updating about isolated traits. However, in a real-life situation humans rarely learn about traits in an isolated manner but rather update their whole view of a person's personality (e.g., if you see someone being impolite your view of that person as a whole can change in a negative manner).

Granularity was defined on two levels of detail with which participants could represent the personality space. Participants seemed to use both levels of granularity in our models depending on the specifics of each experiment. We argue that coarse granularity roughly corresponds to the use of schemata[26]. Schemata are representations of features that often co-occur and influence learning and perception[26]. These representations or cognitive structures contain units of

information and their links[46]. Likewise, our implementation of coarse granularity entailed the factors of the Big-Five (i.e., items within a factor occur together most often and are summarized by a single average value). Conversely, updating with fine granularity links all traits together based on their correlations. Furthermore, splitting granularity in two levels of detail was inspired by work of Klein et al.[28] who showed that trait-representations are stored as two distinct memories (episodic and semantic). Episodic memories correspond to single exemplars of social information and semantic representations are more abstract and generalized.

Both coarse- and fine-grained structures described participants' behaviour depending on experimental manipulations. Interestingly, this shows that participants represented and accessed both types of structures when beneficial. Our findings might indicate that humans flexibly change their representations of social knowledge to match implicit task demands. Importantly, participants could only derive clues about how to adapt their strategy from trial-level item and feedback information, as the general task frameworks and instructions were kept very similar across experiments. We believe that this flexible updating of abstract representations also allows for the ability to generalize across widely differing stimuli, as evidenced in Experiment 5.

The granularity of knowledge structures employed in this study were chosen to represent two extreme levels (coarse and fine). In many cases, humans probably do not use either of the extremes but a level of generalization that lies on a continuum between these extremes. Because coarse- and fine-granularity are two levels of granularity, they are inherently related. We attempted to tease their unique contribution apart with a GLM analysis, which, however, was limited due to the high correlations between the three regressors. Results from the confusion matrices of all experiments indicated that models could differentiate between the different levels of granularity. This made us confident that the computational modelling approach was better suited in distinguishing between these two levels of granularity. In future work we aim to investigate the representations at various levels of granularity across this continuum.

Social learning is a very broad domain. Even within a certain social learning domain, there is large situational variability that determines which learning strategy is most suitable. In five consecutive experiments that vary with respect to participant populations, employed items, and to-be learned profiles, we introduced a defined set of computational models that generalize across experiments and pick up the systematic and subtle differences between experiments. The goal of this manuscript was to introduce the modelling framework in a clear and concise manner and not be exhaustive in introducing its potential applications. Nonetheless, we would like to argue for the potential broad scope of their applicability to a majority of social learning settings. As a first step for this broader scope, we included Experiment 5, in which we used a different set of stimuli for which, as expected, the models still functioned as in the former experiments.

We conjecture that these models are especially useful for learning about personality traits, as shown over the five experiments. However, a very similar model space has also successfully been applied to learning about others' preferences[39]. The models integrated similarity structures that can be derived from preference items (coarser grained structures such as candy and fruit and fine-grained item-level correlations). A fine-grained similarity learning model captured how participants learned about others' preferences best.

Moreover, recent studies have found neural evidence that humans encode specific social knowledge structures employed in our models (i.e., fine grained correlation matrices)[47,48] suggesting a central role of item-level similarity structures in social cognition. The relative simplicity of our models makes them adaptable to various multi-dimensional representations across different learning domains. These results are corroborated outside of the social domain by Roweis & Saul[49] who highlight human expertise in representing complex abstract structures akin to the granularity structure introduced here.

In a general modelling framework we argue that our models relate to differences on a continuum between model-free (MF) and model-based (MB) RL[50]. The details captured by granularity reveal complexities that MF-RL does not capture: MF learns a single value for any given stimulus in a given learning situation. Likewise, our models with coarse granularity learn a summary value for each of the Big-Five factors. In their simpler form, we thus think of our models using coarse granularity to be more similar to MF-RL rather than to full MB-RL (i.e., coarse granularity functions as a look-up chart of traits). The more complex models with fine granularity can be viewed as more like MB-RL as they incorporate a full representation of the traits and the similarities between them. This split—akin to differences between MB-RL and MF-RL—suggests that our models can distinguish between more costly optimal models and more efficient heuristic models, similar to models on optimal versus heuristic decision-making[51,52].

Studying social interactions in a well-controlled experimental setting forces limitations on a study (i.e., in the current experiment some limitations were necessary to establish the current modelling paradigm). First, even though we used RW learning models to capture participants' behaviour, the task, in which participants received direct and exact feedback, bears some resemblance to a supervised learning problem. Supervised models have thus far not been used to explain our task and could pose as interesting models for future research.

A related limitation is that our social learning task (including the given estimates and the received feedback) was presented using exact numbers. In many social situations, such estimates and feedback would be given in the form of verbal descriptions (e.g., He is quite generous, and I am very generous). Numerical feedback as used in our task is more common in school or work settings as part of performance evaluations (e.g., grades, scorings, etc.).

Future studies should therefore investigate how people translate numerical into verbal evaluations (and vice versa) in similar social learning tasks. Furthermore, learning about others' personality is sometimes more directly related to actions e.g., approaching someone who seems friendly. Expanding the task to include actions based on feedback about someone's personality (e.g., a cooperative foraging task) could therefore further heighten the external validity.

The total amount of trait items used is on the smaller side, e.g., for Experiment 5 we used 50 items (10 items per factor). Expanding the total number of items could offer a more precise look at learning about personality traits. Specifically, it could offer a more rigorous look at the continuum between the coarse and fine-grained extremes.

Future research should explore how people learn about the reference points and granularity in the first place. Based on the suggestions by Klein et al.[28] that memory moves from a reliance on exemplars to more abstract representations, it would be interesting to explore if the concept of granularity also evolves over time during learning—concurrently exploring if humans use even more levels of granularity between the current two extremes and how these levels get adjusted. Furthermore, if and how humans switch within reference points and granularity within and between tasks are interesting avenues for future research. Reference points could be explored with a host of other groups (e.g., other out-groups, (racist) stereotypes, etc.), especially since learning from others seems to be influenced by whether they belong to an in- or out-group[27,53].

The literature in social neurosciences points towards parts of the (medial) prefrontal cortex as one of the most important regions for social cognition[11,54,55]. It would be interesting to explore if and how the concepts that we introduce in these experiments are represented in the brain and if the changes mentioned above can be tracked as dynamic processes in the human prefrontal cortex[56].

As mentioned previously, we expect that the learning models presented here will also be applicable to different domains of learning

such as other domains of social learning (e.g., learning for others[8])— and potentially to different types of non-social, abstract learning.

In summary, we tested how humans learn about the multi-dimensional personality of others. We specifically focused on how social knowledge structures shape this learning and introduced two social knowledge structures that can be mathematically specified in computational models. We found evidence that participants use prior convictions akin to schemata to learn about others who belong to different groups. Crucially, participants displayed a representation of the multidimensional similarity structure of personality traits and were able to apply this on several levels of abstraction (within constraints). The computational models introduced in this study are robust, simple, and widely applicable to multidimensional (social) learning situations.

## Methods

### General
We present five experiments. All experiments are similar to each other and vary on a couple factors per experiment in order to test specific components within our computational models. In particular, these variations aim at distinguishing: Reference Points, Granularity and overall robustness of the models.

### Participants
All five experiments in this study were conducted in accord with the Declaration of Helsinki and approved by the local research ethics committee (Ethik–Kommission der Ärztekammer Hamburg, Number: PV5746). All participants gave written informed consent using a form approved by the ethics committee and were compensated the regular hourly fee (€9,-) for behavioural studies. On average, one experiment took around 30 min. Participants for all experiments were recruited through online advertisement. Experiments 1–3 were administered to independent participant samples. Experiments 4 & 5 were tested on the same sample of participants in the same testing session. Participants' details per experiment are summarised in Table 2. Participants had to meet the following inclusion criteria: (1) age between 18 and 40 years, (2) German native speakers, (3) normal or corrected to normal vision, and (4) no history of neurological or psychiatric disorders. In Experiments 1–3, participants could only partake in one of the experiments; participation in one experiment thus directly excluded them from partaking in any of the four other experiments.

### Preregistration
Experiments 4 & 5 were preregistered on the Open Science Framework prior to data collection (https://osf.io/8r6gv). In this preregistration, we specified our expected sample size, exclusion criteria, measured variables and computational models. Most of these considerations also applied to the first three experiments, which we did not preregister. We followed the analyses specified in the preregistration without deviations. However, only the computational modelling approach was preregistered and any additional analyses were therefore not preregistered.

### Task
All five experiments shared the same structure, but differed in their content (i.e., words and profiles). The first experiment is described below. Any differences between Experiment 1 and the following experiments are detailed in the section differences between experiments. The main task in the experiments was a social learning task followed by (self-) rating tasks.

In experiment 1 we tested 35 participants (23 females, mean age 24.37, SD 3.66). Participants performed a social learning task about different trait profiles, where a profile consisted of sparse information about a person (name, age, time since starting their university studies and time spent living by themselves) and self-ratings on a selection of traits selected to resemble items found in the Big-Five (Supplementary Tables 1–3). The information and self-ratings from these profiles were selected from an independent previous study (Korn et al., 2012), with fictitious and randomly assigned common names. Instructions for the whole experiment were given orally and presented on the screen at the start of the experiment. Participants were given the opportunity to ask questions after the instructions and could start the experiment themselves through a button press. The learning task consisted of four runs, i.e., one run per profile. Participants learned about a different profile in each run (Fig. 1a). As described above, the person of the profile in question was briefly introduced at the start of the run. Two of the profiles were given female names and two of the profiles were given male names. Each experimental trial started with the presentation of a trait word (e.g., generous, arrogant), a blank space underneath, and the rating scale (1 = does not apply at all; 8 = does apply very much). Participants had four seconds to estimate how they thought the persons in question rated themselves on this trait by choosing the appropriate number [1–8] on the keyboard. Directly after this estimate participants received feedback (i.e., self-ratings from the person of the profile) for a duration of two seconds, for a total of 60 trials. This means that participants could learn about each person's self-ratings over time.

After the social learning task, participants were asked to give their self-ratings on 80 traits, i.e., the full list of 40 positive and 40 negative words used in previous studies (Korn et al., 2012; 2014). A subsection of 30 positive and 30 negative traits was used in the learning part of the experiment. All experiments were presented using the MATLAB toolbox Cogent.

### Differences between experiments
Tables 1 & 2 summarise the differences between all five experiments. The first experiment was used as a reference to subsequent experiments. In Experiments 2–5, we manipulated key features of the first experiment and tested whether these changes resulted in different fits of the computational models. Supplementary Tables 1–3 list the stimuli used.

**Experiment 2: Constructed Profiles & Narrow Traits.** The sample in experiment 2 consisted of 42 participants (28 females, mean age 23.67, SD 3.19). Experiment 2 consisted of artificial profiles of a narrower trait space. That is, 60 trait words were selected pertaining to the two Big-

**Table 2 | Overview of the participants for the five experiments**

| Experiment | n Tested | Age (mean) | Age (SD) | n Female | n Final | n Profiles removed | n Sessions removed |
|---|---|---|---|---|---|---|---|
| 1 (Original) | 35 | 24.37 | 3.66 | 23 | 35 | 0 | 1 |
| 2 (Constructed Profiles) | 42 | 23.67 | 3.19 | 28 | 42 | 0 | 0 |
| 3 (Two Factors) | 59 | 25.37 | 4.95 | 30 | 59 | 0 | 2 |
| 4 (Fashion Models) | 30 | 25.38 | 4.91 | 15 | 29 | 1 | 0 |
| 5 (IPIP items) | 30 | 25.38 | 4.91 | 15 | 28 | 2 | 8 |

The same group of people participated in Experiments 4 & 5.

Five factors agreeableness and conscientiousness (with 30 positive traits for each of the two factors). Specifically, 15 trait words were selected from the original list of 80 traits from the previous studies. A set of 45 trait words were newly introduced, which resulted in a total of 125 trait words used overall. In Experiments 2–4, participants made self-ratings on all of these 125 words.

The self-ratings of the profiles about which participants in Experiment 2 learned were simulated. Specifically, we specified a mean value for each of the two factors, which could be 5 or 7 on the 8-point scale. We specified four profiles with the four possible combinations of a mean of 5 or 7 for the two factors (i.e., 5 & 5, 5 & 7, 7 & 5, and 7 & 7). For each trait word, we individually added random variability (SD = 1) using the standard random normal distribution (randn) in MATLAB. After adding this random variation, ratings were rounded to their nearest integer and capped to the anchor points [1–8]. All four profiles were given names that matched the gender of the participant.

**Experiment 3: real profiles & narrow traits**. Experiment 3 had a sample of 59 participants (30 females, mean age 25.37, SD 4.95). Profiles in Experiment 3 were veridical self-ratings belonging to two of the Big-Five factors (agreeableness & conscientiousness). We selected these profiles from self-ratings given by participants of Experiment 2.

**Experiment 4: fashion models**. For experiment 4 we tested 30 participants, of which 1 was excluded for missing too many trials. This left a final sample of 29 participants (15 females, mean age 25.38, SD 4.91). Profiles in Experiment 4 were selected from a pre-defined reference group. That is, from a sample of 30 female fashion-models, who provided online self-ratings in a separate study conducted for a Bachelor's thesis (unpublished), from these 30 self-ratings we selected four profiles for the learning task. Information given to participants in the learning task was: female first name (fictionalized), age, years of work as a model, and time spent living by themselves.

At the start of the experiment, we asked participants for pre-ratings to capture their view of the average fashion model's personality on the used stimuli. After giving these pre-ratings, participants completed the standard learning task. They rated four fashion model profiles and received trial-by-trial feedback about the fashion models' own ratings. Both the pre-rating and learning task were performed on the same 60 traits and five factors as in Experiment 1.

**Experiment 5: IPIP items**. The same group of participants as experiment 4 were tested, however two participants were excluded for missing too many trials. This left a final sample of 28 participants (15 females, mean age 25.38, SD 4.91). A different set of items was used for Experiment 5 (Supplementary Table 3). Instead of the trait words used in Experiments 1–4, we used 50 items from the German translation of the International Personality Item Pool (IPIP), also known as Lexical Big-Five factor markers[22]. These items were rated on a 5-point scale as those in the original questionnaire (1 = does not apply at all; 5 = does apply very much).

Information for selection of the profiles was taken from a large online dataset with 50 self-report items on the IPIP from over 1 million participants (Open Source Psychometrics Project; https://openpsychometrics.org/). For our learning task, we selected five profiles with average ratings on 4 out of the 5 factors (mean score within 1 SD) and a divergent score on the remaining factor (mean score above 1 SD). That is, each profile is divergent on one out of five factors. After the learning task, participants judged themselves on the same 50 items.

## Data analysis
**Missing data**. The same exclusion criteria for missing data were applied for all experiments. Within an experiment, a single run would be removed if more than 20% of the answers were missing (due to

participants not answering within four seconds). Data from a participant was completely removed if more than 10% of all answers were missing. This resulted in the exclusion of three complete data sets (two data sets in Experiment 5 and 1 data set from Experiment 4) and nine separate profiles over all experiments combined (see Table 2 for details). These exclusion criteria are in line with our preregistration for Experiments 4 & 5.

**Statistical behavioural analysis**. To test whether the PEs have a downward trend over time (an indication of learning), we calculated the average of the absolute PEs per profile per trial for all participants. Profile data were then averaged into one vector (with length being the number of trials) of the average absolute PEs over time per experiment. To determine whether there was a significant decrease of PEs over time, we calculated the Pearson correlation coefficient on these absolute average PEs and the corresponding trial number. A negative Pearson correlation indicates a decrease in the absolute PEs over the trials. The same procedure was applied to data that was simulated based on the winning (i.e., best fitting) model for that experiment.

Moreover, for each experiment separately, we calculated a GLM that explained the accuracy per trial (i.e., the prediction error) with three regressors. Each regressor was initially tested in a separate regression and captured an integral part of our computational models. In brief, the regressors were the following:

Regressor 1: total number of previous trials seen for each item, (this assesses the relationship between the decrease of the PEs and the number of items seen previously and thus represents learning in the standard Rescorla-Wagner model)

Regressor 2: total number of previous trials that are from the same factor as the current item (this asses the relationship between the decrease of the PEs and the number of items encountered from a specific factor and thus investigates the behaviour captured by the coarse granularity models)

Regressor 3: the summed absolute correlations of the previous items with the current item (this assesses the fine granularity models where the information content of all the previous items is weighted by their correlation to the current item).

In a second-level analysis, participants' individual parameter estimates were subjected to a one-sided one-sample t-test to test if the slope was significantly different from zero in the negative direction (indicating a decrease of the absolute PE over time).

## Computational models
We created computational models that use standard Rescorla-Wagner learning and expand these models with two sources of information: Granularity and Reference Point. Through our computational models, we explore to what degree participants use these concepts (Fig. 2a, Supplementary Fig. 1). We test participant learning on individual profiles but do assume that participants use the same strategy (i.e., model) throughout the experiment.

**Calculation of the PEs**. Models 2–5 make use of the standard Rescorla-Wagner learning model, this model makes use of the prediction error (PE) to update the prediction (P) for the following trial. The prediction error for all models is the feedback (F) on a certain trial (t) minus the prediction (P) on that trial.

$$PE_t = F_t - P_t \tag{1}$$

**Granularity**. Granularity refers to the level of detail with which participants represent the others' traits. This means that one can either have a summary value per (Big-5) factor, or learn a separate value for each trait per person (Fig. 2a, and Supplementary Fig. 1d, e). The former, Coarse Granularity, is mathematically defined by having a single summary value for each factor which will be updated only when one

learns about a trait that belongs to that factor. The latter, Fine Granularity, assumes one learns a separate value per trait but also updates all other traits based on how similar (i.e., correlated) they are to this current trait. In experiments 1–4, these similarity matrices were calculated using Pearson's correlation on self-ratings from two published independent laboratory studies[41,57] as well as data from previous online studies[58]. Specifically, we selected four participants from the 27 participants in Korn et al. (2012)[57] for the four profiles in Experiment 1. The self-ratings of the remaining 23 participants as well as the self-ratings from the 78 healthy control participants in Korn et al. (2014)[41] were used for the similarity matrix. These 101 participants who completed laboratory studies gave ratings on the 80 traits of the original list. Additionally, for the calculation of the similarity matrix we included online ratings of 734 participants, who each gave self-ratings on 50 pseudorandom traits from the overall list of 125 traits. For Experiment 5, we calculated the correlation matrix from the (>1 million) sample from the Open Source Psychometrics Project (https://openpsychometrics.org/).

**Reference points.** Reference points refer to a-priori expectation of a person's personality. This means that a person has an idea of the trait ratings from an average person who belongs to the same group as the person who is being judged and uses these ratings next to regular learning (Supplementary Fig. 1b, c) (e.g., most people from this specific group are generous I therefore should rate them higher on generous, but I perceive them, on average, to have less diligence, so I should score them lower on this trait). The reference point data was calculated by taking the mean value per trait of the same datasets that were used to calculate the similarity matrices. In Experiment 4, we used the stereotypical ratings given by participants before the learning task. To test whether there was a difference between the self-ratings of the independent study and the stereotypical ratings we calculated an independent samples t-test on the trait averages.

**Model 1: No learning.** Model 1 assumes that participants perform a linear transformation of the reference point (RP) population mean to predict (P) others' personalities. This model performs like a standard linear regression where $b_0$ represents the intercept and $b_1$ the slope.

$$P = \text{RP}*b_1 + b_0 \tag{2}$$

**Model 2: coarse granularity.** Updates an average factor value (F, factor) for the next trial estimate based on the prediction error (PE) and the current estimate. That is, on a trial-by-trial basis the model updates a value that represents the average value for this factor. The speed of learning is determined by the free parameter α (bounded [0 1]), with a second free parameter; starting value (bounded [1 8]), determining the value at which each factor will be initialized. That is, the starting value determines the first guess the model makes when a new factor or profile is presented (i.e., the value at $P_{(0,F)}$).

$$P_{(t+1,F)} = P_{(t,F)} + \alpha*\text{PE} \tag{3}$$

**Model 3: coarse granularity & population RP.** Model 3 expands Model 2 by adding the reference point (RP) in form of a population mean per trait. During each trial model 3 learns an average factor value like model 2 but it adds information based on the Reference Points (i.e., the average population rating on this trait). Information is integrated using the weighting parameter γ (bounded [0 1]). This parameter determines how much participants rely on just the RW from model 2 or the RP (0.5 indicating that both are used equally). Like Model 2, Model 3 also uses the parameters learning rate (α) and starting value for its initial estimate for a new factor or profile.

Parameters α, γ, and the starting value are free parameters.

$$P_{(t+1,F)} = \gamma*\text{RP} + (1 - \gamma)*(P_{(t,F)} + \alpha*\text{PE}) \tag{4}$$

**Model 4: fine granularity.** Model 4 employs fine-grained granularity and updates all items (All) at once based on how correlated they are to the current item (Supplementary Fig. 1e). This similarity matrix (SIM) was calculated before the experiment based on separate samples from previous studies[41,57]. Thus on a trial-by-trial basis Model 4 updates the current trait based on the PE and the learning rate (i.e., it is perfectly correlated with itself) all the other traits will get updated based on the PE, the learning rate and their similarity to the current item (i.e., their correlation), which means that traits that are more similar get updated more. Model 4 also makes use of the free parameter starting value. Because this model updates all items for every step of learning the starting value is not just initialized for the first value but rather all items. This ensures that the model can updates all items from the onset of learning.

$$P_{(t+1,\text{All})} = P_{(t,\text{All})} + \alpha*\text{PE}*\text{SIM} \tag{5}$$

**Model 5: fine granularity & population RP.** Like the coarse granularity models, Model 5 expands the previous model (i.e., Model 4) by adding the reference point (RP) in the form of the population mean. That is, for every trial learning happens (as in model 4) but additional information in the form of an average trait rating is added. The degree to which this information (i.e., RP and Granularity) is integrated is determined by γ. Similar to Model 4, Model 5 uses the starting value to initialize all item estimates at the onset of learning. Free parameters are α, γ, and starting value.

$$P_{(t+1,\text{All})} = \gamma*\text{RP} + (1 - \gamma)*(P_{(t,\text{All})} + \alpha*\text{PE}*\text{SIM}) \tag{6}$$

**Model additions: stereotypical reference points.** In the preregistration for Experiment 4, we added a variation of the reference point to the models specified above. Next to the standard reference points, we also tested models using the stereotypic view held by the participant prior to learning. This stereotypic view in the reference point (STE) is added for Models 1, 3 & 5.

$$\text{Model 1} - \text{STE } P = \text{STE}*b_1 + b_0 \tag{7}$$

$$\text{Model3} - \text{STE } P_{(t+1,F)} = \gamma*\text{STE} + (1 - \gamma)*(P_{(t,F)} + \alpha*\text{PE}) \tag{8}$$

$$\text{Model5} - \text{STE } P_{(t+1,\text{All})} = \gamma*\text{STE} + (1 - \gamma)*(P_{(t,\text{All})} + \alpha*\text{PE}*\text{SIM}) \tag{9}$$

**Model additions: positive/negative learning rate.** To explore if participants learned differently from positive and negative feedback, we constructed additional models that use two parameters for the learning rate. These two parameters thus have a separate learning rate for positive and negative PEs. All models except for model 1 [No Learning] were adapted in this manner (i.e., Models 2–5 all make use of the following two α's that depend on the sign of the PE in the current trial).

$$a = \alpha^- \text{ if } PE < 0$$

$$a = \alpha^+ \text{ if } PE \geq 0$$

**Model additions: self-ratings as reference points.** Similar to the stereotypical models in Experiment 4 we included additional models that used participants' self-ratings as RPs. To achieve this the models that originally used the population RP (i.e., Model 1, Model 3, and Model 5) were duplicated and adapted to use each participants' self-

ratings, resulting in 11 total models for Experiment 4 and 8 total models for the other Experiments.

$$\text{Model 1} - \text{Self } P = \text{Self}*b_1 + b_0 \tag{10}$$

$$\text{Model 3} - \text{Self } P_{(t+1,F)} = \gamma*\text{Self} + (1-\gamma)*(P_{(t,F)} + \alpha*\text{PE}) \tag{11}$$

$$\text{Model 5} - \text{Self } P_{(t+1,All)} = \gamma*\text{Self} + (1-\gamma)*(P_{(t,All)} + \alpha*\text{PE}*\text{SIM}) \tag{12}$$

**Model fitting and comparison.** For model fitting, the free parameters are all initialized at the average between the maximum and minimum bounds (i.e., 0.5 for both α and γ). The first prediction ($P_0$) was fitted as the free parameter: starting value.

Optimal model parameters were determined using linear least squares estimation. In detail, optimization used a non-linear Nelder-Mead simplex search algorithm (implemented in the MATLAB function *fminsearch*) to minimize the sum of squared errors (SSE) of prediction over all trials for each participant. Model evidence was then assessed using the Bayesian information criterion (BIC) using the formula:

$$\text{BIC} = n*\ln\left(\frac{\text{SSE}}{n}\right) + k*\ln(n) \tag{13}$$

Here, $n$ is the number of trials and $k$ is the number of free parameters in the model (BIC thus penalizes for an increase in model complexity). Models were compared using both fixed- and random-effects analyses. For fixed-effects analyses we calculated the log-group Bayes factor (BF): all model BIC scores are summed across participants and then subtracted from the value of the reference model (worst scoring model) where the best performing model has the lowest score. Furthermore, for random-effects analyses, we calculated the posterior exceedance probability. This measures the likelihood that any model is more frequent in the comparison set than all other models, corrected for the possibility that observed differences are due to chance. This random-effects family wise comparison was performed using the Bayesian Model Selection (BMS) procedure implemented in the MATLAB toolbox SPM12 (http://www.fil.ion.ucl.ac.uk/spm/; *spm_BMS*, ref. 59).

**Best performing models (in the set).** Normally, models are fit on the answers participants give during the task, this helps us estimate which of the strategies (i.e., models) participants used during learning. But in order to understand the usefulness of the strategies we estimate the best performing model (in the set). This is achieved by fitting the models on the profile answers (i.e., the same way the participants experienced the experiment) instead of the participant answers (see Supplementary Fig. 7 for a more detailed description). Comparing these best performing models with the ones that best fit the participants' strategies we can determine whether participants' strategies were well fit to the task demands.

**Confusion matrix.** To test whether data simulated with a model would also be best fit by that model, we created confusion matrices for every experiment separately. For every confusion matrix we drew random parameter values between [0.2 0.8] 200 times. These values were used to simulate data for the profiles using every model. After the data was simulated, noise was added from a standard normal distribution. Then all models were fitted to the simulated data to find out the extent to which simulated data from a particular model is best fitted by this model opposed to all other models (Supplementary Fig. 3). Ideal recovery is achieved when the confusion matrix results in the identity matrix.

**Parameter recovery.** To test whether parameters could be recovered consistently, we simulated data for all five experiments. For every experiment separately we simulated data for the profiles by drawing random parameter values (between [0 1]) from a random distribution 200 times. With these parameters data was simulated using every model. Then noise (sampled from a normal distribution) was added. After this, all models were fitted to this data and parameters were recovered. We measured fit through correlations; where a higher correlation means better fit. Winning models were marked using green squares in Supplementary Figs. 4–6. For Experiment 4, we omitted the stereotype models as these are functionally the same as the standard models when it comes to simulating data.

### Reporting summary
Further information on research design is available in the Nature Research Reporting Summary linked to this article.

## Data availability
The behavioural data that support the findings of this study are publicly available on Github (https://github.com/dnhi-lab/PerLe) and on Zenodo (https://doi.org/10.5281/zenodo.4697286).

## Code availability
The code used for analysis is publicly available on Github (https://github.com/dnhi-lab/PerLe) and on Zenodo (https://doi.org/10.5281/zenodo.4697286).

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

## Acknowledgements
We thank Antonia Wesseloh, Kevin Rozario, Cristian Ioan, and Alina Dinu for help with data acquisition and Jan Gläscher, Tessa Rusch, Lisa Doppelhofer, and Benjamin Kuper-Smith for helpful discussions. KMMF and CWK are supported by the German Research Foundation (DFG); specifically by an Emmy Noether Research Group [392443797]. GR is supported by the National Institute of Mental Health (NIMH, R01MH116252) and the Simons Foundation for Autism Research Bridge to Independence Award. The funders had no role in the study design, data collection and analysis, decision to publish, or preparation of the manuscript.

## Author contributions
K.M.M.F., G.R. and C.W.K. designed the experiments, developed the analysis procedures, and wrote the paper. K.M.M.F. and C.W.K. collected and analysed the data.

## Funding

## Competing interests
The authors declare no competing interests.
