## [Peer Review File · Nature Communications]

Incorporating social knowledge structures into computational modelsReviewers' comments:

Reviewer #1 (Remarks to the Author):

Frolichs et al, examine how people learn about the personality of others. They take a computational approach to examine the impacts of reference points (ie. the biasing of personality predictions toward a group mean) and trait generalization (the degree to which coarse or fine information about trait covariance is used to automatically transfer learning across trait dimensions). In five studies they show that people typically combine reference points with learning that employs fine-grained knowledge of trait covariance for generalization, however that when such fine-grained knowledge lacks predictive validity, they rely on more coarse information (ie. sharing info across big 5 traits).

I like the premise of this paper and think that it brings up interesting points related to real life learning that are often glossed over in psychology tasks or computational models. The findings regarding generalization across traits are novel, as far as I know, and in general the claims are supported by the data, albeit perhaps with more indirect analyses than would be ideal. That said, I have a number of concerns with the current manuscript, which I have detailed below.

I think the key points require additional behavioral analyses for support – and the current analyses do not scratch the surface of the rich behavioral data collected. Without descriptive analysis of behavior, it is tough to know exactly how well the models are fitting – or what elements of behavioral data might be missing in the model. In my mind, the descriptive analyses should be fairly straight forward – for example, the authors could construct a GLM that examined accuracy as a function of the number of 1) previous trials, 2) previous within factor trials, and 3) the sum of absolute correlations with previously observed traits. This would allow the authors to test their key points while making fewer assumptions – also provide summary statistics that can be tested from simulated model data in order or to validate their sufficiency.

The use of the word optimal in the manuscript feels misleading... I would say in most cases the authors seem to use it as synonymous with beneficial. I was also unclear on what fitting to the feedback actually means. Do the authors mean they chose parameter values that minimized prediction errors? I think it would be useful for the authors to actually characterize what optimal behavior would look like in their task – presumably this could be done by assuming that “traits” are generated from a multivariate distribution with a mean (reference point) and covariance matched to the ground truth data – and inverting this process to do inference. But in any case, they should be more clear about what they mean when they say optimal, as I have strong doubts that any of the models considered here are optimal in either the colloquial or technical sense.

The models could be described more clearly. My understanding is that the fine and coarse granularity models are just using different basis functions for generalizing feedback across traits. A typical basis-function approximation would learn weights for each basis function, and make predictions by combining relevant bases. However, the existing equations make it look like only a single element of the P matrix is updated on each trial. It would be useful if the authors could clarify exactly what the model is updating, and if it is indeed only updating a single prediction, how the model affords generalization.

I don't quite understand the point of figure 4, moreover, I think I would have been much more convinced that the modeling results were robust had the authors done some more nuanced behavioral analysis, and then performed posterior predictive checking to show that the best models were able to capture summary statistics from the data – whereas the worse performing models were not. Not only should synthetic data match learning curves (eg. Figure 2), but they should also be able to recover summary statistics from the descriptive analyses suggested above. It would also be useful if the actual data could be plotted as a function of model predicted response probability for conditions of interest.

It would be useful for the correlation structure to be depicted for experiment 2 – also I was unclear on what correlations were used by the fine-grained model in this experiment. The actual correlation structure for the task, or the inappropriate one more attuned to “real data”?

I was a bit unclear where the reference points come from. Are they taken from the data itself? Is the idea that participants have estimates of these reference points from real world experience? Or from task experience?

Perhaps it is my lack of familiarity with OSF (I’ve never used it), but I am unable to see any details in the link to the preregistration for experiments 4&5. I see that there is a site, but I was unable to verify experimental details/predictions.

The presentation in this manuscript, which includes both writing and figures, could be improved.

Line 446: I’m unclear what exactly is going into this ttest – shouldn’t each trait have its own reference point? How is this hypothesis test yielding a single t-stat.

Reviewer #2 (Remarks to the Author):

Review: “ How do humans learn about other people? Incorporating social knowledge structures into reinforcement learning”

In this manuscript, the authors investigate how people learn about others’ personality across multiple experiments. They find, generally, that a learning model, which considers both personality of the “average person” and the correlation structure of personality traits best explains the data (predictions of others personality).

My opinion of this paper is conflicted. On the one hand, there are several interesting, cool, and novel ideas (especially using the correlation between traits to inform the model), and the methods are rather rigorous. On the other hand, I find the framing of the paper overstated, and am bothered by many seemingly arbitrary modeling choices, and the very specific nature of the experimental paradigm. As it stands, the manuscript seems to describe the rather modest beginnings of a very interesting research program. I wish I could be more positive about the manuscript per se however. I outline my concerns below.

Conceptual. The title of the paper highlights reinforcement learning, but this is a misnomer. The task the authors use is a classic supervised learning setting. Participants provide a prediction and get feedback on the same scale. The prediction error is the difference between prediction and feedback (e.g., Prediction = 3, Feedback = 5, PE = 5-3 = 2). In other words, the task is basically (or exactly) a regression problem. Participants learn, but it’s misleading to call this reinforcement learning. The framing should be changed to acknowledge this.

Limitations of the experimental paradigm. The paradigm the authors use is clear, but also very limited in scope (explicit learning of other people’s personality based on their explicit feedback). Given the RL framing, I anticipated variations of the paradigm where participants actually had to use what they learned for something (payoff relevant). As it stands, the set of studies do not provide the general account of social learning that the introduction and discussion imply. For example, the authors state “Social learning is a very broad domain where even fairly standard social interactions can be different every time. We aim to introduce models that can be applied to a majority of these social learning tasks”, and “we expect that the learning models presented here will also be applicable to different domains of learning such as other domains of social learning “. This is not explained in more detail, and I have a hard time seeing how a model developed for such a specific application could have so

general implications. I wish the authors had pushed this further and tested the logic of their model in other paradigms as well.

Study 2 seems odd to me. The authors created low granularity personality traits and found that a low granularity model fit the data best. Could it logically be any other way?

Analysis. As I mentioned, I do like the correlation matrix approach used for learning about others' "personality". This approach seems to have potential. Furthermore, the modeling analysis is well done and rigorous (e.g., recovery, simulations). However, I am left puzzled by some aspects.

1. The Reference Point is a key aspect of the analysis. However, very limited attention is given to the reference point. As I understand it, the authors simply plug in a fixed population average reference point into the model for all participants (except the stereotype experiment). This seems coarse. The authors note in the discussion that future research should evaluate how the RP is acquired/changed. Why is that not done in the present study? One could imagine that the RP changes as a function of experience, and in other words, is different later in the experiment than at the beginning (where it is more likely to reflect some population prior). Moreover, the Profiles are both female and male. Surely, people could apply a different RP to women and men.

2. I understand the model description as that the model is not learning about the different Profiles, but only about personality traits (pooled across Profiles). This seems very odd -surely learning is about individuals?

Minor.

Was learning worse in Exp.4 than in the other experiments? The PE-trial correlation coefficient seems lower.

The authors report that Model 1 (no learning) was theoretically best in Exp.4. This is not interpreted or explained. Why would this be the case? Why wasn't this evaluated a priori?

Reviewer #3 (Remarks to the Author):

Review for "How do humans learn about other people?" by Frolichs et al. The authors tested how humans learn about the multi-dimensional personality of others. They found that participants use prior ideas that they computational formalize into two ideas: granularity and reference point. To do so, the authors analysed 5 experiments, where some experiment conditions were changed to test the robustness of the results. The paper addresses an original, clever and timely question and is well written. They also compare the best fitting model to optimal behaviour (which is important and not frequently done in the field).

Overall the paper is very solid methodologically (several experiments, including pre-registration). The paper's claims rely heavily on computational modeling and parameters analysis. I was very happy to see that these results were perfectly backed-up by model and parameter recovery. I am quite positive about this study. I only have few suggestions.

1/ I am sorry I will not give a more precise suggestion, but I feel that the behavioral results could be illustrated more in detail. In the current version of the manuscript the authors essentially only show the median abs PE across trial . I wonder whether there is more in the data to show.

2/ On a similar note, it would also actually be informative to show the model simulations of the relevant variables. What is the feature that is not explained by the "losing" models? The simulations will nicely complement the BIC graphs (that are not really the more informative to understand the models' behavior).

3/Finally, I believe that some citations are missing. First, the citations concerning observational learning should be updated in the light of recent work (Najar et al. Plos Biology, 2020). Reference point dependence is new in social learning, but not in "private" reinforcement learning (see Palmintieri et al. Nature Comms, 2015)

Reviewer #1 (Remarks to the Author):

Frolichs et al, examine how people learn about the personality of others. They take a computational approach to examine the impacts of reference points (i.e. the biasing of personality predictions toward a group mean) and trait generalization (the degree to which coarse or fine information about trait covariance is used to automatically transfer learning across trait dimensions). In five studies they show that people typically combine reference points with learning that employs fine-grained knowledge of trait covariance for generalization, however that when such fine-grained knowledge lacks predictive validity, they rely on more coarse information (i.e. sharing info across big 5 traits).

I like the premise of this paper and think that it brings up interesting points related to real life learning that are often glossed over in psychology tasks or computational models. The findings regarding generalization across traits are novel, as far as I know, and in general the claims are supported by the data, albeit perhaps with more indirect analyses than would be ideal. That said, I have a number of concerns with the current manuscript, which I have detailed below.

We thank the reviewer for their positive evaluation and insightful comments, which have all been addressed in our detailed responses below. Addressing these comments has helped us to clarify the manuscript by putting a much greater focus on model free, descriptive analyses.

Point by point reply

1. I think the key points require additional behavioral analyses for support – and the current analyses do not scratch the surface of the rich behavioral data collected. Without descriptive analysis of behavior, it is tough to know exactly how well the models are fitting – or what elements of behavioral data might be missing in the model. In my mind, the descriptive analyses should be fairly straight forward – for example, the authors could construct a GLM that examined accuracy as a function of the number of 1) previous trials, 2) previous within factor trials, and 3) the sum of absolute correlations with previously observed traits. This would allow the authors to test their key points while making fewer assumptions – also provide summary statistics that can be tested from simulated model data in order or to validate their sufficiency.

These are excellent suggestions. Thank you. In our original submission, we have focused almost exclusively on analyses in terms of computational models. We have followed the reviewer's suggestion and performed a number of model-free analyses, including the suggested GLM analyses, which corroborate (and visualize) our key points.

Specifically, we conducted GLMs akin to a standard fMRI analysis. We first calculated three separate regressions that predict the accuracy per trial (absolute prediction error, absPE) per participant. Each of these regressions included one of the three regressors suggested by the reviewer. We list these regressors here and explain how they link to our models:

- 1. Total number of previous trials: This regression assesses the decrease of absPEs over time and thus relates to updating in the standard Rescorla-Wagner model. In previous analyses we had only included correlations along these lines.*
- 2. Total number of previous trials within the factor to which that trait belongs: This regression investigates the behaviour captured by the coarse granularity models (i.e., the more items one has experienced from a factor the better the estimate for that specific factor should be).*

3. *Summed absolute correlations of the previous items: The information content for all previous items is weighted by their correlation to the current item (i.e., it sums the absolute correlation coefficients from all previous items with the current item as a measure of information content). This regression aims at showing in descriptive analyses that fine granularity models can explain participants' behaviour. We were mostly interested in effects of this regressor since one of our main points is that participants use fine-grained item-level correlations when learning.*

Each of these regressors reflects a core feature of the specific models that we use. In a second-level analysis, participants' individual parameter estimates were subjected to a one-sided one-sample t-test to test if the slope was significantly different from zero in the negative direction (indicating a decrease of the absolute PE over time).

Crucially, the parameter estimates from regression #3 (for the summed absolute correlation for the previous items) were significantly different from zero for experiments 1-4 (Figure 1A-D). This provides descriptive analyses that participants used the fine-grained similarity between personality traits to some degree in these experiments.

As expected from our model comparison results, experiments 2 & 3 also showed significant effects for parameter estimates from regressions #1 & #2. In particular, regression #2 (with the total previous number of trials in a specific factor) indicated the use of coarser granularity in these tasks. The significant effects of regressors #1 & #2 might be due to the fact that items in these experiments (i.e., experiments 2 and 3) belonged to only two factors and thus the number of items per factor was larger by design, resulting in more statistical power.

The parameter estimates were not significantly different from zero in experiment 5 (Figure 1E), potentially as a result of the more limited answer scale (1-5 compared to 1-8) and a lesser number of items per factor.

As a caveat for these analyses, we mention that the three regressors were highly correlated (e.g., ρ between .76 - .92 across all five experiments). We therefore conducted separated regressions and did not conduct regressions including all three regressors. As the reviewer notes, these analyses should be taken as descriptive and complementary to the model-based analyses in this manuscript.

*For your convenience, we copied parts of the manuscript that address these changes (highlighted in **bold**) below. Additions are from the results and methods, moreover we added **figure 4** to support our explanations.*

Experiment 1: Real Profiles & Wide traits

... Model-free analyses indicated that participants were learning: absolute PEs decreased over trials (Figure 3A). A pairwise Pearson correlation between trial number and the mean of the absolute PE over all participants showed a large negative correlation, $r(58) = -.525$, $p < .001$, model simulations of the winning model also showed this decrease in absolute PEs over time $r(58) = -.721$, $p < .001$. **Moreover, model-free analysis using a GLM on participants' answer accuracy also indicated the use of the fine-grained correlation structures (Figure 4A).** ...

Experiment 2: Constructed Profiles & Narrow Traits

... The model-free analysis corroborated that participants were learning in the RW sense, by reducing prediction errors over time. We calculated the Pearson correlation coefficient to compare the mean absolute PEs with the trial number and found a negative correlation, $r(58) = -.497$, $p < .001$, which shows that PEs

decreased over trials. Moreover, model simulations with the winning model showed a similar decrease in the absolute PE over time $r(58) = -.598, p < .001$ (**Figure 3B**). **Further model-free evidence for the modelling framework came from a GLM that tested for all three major components of our models (i.e., decrease in the PE over time and the use of both coarse and fine granularity). For this GLM all three regressors were significant, which indicated that participants learned over trials but also seemed to use both coarse and fine granularities (Figure 4B). ...**

Experiment 3: Real Profiles & Narrow Traits

... Model-free analysis, using Pearson correlation between trial number and absolute PEs revealed a negative correlation ($r(58) = -.631, p < .001$) indicating that participants learned over the course of a run (**Figure 3C**). Model simulations using the winning models also showed a decrease in the absolute PE ($r(58) = -.903, p < .001$). **In a further model-free analysis we calculated a GLM with three regressors. These three regressors captured learning over time as well as coarse- & fine-granular learning. We found all three regressors to be significant, indicating that participants learned over time but also made use of both the granularity structures (Figure 4C). ...**

Experiment 4: Fashion Models

... Model-free analysis that tests for a relation between the trial numbers and absolute PEs using Pearson correlation indicated a negative correlation $r(58) = -.551, p < .001$, similarly simulations of the winning model also found a decrease in the abs PE $r(58) = -.740, p < .001$. As in the previous experiments, this decrease in the PE over time is an indication of learning (Figure 3D). **Finally, another model-free analysis that uses a GLM that tests for participants' use of RL and both coarse- and fine- learning indicated, in-line with previous results, that participants used fine-granular representations (Figure 4D). ...**

Experiment 5: IPIP items

... Model-free analysis of the absolute PEs found a negative correlation ($r(48) = -.518, p < .001$) indicating that participants learned over time (**Figure 3E**). Simulations using the winning model on the same dataset revealed a large negative correlation $r(48) = -.918, p < .001$, in line with the real data. **Finally, a GLM with separate regressors that each represent a core part of our models (i.e., decrease of the PEs over time, and the use of both coarse and fine granularity) did not find evidence for their use by participants, potentially as a result of the smaller answer scale (1-5) and a lesser number of items per factor (Figure 4E). ...**

Model-Free Behavioural Analysis

To test whether the PEs have a downward trend over time (an indication of learning), we calculated the average of the absolute PEs per profile per trial for all participants. Profile data were then averaged into one vector (with length being the number of trials) of the average absolute PEs over time per experiment. To determine whether there was a significant decrease of PEs over time, we calculated the Pearson correlation coefficient on these absolute average PEs and the corresponding trial number. A negative Pearson correlation indicates a decrease in the absolute PEs over the trials. The same procedure was applied to data that was simulated based on the winning (i.e., best fitting) model for that experiment.

Moreover, for each experiment separately, we calculated a GLM that explained the accuracy per trial with three regressors. Each regressor was initially modelled as a separate regression and consisted of an integral part of our computational models. In brief, the regressors were the following:

1) total number of previous trials (this assesses the decrease of the PEs over time and thus learning in standard reinforcement learning), 2) total number of previous trials within the factor to which the current item belongs (investigates the behaviour captured by the coarse granularity models), and 3) the summed absolute correlations of the previous items with the current item (this assesses the fine granularity models where the information content of the previous items is weighted by their correlation to the current item). In a second-level analysis, participants' individual parameter estimates were subjected to a one-sided one-sample t-test to test if the slope was significantly different from zero in the negative direction (indicating a decrease of the absolute PE over time).

Figure 4. General linear model on three regressors of interest for all five experiments separately. Model-free analysis on core features of our models. Three separate regressions predicted the accuracy per trial per participant. Each regression had one specific regressor for a specific feature of the models. These regressors reflected:

1) Rescorla-Wagner RL, with the total number of previous trials, 2) Coarse Granularity, with the total number of previous trials within a factor, and 3) Fine Granularity, with the summed absolute correlation coefficients of the previous items with the current item.

Each regressor is indicated by a different colour, individual data points are the parameter estimates per participant (summarized by the boxplots). In experiments 1-4, the parameter estimates for the summed absolute correlation of previous items were significantly below zero (one-sided t-test; * indicates $p < 0.05$, ** indicates $p < 0.001$, no correction for multiple comparisons). These analyses suggest that over the time course of the experiment absolute PEs become smaller depending on the total amount of previous items that had been seen weighed by their correlation to the item in each trial. However, conclusions about results should be taken with caution since all three regressors were highly correlated (ρ between .76 - .92).

2. The use of the word optimal in the manuscript feels misleading... I would say in most cases the authors seem to use it as synonymous with beneficial. I was also unclear on what fitting to the feedback actually means. Do the authors mean they chose parameter values that minimized prediction errors? I think it would be useful for the authors to actually characterize what optimal behavior would look like in their task – presumably this could be done by assuming that “traits” are generated from a multivariate distribution with a mean (reference point) and covariance matched to the ground truth data – and inverting this process to do inference. But in any case, they should be more clear about what they mean when they say optimal, as I have strong doubts that any of the models considered here are optimal in either the colloquial or technical sense.

We completely agree that our use of the word ‘optimal’ has been confusing. We mean the best performing models (in the set of or our models). That is, we determine which model performs best when having to solve the learning task that our participants faced. We did not mean “optimal” in the sense of an a priori optimal inference as for example given in Bayesian inferences for suitable problems. We rather meant best performing in a similar sense that for instance machine learning algorithms are compared with each other: The algorithm winning the competition with the highest accuracy is deemed to be the best current algorithm. This does not necessarily imply that these best performing models perform very well overall.

With these best performing models we mean that models that are fitted to the true outcome ratings (i.e., the ones that participants saw as an outcome after their rating) instead of fitting them to the participant’s ratings (i.e., during ‘regular’ model fitting). This allows us to test how well each model performs—in the place of an in silico participant—if it has full information.

*We have added a specific section in the methods to explain the concept of the best performing model in more detail. Moreover, we have added **Supplementary Figure 7** to support these explanations.*

Best Performing Models (in the set)

Normally, models are fit on the answers participants give during the task, this helps us estimate which of the strategies (i.e., models) participants used during learning. But in order to understand the usefulness of the strategies we estimate the best performing model (in the set). This is achieved by fitting the models on the profile answers (i.e., the same way the participants experienced the experiment) instead of the participant answers (**Supplementary Figure 7**). Comparing these best performing models with the ones that best fit the participants’ strategies we can determine whether participants’ strategies were well fit to the task demands.

Supplementary Figure 7. The rationale of the best performing models (in the set).

Data in this figure are selected from two participants (top and bottom row) from Experiment 2. For the axis, data have been sorted according to time in experiment and according to factor i.e., items have been sorted based on when they were encountered within the factors agreeableness and conscientiousness. The best performing models are encountered in Figure 2C and Figure 5E-H

The left panels (A & D) show the profile answers in grey (this is the feedback participants received) and participant answers are indicated in green. During regular model fitting, models are fitted on these participant answers (rightmost panels C & F), here we see that the top participant's answers are best explained by model 5 (orange) and the answers by the bottom participant by model 1 (blue).

The middle column (B & E) shows the 'best performing models', these are models fitted on profile answers (grey line) i.e., models are fitted as if they have full information about the profiles, like participants had. For both profiles model 5 is the best performing model in our set of models.

Comparing the best fitting models with the models that best explained participant data helps us in understanding whether participants used an effective strategy or not.

3. The models could be described more clearly. My understanding is that the fine and coarse granularity models are just using different basis functions for generalizing feedback across traits. A typical basis-function approximation would learn weights for each basis function, and make predictions by combining relevant bases. However, the existing equations make it look like only a single element of the P matrix is updated on each trial. It would be useful if the authors could clarify exactly what the model is updating, and if it is indeed only updating a single prediction, how the model affords generalization.

*Both the fine and coarse granularity models generalize over all traits. The coarse granularity models update one value per factor based on the PEs for each trait that belongs to the factor in question. As the reviewer correctly notes, one could understand this updating process in terms of applying the PE to a basis function (which is a vector with the length of the number of traits). For coarse granularity models, there are as many basis functions as factors. All items within a factor get 1's and those outside the factor get 0's, i.e., a PE is multiplied by 1 for traits in the factor and by 0 for traits outside the factor. The fine granularity models update all traits based on how similar (i.e., correlated) all traits in the set are to the current trait in question (traits that are related will be updated more compared to traits that are less related). This means that the currently used basis function consists of the column of the current item within the correlation matrix (see **Supplementary Figure 1E**).*

We have expanded the explanations of the models in the methods (changes highlighted in **bold**). To supplement these explanations we created **Supplementary Figure 1** to function as a tutorial of each of the model specifics. Finally, we updated **Figure 1B** to be a clearer overview of the models.

Model 1: No Learning

Model 1 assumes that participants perform a linear transformation of the reference point (RP) population mean to predict (P) others' personalities. **This model performs like a standard linear regression where b_0 represents the intercept and b_1 the slope.**

$$P = RP * b_1 + b_0$$

Model 2: Coarse Granularity

Updates an average factor value (**F, factor**) for the next trial estimate based on the prediction error (PE) and the current estimate. **That is, on a trial-by-trial basis the model updates a value that represents the average value for this factor. The speed of learning is determined by the free parameter α (bounded [0 1]), with a second free parameter; starting value (bounded [1 8]), determining the value at which each factor will be initialized.**

$$P_{(t+1,F)} = P_{(t,F)} + \alpha * PE$$

Model 3: Coarse Granularity & Population RP

Model 3 expands Model 2 by adding the reference point (RP) in form of a population mean per trait. **During each trial model 3 learns an average factor value like model 2 but it adds information based on the Reference Points (i.e., the average population rating on this trait). Information is integrated using the weighting parameter γ (bounded [0 1]). This parameter determines how much participants rely on just the RL from model 2 or the RP (0.5 indicating that both are used equally).** Like Model 2, Model 3 also uses the parameters learning rate (α) and the starting value for its initial estimate for a new profile. Parameters α , γ , and the starting value are free parameters.

$$P_{(t+1,F)} = \gamma * RP + (1 - \gamma) * (P_{(t,F)} + \alpha * PE)$$

Model 4: Fine Granularity

Model 4 employs fine-grained granularity and updates all items (**All**) at once based on how correlated they are to the current item (**Supplementary Figure 1E**). This similarity matrix (SIM) was calculated before the experiment based on separate samples (see section Granularity). **Thus on a trial-by-trial basis, Model 4 updates the current trait based on the PE and the learning rate (i.e., it is perfectly correlated with itself) all the other traits will get updated based on the PE, the learning rate and their similarity to the current item (i.e., their correlation), which means that traits that are more similar get updated more.** Model 4 also makes use of a starting value, since this model updates all items for every step of learning the initial estimate of the starting value is initialized for all items at the onset of learning.

$$P_{(t+1,All)} = P_{(t,All)} + \alpha * PE * SIM$$

Model 5: Fine Granularity & Population RP

Like the coarse granularity models, Model 5 expands the previous model (i.e., Model 4) by adding the reference point (RP) in the form of the population mean. **That is, for every trial learning happens (as in model 4) but additional information in the form of an average trait rating is added. The degree to which this information (i.e., RP and Granularity) is integrated is determined by γ .** Similar to Model 4,

Model 5 uses the starting value to initialize all item estimates at the onset of learning. Free parameters are α , γ , and starting value.

$$P_{(t+1,All)} = \gamma * RP + (1 - \gamma) * (P_{(t,All)} + \alpha * PE * SIM)$$

Supplementary Figure 1. Highlighting the main parts of our modelling approach. All data is simulated and purely intended for visualization purposes.

A. Behavioural Data. An idealized learning response pattern over time. The black line represents to-be-learned values on a given task (e.g., self-ratings on personality traits by a certain person). The grey line represents a participant's answers trying to estimate the self-ratings. This participant is learning because over time her answers get closer to the outcome (self-ratings). Similarly, this is indicated by the decrease in the absolute prediction error (PE, transparent grey) over time. Using computational models we can attempt to capture such behaviour. In the following sections we

explain several models of differing complexity that all focus on different pieces of information regarding the self-ratings.

Building a model

In general, the black line represents self-ratings on personality traits. A real-life experiment would have randomly ordered items but we have ordered them based on five arbitrary factors for easier visualization, where each factor consists of multiple trials. Shaded (grey and white) backgrounds indicate the factor and coloured shaded regions in the lower parts of the plots indicate the PEs of the models with the same colour.

B. Factor Reference Points. A simple model could make use of only a Reference Point (RP) per factor (i.e., a person is represented as having a single mean value per factor). Added are two such RPs, one that fits the data reasonably well (green) and one that does not fit the data well (red). The shaded area at the bottom of the plot represents the absolute PE for both RPs. The worse fit of the red RP is indicated by the higher values of the absolute PE.

C. Individual Reference Points. A more complicated model could have RPs for each trait separately since this can capture idiosyncrasies within factors e.g., in Factors 3 and 4 these RPs capture that there is a group of higher and lower rated items. Something that is not present in the previous Factor RP model.

D. Reinforcement Learning. Standard Rescorla-Wagner models use the discrepancy between the expected and actual outcome values (PE) to incrementally update the estimate for the next time step, the magnitude of the updating is determined by the learning rate (usually denoted as: α). In this figure, an agent learns an average value for two factors i.e., individual items get presented that belong to one of two factors, the participant automatically categorizes these into their respective factor and learns an average value per factor. The horizontal coloured lines are the expected averages for the first (blue) and second (green) factor where shades indicate the time steps in learning. The first estimate for the first factor (blue) results in a PE of 3.4, after updating this estimate again the resulting PE is: 1.8, a third estimation results in a PE of: 1. Estimates in this figure are spaced out for visualization purposes, for the same reason the factors have been presented in a ordered fashion instead of interspersed.

E. Similarity Reinforcement Learning. Rescorla-Wagner models learn only one value at a time, this can be a trait or an average value (e.g., per factor). However, real-life learning happens broadly from few examples. A mechanism that can explain such generalized learning is to update all traits based on how correlated they are to the current trait.

In the top section, the black line represents a subsection of self-ratings on two factors. For visualization sake it is assumed that the participant expects each trait rating to be zero (grey dots on the x-axis). In this example the participant gets feedback on item 5 (black dot), which results in a PE of 5.11 (i.e., the difference between the expectation [0] and the outcome [5.11], the vertical black line between the two black dots). Instead of multiplying this by the learning rate and updating solely this item, the participant can update all traits based on their correlation to this current item. The similarity matrix on the right shows the correlations between items. Since the model is learning about the fifth item, the fifth row (could also be the fifth column) is selected to update all other items. On the bottom, this row has been enlarged and correlation coefficients have been added. Each item is then updated by multiplying the PE, the learning rate, and the correlation coefficient resulting in the new estimates (red dots). Item 5 is updated according to the standard RL. While only receiving feedback about item 5, updates occur for all other correlated items in the set. Items for which the correlation is (close to) zero are not simultaneously updated because learning about item 5 gives no or very little information about uncorrelated items.

F. Combining Factor RP & Similarity RL. Using a weighting parameter (γ) we can combine the previously discussed models into combination models. The shaded blue lines are several weighted combinations of the Factor RP and the Similarity RL. Notably, if the weighting parameter is either 0 or 1 it is simply one of the subcomponents (Factor RP or Similarity RL) of this combination model. In this example, equally weighting both subcomponents (γ : 0.5) most accurately follows the self-ratings and thus would be the best fitting model. Akin to the learning rate, the γ is usually a free parameter that is estimated during model fitting.

Figure 1. Overview of the experiment and models used

In this study, we tested computational models of how humans learn about others. In five distinct experiments participants performed a social learning task on several profiles of other persons (with every profile being presented in a separate run).

A) General overview of the learning task: Two trials for Experiment 1 (left) and Experiment 5 (right) are shown. During the learning task, participants estimated which self-ratings a person (here called profile) had given for specific traits (on a Likert scale from “does not apply at all” to “does apply very much”). After each estimate, participants received direct feedback in the form of the actual self-rating of that person. This process continued for all traits (in random order). The tasks shown here are translated to English, original German instructions can be found in **Supplementary Tables 1-3**.

B) To explore participant’s behaviour over the 5 experiments we constructed 5 main computational models (listed in the bottom right). Two main concepts varied in our computational models: First, **Reference Points (RPs)** represent what participants can use as a basis for estimating an average person (from an in- or out-group). Participants can use the RPs on each trait to compare this average person’s rating with their current estimate for the other person. Models 1, 3 and 5 used RPs and models 2 and 4 did not. In all experiments, we use a population mean of the average population (shown here for all items used in Experiment 5). Traits are ordered based on the Big-Five Factors that are shown in different shades of grey. In Experiment 4, we additionally used the stereotypical ratings of fashion models and we constructed models that use Self-Ratings as RP (**Supplementary Figure 10**). Second, **Granularity (G)** refers

to the level of detail in the represented structure of others' personality traits. The granularity matrix generalizes the PEs across similar items in two distinct ways (i.e., per Big-Five factors for "coarse granularity" and per individual traits based on how correlated they are to the current trait for "fine granularity"). Using these two concepts, models can be divided into three sets, which are depicted in three different colours:

1). **No Learning** (blue), consists of a single regression model, Model 1 [No Learning] that functions as a baseline model.

2). **Coarse Granularity** (pink), comprises two models that update their future estimates for the whole (Big-Five) factor to which the current adjective belongs, the first model in this set, Model 2 [Coarse Granularity] uses the standard Rescorla-Wagner (RW) function to update the factor estimates for every trial, the second model in this set, Model 3 [Coarse Granularity & Population RP] uses the aforementioned RW-model to update the factor estimates but also integrates information distilled from the population.

3). **Fine Granularity** (orange), consists of two models that update all adjectives. Crucially, the magnitude of updating on any trait depends on the correlation with the current trait (i.e., during learning one updates not only the estimate of the current trait but also those of all other traits based on how correlated they are to the current item). Like the Coarse models the first model in this set, Model 4 [Fine Granularity] updates all items according to the Fine Granularity, whereas the second model, Model 5 [Fine Granularity & Population RP] updates items not only according to the Fine Granularity but also the RP. More detailed descriptions of the models and their separate functionalities can be found in **Supplementary Figure 1**.

All models that use Rescorla-Wagner RL make use of PEs, an example of the PEs of a randomly selected participant is shown in the top right corner.

P = prediction, Int = intercept, RP = reference point, α = learning rate, PE = prediction error, γ = weighting parameter, F = (generalizes over Factor) coarse granularity, All = (generalizes over All items) fine granularity, SIM = similarity matrix.

4. I don't quite understand the point of figure 4, ...

*The idea of Figure 4 (now **Supplementary Figures 11-16**) was to give some more clarity on the model fitting procedure. That is, we performed a grid-search across the parameter space for each model (i.e., 2 or 3 dimensions) to find out where an optimal fit could be achieved. With the optimal fit we mean the lowest sum of squared error (SSE) in each figure. This SSE is indicated by colours where blue colours indicate high and red low SSEs. The results from this grid-search were then compared to participants' fitted parameters. Results indicating that, on the one hand, model-fitting procedures functioned as expected. That is, parameters fitted to participants' responses were largely in the areas with low SSEs, this can be seen by the dots (i.e., the fitted parameters) superimposed on the coloured grid-search image. On the other, that the models were functionally used as designed. This means that there were no a-priori unexpected ways the models could fit data (e.g., by ignoring one parameter completely), indicating that our models were well suited for capturing (parts of) this behavioural data. We think there is validity in such transparency but since these analyses are not addressing our main hypotheses but merely corroborating the validity of our modelling approach, we have now placed these figures in the supplementary materials.*

... moreover, I think I would have been much more convinced that the modeling results were robust had the authors done some more nuanced behavioral analysis, and then performed posterior predictive checking to show that the best models were able to capture summary statistics from the data – whereas the worse performing models were not. Not only should synthetic data match learning curves (eg. Figure 2), but they should also be able to recover summary statistics from the descriptive analyses suggested above. It would also be useful if the actual data could be plotted as a function of model predicted response probability for conditions of interest.

Among other model-free analysis mentioned previously we have updated Figure 3 (previously Figure 2) to illustrate the overlap between independent model simulations of the winning model and real data averaged across participants. Results indicate a good overlap between experimental and simulated data. Results from this model free analysis are mentioned in each of the experiments results. To not lengthen this document unnecessarily these responses are included in response to point 1 by reviewer 1 in the sections that are not bold.

Figure 3. Mean absolute PE over time per experiment for participants and simulated data from the winning model.

A decrease of the PEs over time (i.e., trials in the experimental runs) can be interpreted as learning because participants' estimates got closer to the feedback they received. Participants' absolute PEs are plotted in black, the error (red) is the SEM and the straight gray line is the least squares fit (data on the left y-axis). Model-free analysis: data in shades of blue represents simulations performed on the best fitting models for that experiment (data on the right y-axis), absolute PEs (light blue line), with the SEM in the shaded blue region, the straight (dark blue) line is the least squares fit. Correlation coefficients (ρ) and resulting p-values for both the real and simulated data have been added in their corresponding colour (black participants, blue simulations).

5. It would be useful for the correlation structure to be depicted for experiment 2 – also I was unclear on what correlations were used by the fine-grained model in this experiment. The actual correlation structure for the task, or the inappropriate one more attuned to “real data”?

All experiments used Pearson’s correlations calculated from relevant large independent data sets (Korn et al., 2012, 2014; Oganian, Heekeren, Korn, 2018) for experiments 1-4 and the Open Source Psychometrics Project (<https://openpsychometrics.org/>) for experiment 5. The fine-grained models used correlation matrices, derived from these independent studies, on the specific traits that occurred in each of the five experiments (e.g., for experiment 2: 30 traits on agreeableness and conscientiousness, see Figure below).

The profiles for experiment 2 were simulated by taking an average factor value and adding random noise to create individual trait ratings. This means that the innate similarities that are usually found in real self-ratings were absent in these profiles. Therefore, the appropriate correlation structure to represent experiment 2 would be the coarse granularity. This was already present in Figure 1 of the manuscript but has been clarified in its caption.

We would like to emphasize that it was an open empirical question whether participants would use these coarse granularities or still use the fine granularity (like they did in experiment 1). Neither of the five experiments entailed external information as to which strategy to use. The only information participants received was the profile feedback after each trial. Therefore, the shift in strategies is not a logical consequence of a given experiment but a rather surprising finding.

In the methods section of the manuscript we have added a section dedicated to granularity. We have also added the figure below as an extra visualization. This figure is not included in the manuscript.

Granularity

Granularity refers to the level of detail with which participants represent the others’ traits. This means that one can either have a summary value per (Big-5) factor, or learn a separate value for each trait per person (Figure 1B, and Supplementary Figure 1D, E). The former, Coarse Granularity, is mathematically defined by having a single summary value for each factor which will be updated only when one learns about a trait that belongs to that factor. The latter, Fine Granularity, assumes one learns a separate value per trait but also updates all other traits based on how similar (i.e., correlated) they are to this current trait. In experiments 1-4, these similarity matrices were calculated using Pearson’s correlation on self-ratings from two published independent laboratory studies (Korn et al., 2012, 2014) as well as data from previous online studies (see Oganian, Heekeren, Korn, 2018). Specifically, we selected four participants from the 27 participants in Korn et al. (2012) for the four profiles in Experiment 1. The self-ratings of the remaining 23 participants as well as the self-ratings from the 78 healthy control participants in Korn et al. (2014) were used for the similarity matrix. These 101 participants who completed laboratory studies gave ratings on the 80 traits of the original list. Additionally, for the calculation of the similarity matrix we included online ratings of 734 participants, who each gave self-ratings on 50 pseudorandom traits from the overall list of 125 traits. For Experiment 5, we calculated the correlation matrix from the (> 1 million) sample from the Open Source Psychometrics Project (<https://openpsychometrics.org/>).

The similarity matrix used for experiments 2 and 3 (this figure has not been added in the manuscript).
Both agreeableness and conscientiousness consist of 30 traits.

6. I was a bit unclear where the reference points come from. Are they taken from the data itself? Is the idea that participants have estimates of these reference points from real world experience? Or from task experience?

The reference points are based on the same independent (self-rating) data sets that are used for calculating the item-level correlations. The reference points are calculated by taking an average over these self-ratings. For experiment 4 we used these average self-ratings and also fashion model ratings that participants gave prior to the learning task. Moreover, in response to question 4.1 by reviewer 2 we have also added participant self-ratings as RPs. All these RPs were, of course, different in their scores (also see Figure 6 in this document).

We assume that participants have such structures internalized from real world experience (i.e., over time one learns the average scores for specific groups of people). Reference points refer to prior expectations of a person's personality. The idea being that this person belongs to a specific group (e.g., students, fashion models) which provides information about their average personality trait ratings. Consequently one can use these RPs next to regular learning (e.g., most people from this specific group are generous I therefore should rate them higher on generous, but I perceive them to have less diligence, so I should score them lower on this trait).

In the methods section we have added a section dedicated to the reference points.

Reference Points

Reference points refer to a-priori expectation of a person's personality. This means that a person has an idea of the trait ratings from an average person who belongs to the same group as the person who is being judged and uses these ratings next to regular learning (Supplementary Figure 1B, C) (e.g., most people from this specific group are generous I therefore should rate them higher on generous, but I perceive them, on average, to have less diligence, so I should score them lower on this trait). The reference point data was calculated by taking the mean value per trait of the same datasets that were used to calculate the similarity matrices. In Experiment 4, we used the stereotypical ratings given by participants before the learning task. To test whether there was a difference between the self-ratings of the independent study and the stereotypical ratings we calculated an independent samples t-test on the trait averages.

7. Perhaps it is my lack of familiarity with OSF (I've never used it), but I am unable to see any details in the link to the preregistration for experiments 4&5. I see that there is a site, but I was unable to verify experimental details/predictions.

Thank you for this comment, the link was to the 'homepage' of the preregistration but has been updated to the actual page (<https://osf.io/8r6gv>) in all three sections where it is linked.

Only one example has been added to not lengthen this document unnecessarily.

Experiments 4 & 5 were preregistered on the Open Science Framework prior to data collection (<https://osf.io/8r6gv>). In this preregistration, we specified our expected sample size, exclusion criteria, measured variables and computational models. Most of these considerations also applied to the first three experiments, which we did not preregister.

8. The presentation in this manuscript, which includes both writing and figures, could be improved.

Integrating all reviewers' comments gave us new views on the manuscript, as a consequence, we have rewritten large parts of the manuscript to be more clear and coherent (mainly Discussion and Methods section). Moreover, the figures all have a coherent style. The specific colour combinations used to represent the models were chosen to also be visible for people with colour vision deficiency. We are using the same colours throughout the manuscript to be coherent (e.g., the parameter recoveries in the supplementary materials reflect the model colours used in Figures 1, 3, and 5). Moreover, we have included and updated figures in both the main text as well as supplementary materials to support our writing. In the manuscript Figures 1, 2, 3, and 5 have been updated and Figure 4 has been added. Moreover, Supplementary Figures: 1, 7, 8, 10 are newly added and Supplementary Figures: 3-6 have been updated.

9. Line 446: I'm unclear what exactly is going into this ttest – shouldn't each trait have its own reference point? How is this hypothesis test yielding a single t-stat.

The RPs represent a single value per trait (i.e., an average calculated from an independent sample). Because we have two independent RP samples (student and fashion models) with the same items we can simply run a paired t-test on these two samples. This shows us whether the RPs as a whole are different (see Figure below).

We have included a more thorough explanation of this t-test in the results section of Experiment 4.

Experiment 4: Fashion Models

... We investigated whether participants held stereotypical views of the fashion models. Before starting the learning task participants were asked to rate their impression of an average fashion model on all 60 traits that were used in the experiment. From these ratings, we calculated a new reference point that was based on these perceived fashion model self-ratings. To compare the previously used student reference points (M: 6.01, SD: .69) with the perceived fashion model reference points (M: 5.13, SD: .32) we conducted a paired sample t-test, this indicated that there was a significant difference between the student reference point and that of the fashion models, $t(59) = -9.7137$ $p < .001$

Comparing Reference Points: student self-ratings (orange) with expected model fashion ratings (blue) (this figure has not been added in the manuscript). Difference between groups is confirmed using a paired sample t-test between the students (M: 6.01, SD: .69) and expected fashion models (M: 5.13, SD: .32) self-ratings.

Reviewer #2 (Remarks to the Author):

Review: " How do humans learn about other people? Incorporating social knowledge structures into reinforcement learning"

In this manuscript, the authors investigate how people learn about others' personality across multiple experiments. They find, generally, that a learning model, which considers both personality of the "average person" and the correlation structure of personality traits best explains the data (predictions of others personality).

My opinion of this paper is conflicted. On the one hand, there are several interesting, cool, and novel ideas (especially using the correlation between traits to inform the model), and the methods are rather rigorous. On the other hand, I find the framing of the paper overstated, and am bothered by many seemingly arbitrary modeling choices, and the very specific nature of the experimental paradigm. As it stands, the manuscript seems to describe the rather modest beginnings of a very interesting research program. I wish I could be more positive about the manuscript per se however.

I outline my concerns below.

We thank the reviewer for the critical points raised. Addressing these has helped us to clarify the writing and analyses in the manuscript. We believe to now put the scope of our models in a better perspective and support them with additional analyses.

Point by point reply

1. Conceptual. The title of the paper highlights reinforcement learning, but this is a misnomer. The task the authors use is a classic supervised learning setting. Participants provide a prediction and get feedback on the same scale. The prediction error is the difference between prediction and feedback (e.g., Prediction = 3, Feedback = 5, PE = 5-3 = 2). In other words, the task is basically (or exactly) a regression problem. Participants learn, but it's misleading to call this reinforcement learning. The framing should be changed to acknowledge this.

We thank the reviewer for this comment. We would like to clarify that this learning task can be modelled as both a regression problem and as a reinforcement learning problem (according to variants of the Rescorla-Wagner algorithm). Indeed, we have used both approaches and compared the relevant models using appropriate metrics for model comparison. Following this comment and comment 1 by reviewer 1, we have added further regression analyses, which corroborate the analogy between regressions and RL.

We specify the differences here:

The reinforcement learning term summarized a key component of most models, updating expectations about someone's personality traits based on trial-by-trial feedback. Models 2-5 are hybrid models that combine Rescorla-Wagner reinforcement learning (Rescorla & Wagner, 1972) with the two knowledge structures: reference points and granularity). The Rescorla-Wagner model is a subtype of reinforcement learning and is based on models for classical conditioning. Agents using this strategy 'learn' over multiple trials by constantly updating a discrepancy between expectation and outcome, where the magnitude of updating (i.e., the speed of learning and discounting the past) is weighted by the learning rate (usually denoted by α). As the reviewer correctly states the prediction error for a given trial t can be formulized in the following way:

$$PE_t = Outcome_t - Expectation_t$$

In our experiments the expectation is the expected self-rating and the outcome the feedback of the actual self-ratings. The Rescorla-Wagner formula for updating the prediction on the next trial ($t+1$) using the PE would be:

$$\text{Prediction}_{(t+1)} = \text{Prediction}_t + \alpha * PE_t$$

Thus the prediction of the next self-rating gets updated based on the summed value of the previous PE multiplied by the learning rate and added to the last self-rating (for which the PE was generated).

To give a brief example in numbers: in Figure 1A of the manuscript a participant has to estimate Lisa's generosity and expects Lisa to 'score' 7, the feedback from Lisa's self-rating is 5, this results in a Prediction Error of $5 - 7 = -2$. Assuming the participant has a learning rate of 0.5, the updated prediction of Lisa's generosity by the participant would be the previous prediction [7] plus the Learning Rate [0.5] multiplied by the Prediction Error [-2] for a new prediction of: $7 + (0.5 * -2) = 6$. It is important to note that this expectation is a 'cached' value in the Rescorla-Wagner models. That is, this value is being tracked in memory and gets overwritten after every trial to get closer and closer to the true estimate. So the expectation for another trial of generosity would be the new value 6, resulting in a PE of -1 and consequently a new expectation of 5.5. It is because of this caching that these model cannot be reduced to regressions.

We do indeed employ a standard regression in model 1 [No Learning]. However, this standard regression does not learn over trials (like reinforcement learning) but rather fits a linear relationship between the RP and the profile answers to all trials directly. In the main domains of machine learning, regressions is categorized as a supervised learning system whereas reinforcement learning is in its own domain (Bonaccorso, 2017).

We clarified these distinctions in the introduction (see below) and the model section of the methods (please see full copied section in response to reviewer 1 point 1). Moreover, we have added **Supplementary Figure 1** explaining the subcomponents of the models in more detail and have improved **Figure 1B** to clearly introduce the models.

Introduction

... In this study, we aimed at testing how humans learn about the multi-dimensional personality of others. In order to account for the complexities of learning about human personality, **we employed hybrid learning models that weigh prior experience, contextual knowledge, and RL (Camerer & Ho, 1999)**. We therefore constructed and tested a number of computational models with varying complexities, from a simple linear regression that functions as a baseline to hybrid models that combine standard Rescorla and Wagner RL models (Rescorla & Wagner, 1972) with two social knowledge structures that we refer to as **Reference Points (RPs)** and **Granularity (G)**. ...

2. Limitations of the experimental paradigm. The paradigm the authors use is clear, but also very limited in scope (explicit learning of other people's personality based on their explicit feedback). Given the RL framing, I anticipated variations of the paradigm where participants actually had to use what they learned for something (payoff relevant). As it stands, the set of studies do not provide the general account of social learning that the introduction and discussion imply. For example, the authors state "Social learning is a very broad domain where even fairly standard social interactions can be different every time. We aim to introduce models that can be applied to a majority of these social learning tasks", and "we expect that the learning models presented here will also be applicable to different domains of

learning such as other domains of social learning “. This is not explained in more detail, and I have a hard time seeing how a model developed for such a specific application could have so general implications. I wish the authors had pushed this further and tested the logic of their model in other paradigms as well.

We completely agree with the reviewer that adding other domains of learning would be very interesting. As shown over the five experiments our model framework can generalize to different experiments and profiles within a social learning paradigm.

These claims are reinforced by previous work from our lab (Rosenblau et al., 2020) that showed that the similarity models investigated here can successfully be applied to learning about others’ preferences. That is, the similarity structure that can be derived from preference items (e.g., candy and fruit for coarse grained food items and chocolate vs cookie for fine grained candy items) can explain parts of participants’ learning about others’ preferences.

Moreover, recent studies have found neural evidence that humans encode specific social knowledge structures employed in our models (i.e., fine grained correlation matrices) (Stolier et al., 2020; Thornton & Tamir, 2021) suggesting a center role of item-level similarity structures in social cognition. Here we focus on establishing the social knowledge structures that people use during learning. In future experiments, we plan to use these knowledge structures in decision making tasks to investigate their role for action selections or payoffs.

Concerning the scope of the experiment, we expect these models to generalize to various (non-) social contexts. What makes these models so widely applicable, in our opinion, is the use of standard statistical metrics that are shared among various contexts and learning processes. It is of course our intention to explore in future work how well this framework can generalize to other paradigms (e.g., decision making, preferences), but would like to argue that this would muddy the manuscript more than enlighten it at this time point

*This more detailed explanation of our views on the scope of the models has been added in the discussion. Additions are highlighted in **bold**.*

Scope of models

Social learning is a very broad domain. Even within a certain social learning domain, there is large situational variability that determines which learning strategy is most suitable. In five consecutive experiments that vary with respect to participant populations, items and the to-be learned profiles, we introduced a defined set of computational models that generalize across experiments and pick up the systematic and subtle differences between experiments. The goal of this manuscript was to introduce the modeling framework in a clear and concise manner and not be exhaustive in introducing its potential applications. Nonetheless, we would like to argue for the potential broad scope of their applicability to a majority of social learning settings. As a primer for this broader scope we included experiment 5, in which we used a different set of stimuli for which, as expected, the models still functioned like the former experiments.

We conjecture that these models are especially useful for learning about personality traits, as shown over the five experiments. However, a very similar model space has also successfully been applied to learning about others’ preferences (Rosenblau et al., 2020). The models integrated similarity structure that can be derived from preference items (coarser grained structures such as candy and fruit and fine-grained item-level correlations). A fine-grained similarity learning model captured how participants learned about others’ preferences best.

Moreover, recent studies have found neural evidence that humans encode specific social knowledge structures employed in our models (i.e., fine grained correlation matrices) (Stolier et al., 2020; Thornton & Tamir, 2021) suggesting a central role of item-level similarity structures in social cognition. The relative simplicity of our models makes them adaptable to various multidimensional representations across different learning domains. These results are corroborated outside of the social domain by Roweis & Saul (2000) who highlight human expertise in representing complex abstract structures akin to the granularity structure introduced here.

In a general modelling framework we argue that our models relate to differences on a continuum between model-free (MF) and model-based (MB) RL (Dolan & Dayan, 2013). The details captured by granularity reveal complexities that MF-RL does not capture: MF learns a single value for any given stimulus in a given learning situation. Likewise, our models with coarse granularity learn a summary value for each of the Big-Five factors. In their simpler form, we thus think of our models using coarse granularity to be more similar to MF-RL rather than to employ full MB-RL (i.e., coarse granularity functions as a look-up chart of traits). The more complex models with fine granularity can be viewed as more like MB-RL as they incorporate a full representation of the traits and the similarities between them. This split—akin to differences between MB-RL and MF-RL—suggests that our models can distinguish between more costly optimal models and more efficient heuristic models, similar to models on optimal versus heuristic decision-making. (Korn & Bach, 2019; Korn & Bach, 2018).

3. Study 2 seems odd to me. The authors created low granularity personality traits and found that a low granularity model fit the data best. Could it logically be any other way?

Our hypothesis for study 2 was that due to the artificial profiles, participants would pick up on the lack of relationships between items and revert to a simpler model. But it could still be logically otherwise. Please allow us to specify how: Participants never get any specific clues about how the profiles are organized (i.e., that items can be sorted by factor, how many of such factors are presented per profile, and how many items per factor). Moreover, participants are not specifically instructed on how to learn or update their estimates. Therefore, whether participants use the complex model 5 [Fine Granularity and Population Reference Points] or a simpler granularity structure does not logically follow from the study setup or instructions. Instead, participants represent finer- or coarser-grained knowledge based on their perception of the task at hand without any clues given.

*We have made this clearer in the results of experiment 2 and in the section dedicated to granularity in the discussion. Changes highlighted in **bold**.*

Experiment 2: Constructed Profiles & Narrow Traits

... We hypothesized that for this experiment using a fine-grained similarity structure (i.e., single trait similarities) should be less advantageous than employing coarse granularity (e.g., average values per Big-Five factor). **Otherwise, the experimental set-up was analogous to the first experiment (i.e., participants were not aware that the profiles were artificial nor that they would only learn about two factors). Participants were not given any clues that would encourage them to use a coarse instead of fine granularity structure during learning. ...**

Granularity

... Our findings might indicate that humans flexibly change their representations of social knowledge to match implicit task demands. **Importantly, participants could only derive clues about how to adapt their strategy from trial-level item and feedback information, as the general task frameworks and instructions were kept very similar across experiments.** ...

4. Analysis. As I mentioned, I do like the correlation matrix approach used for learning about others' "personality". This approach seems to have potential. Furthermore, the modeling analysis is well done and rigorous (e.g., recovery, simulations). However, I am left puzzled by some aspects.

Thank you for these positive comments. Please allow us to explain our methods in more detail below.

4.1. The Reference Point is a key aspect of the analysis. However, very limited attention is given to the reference point. As I understand it, the authors simply plug in a fixed population average reference point into the model for all participants (except the stereotype experiment). This seems coarse. The authors note in the discussion that future research should evaluate how the RP is acquired/changed. Why is that not done in the present study? One could imagine that the RP changes as a function of experience, and in other words, is different later in the experiment than at the beginning (where it is more likely to reflect some population prior). Moreover, the Profiles are both female and male. Surely, people could apply a different RP to women and men.

We would like to clarify that we did not aim to test how the RPs change during our learning task but rather whether they were used over multiple experiments/ manipulations. The fact that the RPs used in our models are from a completely independent dataset but still are a significant contribution to the models is rather strong evidence that rough schemata, i.e., aggregated information about similar people, are an important part of (social) learning. Suggesting its change over time as a point of interest for future research is, in our eyes, a logical continuity of the initial establishment of its use.

*To still expand upon the RPs we have decided to add participants' self-ratings as RPs because using oneself as a reference is the easiest starting point to learning about strangers. Moreover, with lack of experience with other groups a consistent and reliable RPs would be to use one's own self-ratings on traits. For all models that use the population RPs we added models that use the participants' self-ratings as RPs and fitted these models on all experiments together with the original models (**Supplementary Figure 10**). Results were clear, for all experiments the models using self-ratings were worse than those using the population RPs indicating that participants had a better population average RPs available than using their own ratings.*

In a similar vein to limiting the current scope of the manuscript, we have decided to first find evidence for the use of both knowledge structures in this manuscript, hoping to present a more concrete modelling framework that can be expanded upon in later studies. Since winning models across experiments indicate the consistent use of RPs, we presently deem this enough evidence for their use across learning tasks.

We have created a section dedicated to the reference points in the methods. In this section we explain the origins and uses of the reference points in greater detail. The results from model comparison for the

models using self-ratings as their RP have been added in a separate section of the results under 'Additional Analysis'.

Reference Points

We defined reference points as “an average person (of a group) one has in mind” and formalized them as the average of a group of independent self-ratings (i.e., individual trait ratings calculated from an independent sample). RPs were used as points of comparison during learning by participants in all experiments except for Experiment 5. We surmise the absence of RPs in Experiment 5 to be related to the smaller scale of the answer options (i.e., 1-5 compared to 1-8 for the other experiments) which decreases their usability.

The most striking use of the RP was exhibited in Experiment 4, where participants were tasked with learning about people from an out-group (i.e., fashion models). During this learning task they used a different (stereotypical) RP than the one they had been using for the other experiments where they learned about people from an in-group (i.e., students). Interestingly, the best performing models indicated participants would be better off using Model 1 [No Learning]. Additional analysis that compared the fashion model profiles and the student profiles from experiment one showed higher correlations among the fashion model profiles and the standard population RP. This indicates that one could potentially estimate the average fashion model accurately with only the population RP available (**Supplementary Figure 8**). Notably, participants were persistent in their use of a learning model scaled by fine granularity knowledge and the stereotypical reference point. We interpret this finding as a case of “stereotypical reference points” in line with Jussim et al., (1995) who define stereotypes as cognitive categories that people use when thinking about groups and about individuals from those groups. It is widely accepted that stereotypic expectancies can guide learning (Hamilton, Sherman, and Ruvolo, 1990), as exemplified in Experiment 4. ...

Self-Ratings as Reference Points

With lack of experience with other groups a consistent and reliable reference point would be to use one's own self-ratings on traits. For all models that use the population RP we added models that use the participants' self-ratings as RP (**Supplementary Figure 10**) and fitted these models on all experiments. For all experiments the models using self-ratings were worse than those using the population RP, indicating that participants relied on a more accurate population RP than on their own self-ratings.

Supplementary Figure 10. Model comparison of all models (see **Figure 2** and **Figure 5**) with additional models that use the participants' self-ratings as reference point (RP). Models that use self-ratings as their RP are always a little worse than those that use the population average RP.

4.2. I understand the model description as that the model is not learning about the different Profiles, but only about personality traits (pooled across Profiles). This seems very odd -surely learning is about individuals?

Yes, learning is about individual participants!

Each profile consists of the self-ratings of an individual person coupled with sparse personal information (name, age, time living on their own). Participants are expected to learn about four (experiments 1-4) or five (experiment 5) of such profiles per experiment. Participants learn about these individuals and our models account for learning about multiple individuals. The assumption we make is that participants use the same strategy (i.e., model) throughout the experiment across profiles.

*This has been clarified in the task description in the methods. Changes highlighted with **bold**.*

Task

All five experiments shared the same structure, but differed in their content (i.e., words and profiles). The first experiment is described below. Any differences between Experiment 1 and the following experiments are detailed in the section “differences between experiments.” The main task in the experiments was a social learning task followed by (self-) rating tasks.

Participants performed a social learning task about different trait profiles, where a profile consisted of sparse information about a person (name, age, time since starting university studies and time spent living by themselves) and self-ratings on a selection of traits selected to resemble items found in the

Big-Five (see **Supplementary Tables 1-3**). The information and self-ratings from these profiles were selected from an independent previous study (Korn et al., 2012), with fictitious and randomly assigned common names. Instructions for the whole experiment were given orally and presented on the screen at the start of the experiment. Participants were given the opportunity to ask questions after the instructions and could start the experiment themselves through a button press. **The learning task consisted of four runs, i.e., one run per profile. Participants learned about a different profile in each run** (see **Figure 1A**). As described above, the person of the profile in question was briefly introduced at the start of the run. Two of the profiles were given female names and two of the profiles were given male names

5. Minor.

Was learning worse in Exp.4 than in the other experiments? The PE-trial correlation coefficient seems lower.

No, learning was not worse in experiment 4. The correlation coefficients/ effect sizes for prediction error decreases over time in all experiments are approximately -.50. We have updated Figure 3 and now also include the numerical correlation coefficients and the slope of the correlation. The correlation coefficients for experiments 1 to 5 are very similar (experiment 1: -.525, experiment 2: -.497, experiment 3: -.631, experiment 4: -.551, experiment 5: -.518). Moreover, simulations from the winning model have been added to this figure.

Figure 3 has been added in this document on page 12.

6. The authors report that Model 1 (no learning) was theoretically best in Exp.4. This is not interpreted or explained. Why would this be the case? Why wasn't this evaluated a priori?

*Looking at the optimal parameter space plots (**Supplementary Figures 11-16**) indeed shows that the optimal space (i.e., lowest error) is achieved by model 1. This indicates that when using an effective strategy one does not need to learn about the fashion models but can simply use a transformed population RP. We conjecture this could be true when fashion models would have self-ratings very similar to the population and/or among each other (i.e., the population RP explains large parts of the variance, or learning about one fashion model is enough to know about all). To check how related profiles and the population average were we calculated pairwise correlations among profiles and population RP for both experiments 1 and 4 (**Supplementary Figure 8**). Experiment 1 was chosen as comparison because it comprised of the same items as experiment 4. When comparing the correlations between experiment 1 and 4 the coefficients seem to be slightly higher between the profiles and population RP for experiment 4 (bottom rows), giving some evidence for the best fit of model 1 [no learning]. Interestingly, this indicates that participants would be better off using just the population RP for Experiment 4, the fact that this is not reflected in participants' strategies indicates the persisting nature of the stereotypes.*

*These results have been updated in the results section of experiment 4 and the reference point section in the discussion (changes highlighted with **bold**).*

Experiment 4: Fashion Models

... Model comparison confirmed our preregistered hypothesis, both fixed- and random-effects analyses indicated Model 8 [*Fine granularity & Stereotype RP*] as the winning model (**Figure 5C & Supplementary**

Figure 2C), suggesting that participants used a fine-grained representation of the personality structure together with stereotypical reference points. **Surprisingly, the best performing model indicated that Model 1 [No Learning] was the best strategy, potentially because of the lower variance between profiles (Supplementary Figure 8). ...**

Reference Points

... The most striking use of the RP was exhibited in Experiment 4, where participants were tasked with learning about people from an out-group (i.e., fashion models). During this learning task they used a different (stereotypical) RP than the one they had been using for the other experiments where they learned about people from an in-group (i.e., students). **Interestingly, the best performing models indicated participants would be better off using Model 1 [No Learning]. Additional analysis that compared the fashion model profiles and the student profiles from experiment one showed higher correlations among the fashion model profiles and the fashion model profiles and the standard population RP. This indicates that one could potentially estimate the average fashion model accurately with only the population RP available (Supplementary Figure 8). Notably, participants were persistent in their use of a learning model scaled by fine granularity knowledge and the stereotypical reference point. ...**

Supplementary Figure 8. Pairwise correlations of profiles and population averages for experiment 1 and 4. The correlation coefficients between profiles and the RP seem to be higher for experiment 4 than experiment 1. This additional analysis was performed to understand why the best performing model for experiment 4 (**Figure 5G**) indicated that the No Learning model was best. One potential explanation is that the fashion model profiles are more correlated than the profiles from other studies. Thus needing less information to do the task efficiently. Interestingly enough, participants still seemed to use a complex strategy [Model 5] during this task.

Reviewer #3 (Remarks to the Author):

Review for “How do humans learn about other people?” by Frolichs et al. The authors tested how humans learn about the multi-dimensional personality of others. They found that participants use prior ideas that they computational formalize into two ideas: granularity and reference point. To do so, the authors analyzed 5 experiments, where some experiment conditions were changed to test the robustness of the results. The paper addresses an original, clever and timely question and is well written. They also compare the best fitting model to optimal behavior (which is important and not frequently done in the field).

Overall the paper is very solid methodologically (several experiments, including pre-registration). The paper’s claims rely heavily on computational modeling and parameters analysis. I was very happy to see that these results were perfectly backed-up by model and parameter recovery. I am quite positive about this study. I only have few suggestions.

We thank the reviewer for their positive evaluation of the manuscript and insightful suggestions, which led to significant improvements in our original submission. We have taken to heart the reviewer’s suggestion to expand the model-free analysis of the data.

Point by point reply

1. I am sorry I will not give a more precise suggestion, but I feel that the behavioral results could be illustrated more in detail. In the current version of the manuscript the authors essentially only show the median abs PE across trial. I wonder whether there is more in the data to show.

*We agree that previous information in this regard has been sparse, in line with previous reviewer comments we have expanded the analyses on these behavioural measures. Results confirm our previous analyses and indicate participants’ use of PEs throughout the experiments (**Figure 4**).*

*An important additional analysis we performed was to include model simulations for all models. We have updated **Figure 3** to be more informative and included the winning model simulations. For a detailed response please refer to comment 4 from reviewer 1.*

Please also consult our response to point 4 from reviewer 1 for a detailed description of the model-free analysis. Explanations of each of these procedures have been added in the Methods section under the header ‘Model-Free Behavioural Analysis’ (see below). In specific, Figures 3 and 4 have been added or updated to support these explanations.

Model-Free Behavioural Analysis

To test whether the PEs have a downward trend over time (an indication of learning), we calculated the average of the absolute PEs per profile per trial for all participants. Profile data were then averaged into one vector (with length being the number of trials) of the average absolute PEs over time per experiment. To determine whether there was a significant decrease of PEs over time, we calculated the Pearson correlation coefficient on these absolute average PEs and the corresponding trial number. A negative Pearson correlation indicates a decrease in the absolute PEs over the trials. The same procedure was applied to data that was simulated based on the winning (i.e., best fitting) model for that experiment.

Moreover, for each experiment separately, we calculated a GLM that explained the accuracy per trial with three regressors. Each regressor was initially modelled as a separate regression and consisted of an integral

part of our computational models. In brief, the regressors were the following: 1) total number of previous trials (this assesses the decrease of the PEs over time and thus learning in standard reinforcement learning), 2) total number of previous trials within the factor to which the current item belongs (investigates the behaviour captured by the coarse granularity models), and 3) the summed absolute correlations of the previous items with the current item (this assesses the fine granularity models where the information content of the previous items is weighted by their correlation to the current item). In a second-level analysis, participants' individual parameter estimates were subjected to a one-sided one-sample t-test to test if the slope was significantly different from zero in the negative direction (indicating a decrease of the absolute PE over time).

2. On a similar note, it would also actually be informative to show the model simulations of the relevant variables. What is the feature that is not explained by the “losing” models? The simulations will nicely complement the BIC graphs (that are not really the more informative to understand the models' behavior).

As mentioned in the previous comment, we have included these additional analyses in response to earlier comments already. Results from these analyses indicate that participants seem to consistently use the fine granularity (in Experiment 1-4) but, further confirming results from model comparison, also seem to use the coarse granularity in experiments 2 and 3.

To answer the reviewers comment in brief, we performed a GLM that predicts the accuracy of responses based on three regressors. Each explaining an integral part of our models i.e., 1) decrease in PEs and stands for the use of standard reinforcement learning, 2) previous items in the current factor investigates coarse granularity, 3) summed correlations of the previous items indicates fine granularity.

For a more detailed response please refer to comment 1 by reviewer 1 as well as Figure 1 (pages 1-2 of this document).

In the manuscript, additions have been made to the following sections: in the model free sections of each experiment and in the methods. Please also consult point 1 by reviewer 1 for a more detailed answer to this question as well as text copied from the manuscript.

3. Finally, I believe that some citations are missing. First, the citations concerning observational learning should be updated in the light of recent work (Najar et al. Plos Biology, 2020). Reference point dependence is new in social learning, but not in “private” reinforcement learning (see Palmintieri et al. Nature Comms, 2015).

These omissions have been corrected!

References

Bonaccorso, G. (2017). *Machine learning algorithms*. Packt Publishing Ltd.

Bodenhausen, G. V. (1990). Stereotypes as judgmental heuristics: Evidence of circadian variations in discrimination. *Psychological Science, 1*(5), 319-322.

Stolier, R. M., Hehman, E., & Freeman, J. B. (2020). Trait knowledge forms a common structure across social cognition. *Nature human behaviour, 4*(4), 361-371.

Thornton, M. A., Weaverdyck, M. E., Mildner, J. N., & Tamir, D. I. (2019). People represent their own mental states more distinctly than those of others. *Nature communications, 10*(1), 1-9.

Thornton, M. A., & Tamir, D. I. (2021). Six dimensions describe action understanding: The ACT-FASTaxonomy. *Journal of Personality and Social Psychology*.

REVIEWER COMMENTS

Reviewer #1 (Remarks to the Author):

The revised manuscript from Frolichs and colleagues has dealt with several of the issues that I raised in the previous round of review. I still believe that the paper makes interesting points, however I still have one major issue.

Communication is still a major problem. And honestly, I'm now wondering whether I fully understand what is being reported, which makes me slightly uncertain of my evaluation of the scientific merit of the work. In addition to this, the organization of the paper makes it difficult to read – in order to evaluate results from a single experiment it is necessary to look through data that are presented across multiple figures. I still really like the experiments and results, but honestly, I think a major organizational overhaul should be done to make this work accessible to the broad audience it deserves.

The figures are particularly problematic, here are questions and comments relating to each figure:

Figure 2: what is being plotted here? why Bayes factors compared to model 2? Are you sure this isn't BIC – numbers look more consistent with BIC? Without finding the exact section of the main text that describes panel B, it is impossible to understand what it is, or how it differs from A. In terms of overall organization, why is this figure before the description of the task data (ie. learning curves, regression coefficients to capture it)? Why is there a giant panel used to make a legend when there is space on figure to label the models directly?

Figure 3: X and Y axes should be labeled. What is model 8? There were only 5 introduced. What experiment are the data from? Presumably experiment 1—but since all 5 experiments have already been introduced, it is not totally clear. Also there are multiple model 5s? Why does the title say simulated – the black line is actual participant data, isn't it? Why do the trial-to-trial jumps in data exceed the SEM? Presumably bc all subjects saw exact same sequence of trials, and so one of the major sources of variability is correlated across participants. In this case, I think it would be useful to construct bins of multiple trials to show performance, rather than to plot the single trial average.

Figure 4: x labels should be provided. Description of regressors/coefficients in caption is totally unclear.

Figure 5: This figure is almost completely uninterpretable without additional information. First of all, model labels on the figure would be helpful. However, as noted above, the overall organization of the paper would be easier to follow if figures were organized according to experiments rather than data type.

In addition to these issues related to the figures, I have the following remaining points:

For exp 1, was parameterization of models that produced most accurate predictions similar to the parameterization that provided best fit to participants? This seems to be an important piece of information that is not included.

A better description of the regression coefficients in the results section would be helpful.

Why are learning curves and posterior predictive checks only shown for experiment 1? As suggested above, this information would be useful for each experiment.

For experiment 3, the best fitting model used coarse representations, but model-free coefficients indicated that participants used a mixture of coarse and fine representations. Do simulated data from

coarse representation model reproduce this result? Stated differently, are participants doing something that is different from any of the models fit – that combines aspects of both and fine for this experiment? In general, it would be useful to have this sort of posterior predictive check on model simulations for each experiment. As noted above, I found it very difficult to evaluate results from individual experiments as I needed to look through multiple figures for relevant information. I think it would be much easier on reader if the figures were organized by study, using a consistent format for data presentation for each study.

Reviewer #2 (Remarks to the Author):

The authors have addressed some of my issues, but have not really engaged with others.

Conceptual. I still believe that calling the task/model “reinforcement learning” is a misnomer. The task – passively predicting a variable, and receiving exact feedback, has very little to do with the “reinforcement learning problem” in the Sutton & Barto sense. Using a Rescorla Wagner model does not equal reinforcement learning in my opinion. In my view, calling it a “learning task” or “prediction learning task” would be more accurate.

What I meant with a “regression problem” was that the task is a supervised learning problem (not RL). The participants receive exact (supervised) feedback. I do not see how this is different from other supervised learning tasks (eg vision). As the authors point out, supervised learning and RL are different ML domains.

This is a minor point, but in my opinion worth emphasizing.

Limitations of the experimental paradigm. The authors are providing a very detailed analysis of their task, as well as several replications. Still, I have a hard time seeing what real-life situations/social behavior this tasks model.

When do people make explicit predictions about another person’s personality traits, and this person provides explicit, numeric accuracy feedback? Almost never I would think. I of course appreciate that all experimental tasks are “models”, but as far as a task is meant to capture something “social”, external validity is an issue.

In other words, I am not sure the results of the paper teach us anything more about social learning and cognition. I was somewhat disappointed that the authors had not made any efforts in establishing the relevance of their experimental task and computational mechanisms for social behavior more generally by showing generalization to (some) task with clearer external validity.

All in all, this paper (in both original and revised form) provides very detailed and rigorous analyses of a specific task. These results are surely of interest, but I wonder whether they primarily represent a technical advance (showing how trait correlations and reference points can influence a type of statistical learning), rather than illuminating human social cognition/behavior.

I support publication but would prefer if the authors clearly clarified the limitations (rather than overselling) of their results and approach.

Reviewer #3 (Remarks to the Author):

The authors successfully addressed my concerns.

Reviewer #4 (Remarks to the Author):

How do humans learn about other people? Incorporating social knowledge structures into reinforcement learning.

Frolich's KMM, Rosenblau G, Korn CW

The authors present a manuscript in which they investigate mechanisms of social learning in a series of rigorous studies. Specifically, they were interested in building on the current computational social learning literature by incorporating knowledge about specific personality traits (granularity) and about representations of average individual (i.e., reference points) in addition to standard reinforcement learning mechanisms. The author formalized a host of computational models to implement these two types of knowledge structures in a series of tasks assessing trait learning. Overall, results indicate that computational models incorporating granularity and reference points to different degrees into social learning fit participants' data better in general than models positing no learning.

In general, I appreciate the authors on the rigor of their experiments and the novel contributions of aspects of their modeling approach. I did not review the initial submission of this paper, but much like Reviewer 2, I like the incorporation of correlations between traits into the models. However, I do have some concerns and questions, particularly regarding some aspects of the modeling.

1. It seems as though in response to one of the prior reviewers of the paper who asked for some univariate analyses (i.e., regressions), the authors conducted a number GLMs. However, it also appears that the authors have placed these results throughout the paper in the context of 'model-free' analyses. I would strongly caution the authors against using this nomenclature for regressions/correlations because while in theory they are not computationally based approaches, regressions require models that predict outcomes. Moreover, in the reinforcement learning literature, 'model-free' learning connotes learning that is updated incrementally on a trial-by-trial basis without existing knowledge of the environment, whereas 'model-based' learning incorporates knowledge of the environment. As an example, see a paper by Daw and colleagues (2011, Neuron). I am also a little confused because the authors have a very nice paragraph in the 'scope of models' section of their discussion regarding how their models fit within the model-free vs. model-based distinction, yet their classification of correlations and regressions as model-free seems to not reflect this understanding.
2. I am wondering if the authors can explain something about the modeling. On p10 of the manuscript it is stated that a max of 3 free parameters are implemented: 'all RL models have alpha (learning rate)...combination models make use of gamma (weighting) which determines how much each of two concepts is used...[and] finally, the starting value determines the value at which the model initializes'. This is unclear to me. Is the 'starting value' referring to a separate parameter in the model? Or is it referring to the fact that the starting value for learning rate and weighting parameters are determined by the model and not arbitrarily initialized at, say 0.5 and then allowed to update? If the latter, I wouldn't say that this is a third free parameter in the model. I am also asking because I do not see this 'starting value' free parameter denoted in any of the model equations, and in on p41 in the 'model fitting and comparison' section, it is noted that 'the 'free parameters are all initialized at the average between the. Maximum and minimum bounds). So, where does the starting value free parameter fit in? More clarity on this point would be beneficial in order to truly understand the modeling approach.
3. IN Experiment 4 (Fashion Models), the authors note that Model 8 (fine granularity and stereotype RP) was the 'winning model', but that Model 1 (no learning) was the 'best model'. Can the authors please reconcile these statements? I apologize if I am missing something obvious, but the wording here is confusing in terms of understanding which model best captured how people were learning in this experiment.
4. This leads me to a more general point. I found it a bit difficult to follow and distill the main narrative in the manuscript here. I note that prior reviewers of the paper had similar comments from

the initial version; while I appreciate the extensive work that the authors did in responding to previous comments, I still feel as though the writing in the current version of the paper is convoluted in parts, particularly in the methods and results.

5. I appreciate Reviewer 2's comment about the experimental paradigm being limited in scope, but I do believe that the authors are measured enough in the 'scope of models' section of the discussion now, which mitigates this concern in my view, and in general, aside from the points I raised above that align with Reviewer 2, I think the authors have done a nice job in their response to Reviewer 2's other comments.

Reviewer #1 (Remarks to the Author):

The revised manuscript from Frolichs and colleagues has dealt with several of the issues that I raised in the previous round of review. I still believe that the paper makes interesting points, however I still have one major issue.

Communication is still a major problem. And honestly, I'm now wondering whether I fully understand what is being reported, which makes me slightly uncertain of my evaluation of the scientific merit of the work. In addition to this, the organization of the paper makes it difficult to read – in order to evaluate results from a single experiment it is necessary to look through data that are presented across multiple figures. I still really like the experiments and results, but honestly, I think a major organizational overhaul should be done to make this work accessible to the broad audience it deserves.

We thank the reviewer for their comments and agree that the manuscript and figures have become convoluted in our previous version. Aided by the reviewer's helpful comments, we have now addressed these issues throughout the manuscript, with a focus on reorganizing the figures and on minimizing redundancy.

The figures are particularly problematic, here are questions and comments relating to each figure:

We understand the confusion about the organization of the figures. Initially the figures all represented one type of analysis and to save space, they were combined for multiple experiments. In an effort to reduce confusion due to the convoluted figures, we have separated the figures to now summarize results for each experiment separately. In the results we have introduced subheadings to more clearly delineate what analysis is being discussed, and these subheadings correspond to the specific panel titles in the figure. To aid navigation of the manuscript we placed each figure directly below the results of each experiment.

Directly below, we have copied figure 2, which is the summary figure of experiment 1, the figures of the other 4 experiments are in exactly the same style. This means that if one understand figure 2 they will also understand the figures of experiments 2-5. The only figure that is slightly different is figure 5 (experiment 4) as explained in the manuscript and the figure caption we added three additional models for this experiment for a total of eight models.

The layout (but not the overall content) of the figures has changed. Below, we address each specific question and describe how we have applied these changes in the new figures.

Figure 2. Overview of the main analyses for experiment 1. Results indicate that participants used fine-grained correlation structures.

a) Model comparison results from participants' data using fixed-effects analysis. The fixed-effects analysis uses the difference between the summed BIC values for each participant and each model with those from the losing model (here Model 2 [Coarse Granularity (CG)]). This is the reason that the losing model is always zero.

For this experiment, Model 5 [Fine Granularity (FG) & Population Reference Point (RP)] is the best fitting model. This model uses the average population as a reference point and fine granularity for generalization.

b) Simulated data for the 'best performing model'. The best performing model indicates which of the models in the set can simulate the most optimal task performance. That is, the simulated data is fitted on the actual task instead of participants' responses (see Supplementary Figure 7 for a detailed explanation). These simulations use the same fixed-effects analysis as for panel A. However, since these are model simulations, the exact BIC values depend on the total number of simulations (here $n = 36$ simulation). This means that only relative differences between BICs are interpretable.

As for participants' data, Model 5 [Fine Granularity (FG) & Population Reference Point (RP)] also performed best when testing for the best performing model. This indicates that participants used the best strategy (of the strategies in our set).

c) A decrease of the prediction errors (PEs) over time (i.e., trials in the experimental runs) can be interpreted as learning because participants' estimates got closer to the feedback they received. Both plots display the average absolute PEs over time (black line, grey patch: SEM). We calculate a pairwise Pearson correlation between trial numbers and the mean absolute PEs to determine if the PEs decrease over time (least squares line, LSLine, red line).

Top) Participants' data shows a decrease in the PEs over time ($\rho: -.523$). **Bottom)** Simulated data from the best fitting model (Model 5) shows a similar decrease in PEs over time, indicating that the models 'learned' in a similar way to participants.

d) General linear model on core features of our models. Three separate regressions predicted the accuracy (i.e., the PE) per trial per participant. These regressors roughly corresponded to 1) Rescorla-Wagner RL, 2) the coarse models, and 3) the fine models. Detailed descriptions of the regressors are given in the results of experiment one and the methods section (*standard behavioural analysis*).

Each regressor is indicated by a different colour, individual data points are the parameter estimates per participant (summarized by the boxplots). Conclusions should be taken with caution since all three regressors were highly correlated (ρ between .76 - .92). [One-sided t-test; * indicates $p < 0.05$, ** indicates $p < 0.001$, no correction for multiple comparisons].

CG = coarse granularity, FG = fine granularity, BF = Bayes factors, RP = reference point, # = “number of”, PEs = prediction errors, SEM = standard error of the mean, RL = reinforcement learning.

Figure 2: what is being plotted here? why Bayes factors compared to model 2? Are you sure this isn't BIC – numbers look more consistent with BIC?

The reviewer is correct: The numbers are based on BIC. In specific, we sum the BIC values from all participants for each model. From these summed BIC values we then subtract the summed BIC value from the worst performing model (here ‘Coarse Granularity’). This means that the worst performing model always has a value of 0 in these plots.

This explanation has been included in the figure caption (see caption for Fig 2A above).

Without finding the exact section of the main text that describes panel B, it is impossible to understand what it is, or how it differs from A.

This panel shows simulations based on participant data. We named these simulations the “best performing models” since they elucidate which of the models offers the best strategy for this specific learning task i.e., which of these models is best at performing the task, independent of participant performance (see Supplementary Figure 7 for a detailed explanation of the best performing models). In terms of visualization, this is the same as “panel A” (summed BIC scores relative to the worst performing model in the set). Importantly, since these are model simulations the exact Bayes Factors (BFs) depend on the total number of simulations. This means that only relative differences between BICs are interpretable.

The description has been updated in the figure caption (caption 2B, above). Moreover, we have added clearer titles in the figures to avoid confusion.

In terms of overall organization, why is this figure before the description of the task data (ie. learning curves, regression coefficients to capture it)?

All primary information is now contained in one figure per experiment. We have decided to present the modelling results first since these are the most important outcomes of our experiments. Other analyses and learning curves function as support for these results.

Why is there a giant panel used to make a legend when there is space on figure to label the models directly?

Integrating all panels from one experiment into one figure has reduced the size of the legend considerably.

Figure 3: X and Y axes should be labeled.

Sorry. This has been updated in the panels of the new figures (Figure 2C).

What is model 8? There were only 5 introduced. What experiment are the data from? Presumably experiment 1—but since all 5 experiments have already been introduced, it is not totally clear. Also there are multiple model 5s?

This figure shows the mean absolute PE over the duration of an experiment, separately for each experiment (a-e), combined with the simulated data from the best fitting model for this particular experiment. Since each window is a different experiment and we fitted the same models on each experiment there are multiple “model 5”’s i.e., the winning model for each separate experiment. Experiment 4 had three additional models that used (stereotypic) fashion model estimations instead of the standard reference point. This means that the total number of models for experiment 4 was eight. Model 8 [Fine Granularity & Stereotypic RP], was the best fitting model and therefore used for the simulations.

To reduce this confusion we have placed each of these plots in a separate figure for each experiment. In this figure, we have separated participants’ data from simulated data (Figure 2C), added labels and a new legend. Moreover, the captions have been updated to be more informative.

Why does the title say simulated – the black line is actual participant data, isn’t it?

Yes, you are correct the black line is actual participant data. The “simulated” in the title pertained to the blue line, which is the simulated data from the winning model. The title refers to the model that was best for that specific experiment. We agree that this figure was difficult to understand and therefore have updated it to be clearer.

As mentioned above we have separated the plots with participant and simulated data, updated the labels, title, and legend (Figure 2C).

Why do the trial-to-trial jumps in data exceed the SEM? Presumably bc all subjects saw exact same sequence of trials, and so one of the major sources of variability is correlated across participants. In this case, I think it would be useful to construct bins of multiple trials to show performance, rather than to plot the single trial average.

We thank the reviewer for this astute observation, in working on this problem we realised we mislabelled the axes for these plots in a way that can explain this effect. Namely, instead of plotting the data as it was presented to participants (i.e., plotting the PEs over time) we ordered it per factor over time. This can explain these “peaks” in the data since all participants experienced these new factors at roughly the same time point (see figure below). This error has been corrected in the new figures. Results have not been altered by changing these labels. To show these effects to the reviewer we have added the figure below that shows the two original panels for experiment 1 and 2 on the left and the updated panels on the right. We have not included this figure in the manuscript but could do so if the reviewer seems fit.

Sorted versus unsorted PE's for experiments 1 and 2. The left panels (A, C) show the plots that were originally used for figure 3. In these panels we erroneously ordered the trials per factor (start of each factor is indicated by vertical coloured bars) which caused the peaks that exceed the SEM. Please note, due to missing answers peaks can be a little shifted. The panels on the right (B, D) are the absolute PEs over time as they originally should have been implemented. These panels have been implemented in the new figures. To save space results are shown only for experiment 1 and 2 but were similar for the other experiments.

Figure 4: x labels should be provided. Description of regressors/coefficients in caption is totally unclear.

The separate windows from this figure have been moved to a corresponding figure for each experiment. In these figures the caption has been updated and x-labels have been added.

Figure 5: This figure is almost completely uninterpretable without additional information. First of all, model labels on the figure would be helpful. However, as noted above, the overall organization of the paper would be easier to follow if figures were organized according to experiments rather than data

type.

We apologize for the unclarities. We have followed the reviewer's advice and created figures for each experiment separately. Thereby, the model labels are now contained in each figure. The new figure has been added directly after the reviewer's first question. All other experiments have figures with the same panels.

In addition to these issues related to the figures, I have the following remaining points:

For exp 1, was parameterization of models that produced most accurate predictions similar to the parameterization that provided best fit to participants? This seems to be an important piece of information that is not included.

This is indeed an important piece of information and therefore has been included. The short answer is: Yes, the parametrization overlapped. To save space in the manuscript, we opted to move the details to the supplementary materials. In Supplementary Figures 11-16, we show that the parameters for simulations and real data do indeed largely overlap for all experiments.

*In detail, we performed a grid-search over the whole parameter space for simulated models. That is, for each model we tested the model fit (sum of squared errors) for all possible combination of parameters (with 100 steps between the minimum and maximum for each specific parameter e.g., [0 1] for the learning rate). On these 2- or 3-dimensional plots of simulated fits we plotted the parameters that were fit on participants' data. Results indeed indicate a large overlap between simulated and actual parameters. See **Supplementary Figure 11** below.*

Supplementary Figure 11. Optimal parameter space with fitted parameters from participants in Experiment 1. There is a large overlap between optimal simulated SSE (red patch) and participants' best fitting parameters (grey dots).

Models 1, 2 & 3 (A, B & C) have two free parameters (*intercept* and *slope* for Model 1 and α and *start value* for Models 2 & 3). With only two parameters the whole parameter space can be displayed on a Cartesian plane. Models 3 & 5 (D & E) have three parameters which means their parameter space fills a “cube” (i.e., the three parameters result in three dimensional plots). In order to display these models in a similar manner to the other models, we chose a “slice” out of this cube where the best fit was achieved for the γ parameter (γ -value specified top left). In all plots, fitted parameter values per participant were plotted for the depicted model. White dots indicate participants for which the depicted model did not achieve the best fit. Grey diamonds indicate participants for which the depicted model did achieve the best fit out of all models tested.

A better description of the regression coefficients in the results section would be helpful.

A more detailed description of the regressors has been included in the results section of experiment 1 (page 9, lines: 271-285) and the methods (page 32, lines: 997-1012). For convenience, we copied this section below:

Experiment 1: Real Profiles & Wide traits

... Furthermore, we conducted a general linear model (GLM) analysis that consisted of three separate regressors to predict participants’ answer accuracy (i.e., higher accuracy means lower PE). Each of the regressors represented a substantial part of the models. Regressor 1) captures learning in the standard Rescorla-Wagner model by tracking the total number of previous trials for each item (i.e., if participants are learning one should see a decrease in PEs over trials). Regressor 2) captures the coarse granularity by tracking the total number of previous trials within a factor for each item (i.e., if participants learn based on each factor, one expects to see a decrease in PEs over trials within this factor). Regressor 3) assesses the fine granularity by computing the summed absolute correlations of the previous items with the current item (i.e., assumes the correlation is the information density of an item to the current item, the sum of all previous items thus predicts the decrease in PE). Results from this GLM indicated the use of the fine-grained correlation structures (i.e., regressor three) (**Figure 2D**). ...

Statistical Behavioural Analysis

... Moreover, for each experiment separately, we calculated a GLM that explained the accuracy per trial (i.e., the prediction error) with three regressors. Each regressor was initially tested in a separate regression and captured an integral part of our computational models. In brief, the regressors were the following:

Regressors 1: total number of previous trials seen for each item, (this assesses the relationship between the decrease of the PEs and the number of items seen previously and thus learning in the standard Rescorla-Wagner model)

Regressors 2: total number of previous trials that are from the same factor as the current item (this assesses the relationship between the decrease of PEs and the number of items encountered from a specific factor and thus investigates the behaviour captured by the coarse granularity models)

Regressors 3: the summed absolute correlations of the previous items with the current item (this assesses the fine granularity models where the information content of all the previous items is weighted by their correlation to the current item).

Why are learning curves and posterior predictive checks only shown for experiment 1? As suggested above, this information would be useful for each experiment.

Both learning curves and posterior predictive checks have been shown for all experiments (see explanation to figure 3 above) but should be much easier to find in the new figures. All learning curves and posterior predictive checks corresponded with our hypotheses.

For experiment 3, the best fitting model used coarse representations, but model-free coefficients indicated that participants used a mixture of coarse and fine representations. Do simulated data from coarse representation model reproduce this result?

Stated differently, are participants doing something that is different from any of the models fit – that combines aspects of both and fine for this experiment? In general, it would be useful to have this sort of posterior predictive check on model simulations for each experiment.

*We assume the reviewer meant to say that **experiment 2** (instead of experiment 3) shows the use of coarse representations and that the model-free coefficients indicate a mixture of coarse and fine representations.*

In brief, yes simulated data from coarse models reproduce these results.

But we would like to caution against drawing overly hard conclusions from these results. The coarse- and fine-grained knowledge structures can be thought of as the two extremes of a continuum on which to represent personality (in a similarity matrix way). Fine-grained models subsume parts of the coarse-grained models and vice versa. Because of this shared background they are inherently correlated to some degree—both conceptually as well as mathematically (as mentioned in the figure caption). This makes results from these standard statistical analyses a little harder to interpret as such, which is why we deem our models to be very important since they can better differentiate between fine- and coarse-grained similarity structures. Crucially, results from the confusion matrix (Supplementary Figure 3) are robust indicating that our current models are able to distinguish between these extremes. Of course, participants' strategies might lie somewhere on this continuum as well. Therefore we aim to investigate the continuum between these extremes in future work.

In detail, we have run extra simulations to test these results (see figure below).

For each learning model we separately simulated 200 runs for a dataset with four profiles with 60 items each (240 items total). Each simulation was initialized with random parameter settings. To calculate the GLM we created three separate regressors to predict the answer accuracy (i.e., PEs). In brief, these are the same regressors as those used in the manuscript: regressor 1: captures Rescorla-Wagner learning, regressor 2: coarse granularity, and regressor 3: fine granularity. Please refer to our response to the previous question for a detailed description of the regressors.

As can be seen in panel B, when data is simulated with Model 3 all three regressors are significant as found in experiment 2. Moreover, for experiment three (winning model 5, panel D) results are the exact same, but other experiments do not correspond as well. But as mentioned in the next question these regressors are correlated and results should thus be interpreted with caution.

GLM fitted to simulations from the coarse and fine granular models. Panel B replicates results from experiment 2, all regressors significant for data from Model 3.

As noted above, I found it very difficult to evaluate results from individual experiments as I needed to look through multiple figures for relevant information. I think it would be much easier on reader if the figures were organized by study, using a consistent format for data presentation for each study.

Overall, we would like to thank the reviewer for raising the important issue of lacking clarity in our previous manuscript. We have improved our presentation of data in figures and result sections to improve clarity of all our previously reported analysis procedures and results.

Reviewer #2 (Remarks to the Author):

The authors have addressed some of my issues, but have not really engaged with others.

We would like to thank the reviewer for their comments on the manuscript. We admit that due to the conceptual nature of the reviewer's comments and differences in interpretation depending on the scientific subfield, we may have partly misunderstood or misrepresented the reviewer's perspective in our previous revision. We have tried to remedy this in the current revision by using unambiguous terms and by explaining their definition in the current study.

Conceptual. I still believe that calling the task/model “reinforcement learning” is a misnomer. The task – passively predicting a variable, and receiving exact feedback, has very little to do with the “reinforcement learning problem” in the Sutton & Barto sense. Using a Rescorla Wagner model does not equal reinforcement learning in my opinion. In my view, calling it a “learning task” or “prediction learning task” would be more accurate.

We understand the reviewer's concern about our broad usage of the term reinforcement learning. In the literature of cognitive science/ psychology, the categorization of Rescorla-Wagner models as subtype of Reinforcement Learning is largely accepted (please see these recent papers that all categorize RW as part of RL: Niv et al., 2015; Collins & Shenhav, 2022; Lockwood & Klein-Flügge, 2021; Chierchia et al., 2021). We do understand that for an audience not familiar with this specific literature, it can be a confusing usage of the term RL. Therefore, we have severely limited our use of the term RL, most noticeably in the title (i.e., we have replaced reinforcement learning with computational models). Furthermore, in the introduction we clearly state that we use the Rescorla-Wagner models and that these models, in the cognitive sciences literature, are categorized as part of RL. Everywhere else in the manuscript, we refer to the specific models as RW only.

Some examples of these changes have been copied below (bold parts highlight changes). Namely, the title (page 1), abstract (page 2, lines: 29-31, 36, & 39), and the introduction (page 3, lines 52-70).

How do humans learn about other people? Incorporating social knowledge structures into **computational models**

Abstract

... In this study, we specified and tested potential strategies that humans could employ for learning about others. Standard **Rescorla-Wagner (RW)** learning models only capture part of the learning process because they neglect inherent knowledge structures and omit previously acquired knowledge. ...

Introduction

... **Rescorla-Wagner (RW)** models, that **in the cognitive sciences fall under the wide umbrella term of reinforcement learning (RL)**, entail simple and robust algorithms that characterize dynamic learning processes across a wide range of (non-)social tasks. ...

What I meant with a “regression problem” was that the task is a supervised learning problem (not RL). The participants receive exact (supervised) feedback. I do not see how this is different from other supervised learning tasks (eg vision). As the authors point out, supervised learning and RL are different

ML domains. This is a minor point, but in my opinion worth emphasizing.

We feel that this comment is related to the global use of the term RL within this manuscript. As described above we have changed the overall tone of the manuscript from RL to RW hoping that this will change the expectations and perspective of the audience regarding the models and tasks used in this manuscript. We agree with the reviewer that the current task has a lot in common with a supervised learning problem.

In a similar vein to categorising Rescorla-Wagner models as part of RL, tasks similar to the one we used are commonly used in the cognitive science/ psychology literature to test whether and how individuals reduce prediction errors over time (Niv et al., 2015; Diaconescu et al., 2017; Lockwood et al., 2016; Zaki et al., 2016). Since we are interested in how humans learn about others' personalities over time, the RW models and the task needed to accompany these models are ideal for investigating such trial-by-trial learning.

It is, of course, important to communicate this properly to our readership. We have therefore emphasized this in the limitations section (page 27, line: 814-819).

Limitations

Studying social interactions in a well-controlled experimental setting forces limitations on a study (i.e., in the current experiment some limitations were necessary to establish the current modelling paradigm). First, even though we used RW learning models to capture participants' behaviour. The task, in which participants received direct and exact feedback, bears some resemblance to a supervised learning problem. Supervised models have thus far not been used to explain our task and could pose as interesting models for future research. ...

Limitations of the experimental paradigm. The authors are providing a very detailed analysis of their task, as well as several replications. Still, I have a hard time seeing what real-life situations/social behavior this tasks model.

When do people make explicit predictions about another person's personality traits, and this person provides explicit, numeric accuracy feedback? Almost never I would think. I of course appreciate that all experimental tasks are "models", but as far as a task is meant to capture something "social", external validity is an issue.

In other words, I am not sure the results of the paper teach us anything more about social learning and cognition. I was somewhat disappointed that the authors had not made any efforts in establishing the relevance of their experimental task and computational mechanisms for social behavior more generally by showing generalization to (some) task with clearer external validity.

We understand the reviewer's reservations given the external validity of our experiments and interpret their reservations as twofold. First, the use of numerical scales to represent the magnitude of possessing some personality trait. Second, the fact that participants receive explicit feedback about a person. We would like to address those two main points one after the other.

First, we detail our rationale for using numerical scales. We would like to argue that our task is akin to answering a questionnaire using a Likert scale. Participants are aware that these numbers represent magnitude in an (at least) ordinal fashion. These values are similar to commonplace magnitude representations that quantify social constructs. This is evident in the fact that these Likert scales are commonplace when expressing feelings, impressions of others in dating or work contexts as well as reviews on the internet (e.g., Amazon, Netflix, and Google Maps). For instance, we often rate another's' physical appearance on a number scale. Often, a single number on a numerical scale is used to capture a

person's attractiveness, despite the common knowledge that attractiveness is multifaceted (i.e., physical attributes, personality, and situation).

Second, we explain our rationale for why participants received explicit (numerical) feedback. We do not share the reviewer's view that humans never receive explicit numerical feedback. In fact, numerical ratings and rankings are commonly used to assess individuals' performance and traits (competence, diligence, approachability etc.) at school or in professional contexts particularly in hiring decisions. Even if not expressed in numbers, humans are often very verbal about how much a trait applies to them e.g., "I'm such a diligent person", "She's very smart", or "I am a bit of a slob". Akin to our previous argument, such statements can easily be transformed to numerical values e.g., when someone asks how diligent someone is, it is acceptable to answer "8 out of 10". Therefore, we do not think that the external validity is lacking.

We do agree with the reviewer that "real-life" feedback can be noisier sometimes e.g., if someone buys you a drink, how generous is this person actually? That is, the actual rating often depends on the social situation and one's own viewpoint on the situation (e.g., you don't like this person and therefore do not appreciate the gesture). For this task this does not pose a problem since participants can adjust their viewpoints to fit the explicit ratings. As far as modelling is concerned such noisier feedback would not cause any extra problems if the number of trials are sufficiently large.

As a final point, none of the participants indicated any difficulty in understanding the task or assigning number ratings to others' traits. This was expected since our lab has used a similar social learning paradigm successfully before (Korn et al., 2012).

We have included a limitations section in the discussion to highlight these reservations (page 27, lines: 813-826). This section is copied below:

Limitations

Studying social interactions in a well-controlled experimental setting forces limitations on a study (i.e., in the current experiment some limitations were necessary to establish the current modelling paradigm). First, even though we used RW learning models to capture participants' behaviour. The task, in which participants received direct and exact feedback, bears some resemblance to a supervised learning problem. Supervised models have thus far not been used to explain our task and could pose as interesting models for future research.

A related limitation is that the task and thus feedback was received in exact numbers. In a social situation these number would normally be substituted for a verbal description (e.g., "I am very generous") or an action (e.g., sharing resources shows ones generosity). When learning about others in most social settings people seldom receive exact or direct feedback. Precise number feedback as used in our task is more common in the work or school setting as part of performance evaluations. Future studies can generalize from precise numbers to less precise, verbal quantifications to resemble the most commonly used form of social evaluation. ...

All in all, this paper (in both original and revised form) provides very detailed and rigorous analyses of a specific task. These results are surely of interest, but I wonder whether they primarily represent a technical advance (showing how trait correlations and reference points can influence a type of statistical learning), rather than illuminating human social cognition/behavior.

Thank you for your positive evaluation of our manuscript. We agree that the main advances in this manuscript focus on the technical aspects of the models. Nonetheless, in our view, this manuscript illuminates an extremely interesting part of human cognition that – to our knowledge – has not been discussed in depth before, namely, that humans use rather complex knowledge structures to update their beliefs about others. We therefore believe that this manuscript can help move the field of social neuroscience / social decision making forward on two fronts, first, in applying this robust and replicable model space to other domains of (social) learning, second, in understanding how people represent personality as a conceptual space and how they use this to learn about others.

I support publication but would prefer if the authors clearly clarified the limitations (rather than overselling) of their results and approach.

We have added a limitations section in the discussion to make explicit the limitations of the current paradigm (page 27, lines: 813-835). Parts of this section are copied above in answer to the previous question. The second part is copied below:

... Furthermore, learning about others' personality is sometimes more directly related to actions e.g., approaching someone who seems friendly. Expanding the task to include actions based on feedback about someone's personality (e.g., a cooperative foraging task) could therefore further heighten the external validity.

The total amount of trait items used is on the small side i.e., for experiment 5 we used 50 items (10 per factor) but there exist questionnaires with 120 or even 300 items. Expanding the total number of items could offer a more precise look at learning about personality traits. Specifically it would offer a more rigorous look at the continuum between the coarse and fine-grained extremes.

Reviewer #3 (Remarks to the Author):

The authors successfully addressed my concerns.

Thank you.

Reviewer #4 (Remarks to the Author):

How do humans learn about other people? Incorporating social knowledge structures into reinforcement learning. Frolichs KMM, Rosenblau G, Korn CW

The authors present a manuscript in which they investigate mechanisms of social learning in a series of rigorous studies. Specifically, they were interested in building on the current computational social learning literature by incorporating knowledge about specific personality traits (granularity) and about representations of average individual (i.e., reference points) in addition to standard reinforcement learning mechanisms. The author formalized a host of computational models to implement these two types of knowledge structures in a series of tasks assessing trait learning. Overall, results indicate that computational models incorporating granularity and reference points to different degrees into social learning fit participants' data better in general than models positing no learning.

In general, I appreciate the authors on the rigor of their experiments and the novel contributions of aspects of their modeling approach. I did not review the initial submission of this paper, but much like Reviewer 2, I like the incorporation of correlations between traits into the models. However, I do have some concerns and questions, particularly regarding some aspects of the modeling.

We would like to thank the reviewer for reviewing our manuscript. We appreciate the fresh look at our manuscript that revealed some parts that were still unclear to a first-time reader.

1. It seems as though in response to one of the prior reviewers of the paper who asked for some univariate analyses (i.e., regressions), the authors conducted a number GLMs. However, it also appears that the authors have placed these results throughout the paper in the context of 'model-free' analyses. I would strongly caution the authors against using this nomenclature for regressions/correlations because while in theory they are not computationally based approaches, regressions require models that predict outcomes. Moreover, in the reinforcement learning literature, 'model-free' learning connotes learning that is updated incrementally on a trial-by-trial basis without existing knowledge of the environment, whereas 'model-based' learning incorporates knowledge of the environment. As an example, see a paper by Daw and colleagues (2011, Neuron). I am also a little confused because the authors have a very nice paragraph in the 'scope of models' section of their discussion regarding how their models fit within the model-free vs. model-based distinction, yet their classification of correlations and regressions as model-free seems to not reflect this understanding.

We completely agree! Our initial idea of naming the GLM's "model-free" was indeed to indicate the difference from the actual modelling that our manuscript focusses on. We agree that it is confusing to use this more general meaning of the term while also using the more specific meaning of "model-free" as employed in the reinforcement learning literature. We therefore opted to rename these analyses throughout the manuscript. When mentioning these analyses in general we opted for "statistical analyses", when mentioning them individually we simply used their names (i.e., regression and correlation analysis) instead. This means that model-free is now only used in the reinforcement learning term in the discussion.

We have copied some examples below (changes highlighted in bold) from the Introduction (page 5, lines: 130-131) and Experiment 1 (page 10, lines: 271-275). The part where we describe model-free in the Reinforcement Learning sense has not been altered: Discussion (pages 26-27, lines: 800-811), we have also copied this section below.

Introduction

... Crucially, several **standard analyses (e.g., regressions & correlations)** were conducted to support the model-based analyses. ...

Experiment 1: Real Profiles & Wide traits

Statistical analyses

Analyses that were not based on computational models indicated that participants were learning: absolute PEs decreased over trials (**Figure 2C, top**). A pairwise Pearson correlation between trial number and the mean of the absolute PE over all participants showed a large negative correlation, $r(58) = -.523$, $p < .001$. Furthermore, **we conducted a general linear model (GLM) analysis that consisted of three separate regressors to predict participants' answer accuracy (i.e., higher accuracy means lower PE)**. ...

Scope of models

... In a general modelling framework we argue that our models relate to differences on a continuum between model-free (MF) and model-based (MB) RL (Dolan & Dayan, 2013). The details captured by granularity reveal complexities that MF-RL does not capture: MF learns a single value for any given stimulus in a given learning situation. Likewise, our models with coarse granularity learn a summary value for each of the Big-Five factors. In their simpler form, we thus think of our models using coarse granularity to be more similar to MF-RL rather than to employ full MB-RL (i.e., coarse granularity functions as a look-up chart of traits). The more complex models with fine granularity can be viewed as more like MB-RL as they incorporate a full representation of the traits and the similarities between them. This split—akin to differences between MB-RL and MF-RL—suggests that our models can distinguish between more costly optimal models and more efficient heuristic models, similar to models on optimal versus heuristic decision-making. (Korn & Bach, 2019; Korn & Bach, 2018).

2. I am wondering if the authors can explain something about the modeling. On p10 of the manuscript it is stated that a max of 3 free parameters are implemented: 'all RL models have alpha (learning rate)...combination models make use of gamma (weighting) which determines how much each of two concepts is used...[and] finally, the starting value determines the value at which the model initializes'. This is unclear to me. Is the 'starting value' referring to a separate parameter in the model? Or is it referring to the fact that the starting value for learning rate and weighting parameters are determined by the model and not arbitrarily initialized at, say 0.5 and then allowed to update? If the latter, I wouldn't say that this is a third free parameter in the model.

The starting value is used to initialize the first 'guess' of the model when encountering a new profile or factor. When there is no such starting value we would initialize the first value at the midpoint of the scale (e.g., 4.5 for the scale 1-8). The benefits of the free parameter over such a static value are twofold: first, it allows for a more accurate starting value, second, it permits the model to initialize at a different value for each participant. That is, across all models, we found the starting value to allow for a better model fit than when initialized at a static value.

We agree that this has not been properly conveyed and have therefore updated the computational models sections in the methods section to state this more clearly.

Both sections are copied below (changes highlighted in bold) (page 34, lines: 1069-1071 & lines: 1090-1093).

Methods

Model 2: Coarse Granularity

... with a second free parameter; starting value (bounded [1 8]), determining the value at which each factor will be initialized. **That is, the starting value determines the first ‘guess’ the model makes when a new factor or profile is presented (i.e., the value at $P_{(0,F)}$).**

Model 4: Fine Granularity

... Model 4 also makes use of **the free parameter** starting value. **Because this model updates all items for every step of learning the starting value is not just initialized for the first value but rather all items. This ensures that the model can updates all items from the onset of learning.**

I am also asking because I do not see this ‘starting value’ free parameter denoted in any of the model equations, and in on p41 in the ‘model fitting and comparison’ section, it is noted that ‘the’ free parameters are all initialized at the average between the Maximum and minimum bounds). So, where does the starting value free parameter fit in? More clarity on this point would be beneficial in order to truly understand the modeling approach.

As explained above, the starting value is only used to initialize the first ‘guess’ of the model. To take Model 2 as an example:

$$P_{(t+1,F)} = P_{(t,F)} + \alpha * PE$$

For each new factor (and each new profile) the model has to initialize a new starting value. This would be the value at $P_{(0,F)}$.

Figure 1 and its caption have been updated to state the purpose of the starting value more clearly.

3. In Experiment 4 (Fashion Models), the authors note that Model 8 (fine granularity and stereotype RP) was the ‘winning model’, but that Model 1 (no learning) was the ‘best model’. Can the authors please reconcile these statements? I apologize if I am missing something obvious, but the wording here is confusing in terms of understanding which model best captured how people were learning in this experiment.

We fit our models on two different sets of data. Primarily on participants’ responses, this allows us to determine which of the strategies tested in the model space best describes participants’ answers. In another analysis we fit the models on the feedback given in the task (i.e., the strategies are compared to the veridical feedback of the task participants performed). This allows us to test which model or strategy would have been best at performing the task as a participant. Please refer to supplementary figure 7 (added below).

Supplementary Figure 7. The rationale of the best performing models (in the set).

Data in this figure are selected from two participants (top and bottom row) from Experiment 2. For the axis, data have been sorted according to time in experiment and according to factor i.e., items have been sorted based on when they were encountered within the factors agreeableness and conscientiousness. The best performing models are encountered in **Figure 2C** and **Figure 5E-H**

The left panels (A & D) show the profile answers in grey (this is the feedback participants received) and participant answers are indicated in green. During regular model fitting, models are fitted on these participant answers (rightmost panels C & F), here we see that the top participant's answers are best explained by model 5 (orange) and the answers by the bottom participant by model 1 (blue).

The middle column (B & E) shows the 'best performing models', these are models fitted on profile answers (grey line) i.e., models are fitted as if they have full information about the profiles, like participants had. For both profiles model 5 is the best performing model in our set of models.

Comparing the best fitting models with the models that best explained participant data helps us in understanding whether participants used an effective strategy or not.

Normally, one would expect that participants use the best possible strategy. This means that the best fitting model (on participants' data) and the best performing model (on the task) overlap. This is the case for experiments 1, 3, and 5. In experiment 4 there is no such overlap indicating, in this case, that participants used a strategy that was too complicated. We investigated this previously and found the most likely explanation to be that the fashion model profiles were more correlated with the population reference point (i.e., the average student ratings) than the profiles from other studies (see Supplementary Figure 8). This means that the best strategy for this specific experiment can only rely on these average ratings to do this task optimally. This does not mean that the other models cannot perform well but rather that Model 1 [No Learning] is the simplest model that can perform the task well.

The fact that we find participant use a more sophisticated strategy, similarity learning, shows that similarity learning is a default strategy for participants and that they were not aware of differences in this item set that would have allowed them to shift to a simpler strategy.

We have attempted to state this more clearly in the results section of experiment 4 (page 18, lines: 522-534). Copied below (changes highlighted in bold).

Experiment 4: Fashion Models

Simulations

Simulations of the winning model found a decrease in the abs PE $r(58) = -.740$, $p < .001$, indicating that these models "learned" in a similar way to participants (**Figure 5C, bottom**). Surprisingly, simulations for

the best performing model indicated that Model 1 [*No Learning*] would have been the best strategy for this task (Figure 5B). After analysing the specific profiles used in experiments 1 and 4, we found that the correlation between the fashion model profiles and the standard population RP (i.e., the average student ratings) was higher compared to the profiles for experiment 1 (Supplementary Figure 8). This means that participants did not need all the information that the more complex models provided but rather that the simple *No Learning* model sufficed in capturing the main complexities of each profile. Interestingly, the second best performing model was Model 5 [*Fine granularity & Population RP*], indicating that the very next best strategy would have also used the standard population RP in favor of the stereotypic RP. The fact that participants still used Model 5-STE [*Fine granularity & Stereotype RP*] even though better strategies were available show both the pervasiveness of the stereotypic RPs and that similarity learning is a default strategy.

4. This leads me to a more general point. I found it a bit difficult to follow and distill the main narrative in the manuscript here. I note that prior reviewers of the paper had similar comments from the initial version; while I appreciate the extensive work that the authors did in responding to previous comments, I still feel as though the writing in the current version of the paper is convoluted in parts, particularly in the methods and results.

We understand your concerns and have attempted to resolve these issues. First, as suggested by reviewer 1, we have created a single figure for each experiment. This figure is placed directly below each experiments' respective results section. Second, we have attempted to reduce repetitions between methods and results, opting to keep details about methods largely contained in the methods. E.g., in the first part of the results we summarize our models and experimental task. We have thinned out this section and instead refer to the methods section for details. Third, we have tried to keep a very similar structure in the results section for each experiment. That is, we divide each results section with the same subheadings. These subheadings also roughly correspond to the panel titles in the figure.

5. I appreciate Reviewer 2's comment about the experimental paradigm being limited in scope, but I do believe that the authors are measured enough in the 'scope of models' section of the discussion now, which mitigates this concern in my view, and in general, aside from the points I raised above that align with Reviewer 2, I think the authors have done a nice job in their response to Reviewer 2's other comments.

Thank you.

References

Chierchia, G., Soukupová, M., Kilford, E. J., Griffin, C., Leung, J. T., Blakemore, S. J., & Palminteri, S. (2021). Choice-confirmation bias in reinforcement learning changes with age during adolescence.

Collins, A. G., & Shenhav, A. (2022). Advances in modeling learning and decision-making in neuroscience. *Neuropsychopharmacology*, *47*(1), 104-118.

Diaconescu, A. O., Mathys, C., Weber, L. A., Kasper, L., Mauer, J., & Stephan, K. E. (2017). Hierarchical prediction errors in midbrain and septum during social learning. *Social cognitive and affective neuroscience*, *12*(4), 618-634.

Lockwood, P. L., Apps, M. A., Valton, V., Viding, E., & Roiser, J. P. (2016). Neurocomputational mechanisms of prosocial learning and links to empathy. *Proceedings of the National Academy of Sciences*, *113*(35), 9763-9768.

Lockwood, P. L., & Klein-Flügge, M. C. (2021). Computational modelling of social cognition and behaviour—a reinforcement learning primer. *Social Cognitive and Affective Neuroscience*, *16*(8), 761-771.

Niv, Y., Daniel, R., Geana, A., Gershman, S. J., Leong, Y. C., Radulescu, A., & Wilson, R. C. (2015). Reinforcement learning in multidimensional environments relies on attention mechanisms. *Journal of Neuroscience*, *35*(21), 8145-8157.

Zaki, J., Kallman, S., Wimmer, G. E., Ochsner, K., & Shohamy, D. (2016). Social cognition as reinforcement learning: feedback modulates emotion inference. *Journal of Cognitive Neuroscience*, *28*(9), 1270-1282.

REVIEWERS' COMMENTS

Reviewer #1 (Remarks to the Author):

The authors have now addressed all of my concerns.

Reviewer #2 (Remarks to the Author):

I am content with the revisions. Congrats on an interesting paper!

One comment regarding one of your revisions:

"Studying social interactions in a well-controlled experimental setting forces limitations on a study (i.e., in the current experiment some limitations were necessary to establish the current modelling paradigm). First, even though we used RW learning models to capture participants' behaviour. The task, in which participants received direct and exact feedback, bears some resemblance to a supervised learning problem. Supervised models have thus far not been used to explain our task and could pose as interesting models for future research. A related limitation is that the task and thus feedback was received in exact numbers. In a social situation these number would normally be substituted for a verbal description (e.g., "I am very generous") or an action (e.g., sharing resources shows ones generosity)"

Please check the language here, there are multiple non-sentences and odd wordings. For example "First, even though we used RW learning models to capture participants' behaviour. The task.." The first sentence is incomplete.

Reviewer #3 (Remarks to the Author):

The authors successfully addressed my issues.

Reviewer #4 (Remarks to the Author):

I thank the authors for the detailed and rigorous approach they have taken in responding to my previous comments/concerns. I believe they have more than adequately addressed my previous concerns and recommend publication.

We would like to thank all reviewers for their time and effort, without your comments this manuscript would have been of much lesser quality!

Reviewer #1 (Remarks to the Author):

The authors have now addressed all of my concerns.

Reviewer #2 (Remarks to the Author):

I am content with the revisions. Congrats on an interesting paper!

One comment regarding one of your revisions:

"Studying social interactions in a well-controlled experimental setting forces limitations on a study (i.e., in the current experiment some limitations were necessary to establish the current modelling paradigm). First, even though we used RW learning models to capture participants' behaviour. The task, in which participants received direct and exact feedback, bears some resemblance to a supervised learning problem. Supervised models have thus far not been used to explain our task and could pose as interesting models for future research.

A related limitation is that the task and thus feedback was received in exact numbers. In a social situation these number would normally be substituted for a verbal description (e.g., "I am very generous") or an action (e.g., sharing resources shows ones generosity)"

Please check the language here, there are multiple non-sentences and odd wordings. For example "First, even though we used RW learning models to capture participants' behaviour. The task.." The first sentence is incomplete.

We would like to thank the reviewer for still reading our manuscript with a keen eye the third time around.

We have updated the writing in this section (page 28, lines: 817-827) and added the text below (changes highlighted in bold).

Studying social interactions in a well-controlled experimental setting forces limitations (i.e., in our experiments some limitations were necessary to establish the current modelling paradigm). **First, even though we used RW learning models to capture participants' behaviour, the task, in which participants received direct numerical feedback, bears some resemblance to a supervised learning problem.** Supervised models have thus far not been used to explain our task and could pose as interesting models for future research.

A related limitation is that our social learning task (including the given estimates and the received feedback) was presented using exact numbers. In many social situations, such estimates and feedback would be given in the form of verbal descriptions (e.g., "He is quite generous," "I am very generous"). Numerical feedback as used in our task is more common in school or work settings as part of performance evaluations (e.g., grades, scorings, etc.). Future studies should therefore investigate how people translate numerical into verbal evaluations (and vice versa) in similar social learning tasks.

Reviewer #3 (Remarks to the Author):

The authors successfully addressed my issues.

Reviewer #4 (Remarks to the Author):

I thank the authors for the detailed and rigorous approach they have taken in responding to my previous comments/concerns. I believe they have more than adequately addressed my previous concerns and recommend publication.